# SHARPNESS-AWARE MACHINE UNLEARNING

**Haoran Tang**
Department of Computer Science
Purdue University
`thr@purdue.edu`

**Rajiv Khanna**
Department of Computer Science
Purdue University
`rajivak@purdue.edu`

## ABSTRACT

We characterize the effectiveness of Sharpness-aware minimization (SAM) under machine unlearning scheme, where unlearning forget signals interferes with learning retain signals. While previous work prove that SAM improves generalization with noise memorization prevention, we show that SAM abandons such denoising property when fitting the forget set, leading to altered generalization depending on signal strength. We further characterize the signal surplus of SAM in the order of signal strength, which enables learning from less retain signals to maintain model performance and putting more weight on unlearning the forget set. Empirical studies show that SAM outperforms SGD with relaxed requirement for retain signals and can enhance various unlearning methods either as pretrain or unlearn algorithm. Motivated by our refined characterization of SAM unlearning and observing that overfitting can benefit more stringent sample-specific unlearning, we propose Sharp MinMax, which splits the model into two to learn retain signals with SAM and unlearn forget signals with sharpness maximization, achieving best performance. Extensive experiments show that SAM enhances unlearning across varying difficulties measured by memorization, yielding decreased feature entanglement between retain and forget sets, stronger resistance to membership inference attacks, and a flatter loss landscape. Our observations generalize to more noised data, different optimizers, and different architectures. Our code is available at https://github.com/HaoranTang/sharp-unlearn.

## 1 INTRODUCTION

Deep neural networks have grown so large and complex that retraining a model from scratch to forget even a few samples has become impractically costly in both computation and energy. This challenge has catalyzed the study of machine unlearning: methods that efficiently remove the influence of specific training data without full retraining, aiming to forget designated examples while preserving overall performance. Numerous unlearning strategies have been explored – from influence-based updates that subtract a data point's contribution (Izzo et al., 2021), to fine-tuning with targeted weight sparsification (Jia et al., 2023), to joint optimization approaches that explicitly balance "retain" vs. "forget" objectives by gradient ascent/descent on different data subsets (Kurmanji et al., 2023). However, a fundamental understanding of what makes unlearning effective remains elusive. Key questions persist: How should we trade off forgetting unwanted data versus retaining accuracy on the rest? How do different training algorithms influence unlearning dynamics? Why are some samples inherently harder to forget than others? In practice, the lack of principled answers has led to ad-hoc hyperparameter tuning and unpredictable behavior across tasks. In particular, when a model is simultaneously fed with conflicting retain and forget signals, these signals can interfere and even cancel out during training, hampering the unlearning process (Kurmanji et al., 2023). To date, there are few robust solutions to mitigate this interference, underscoring the need for a deeper theoretical foundation for machine unlearning.

Recent advances in learning theory and optimization hint at possible directions to tackle these issues. First, a signal-versus-noise perspective has provided new insight into model behavior: for example, Chen et al. (2023) formalize how networks learn meaningful patterns while ignoring or memorizing label noise, and Zhao et al. (2024) empirically identify factors that make certain data points harder to forget. Particularly relevant is the Sharpness-Aware Minimization (SAM) method (Foret et al., 2020) that has been shown to seek flatter loss minima and thereby dramatically reduce memoriza-

tion of noisy data, leading to improved generalization in noisy-label settings (Chen et al., 2023). These observations suggest that a model's ability to distinguish true signal from noise may be key to effective unlearning. An optimizer that naturally suppresses memorization of noise might also be better suited for forgetting specific examples when required. To investigate this hypothesis, we quantify each sample's memorization level using established metrics (Feldman, 2020; Feldman & Zhang, 2020), allowing us to rank the "forget set" by difficulty. This enables a controlled study of how different optimization algorithms perform when asked to forget data that the model has learned to varying extents.

We present a comprehensive theoretical and empirical study of machine unlearning through the combined lens of signal-noise decomposition and sharpness-aware optimization. We focus on the challenging scenario where both retain and forget samples are present in each training batch with mixed objectives, and we compare standard Stochastic Gradient Descent (SGD) to SAM in this context. Building on recent theoretical frameworks for ReLU networks (Kou et al., 2023), we derive rigorous results for a two-layer CNN that characterize the unlearning process under each optimizer. Our analysis yields several striking findings. (1) SAM's noise suppression can break down under unlearning: we prove that when tasked with intentionally forgetting a set of samples (treated as "noise"), SAM is forced by objective to abandon its usual denoising behavior – effectively overfitting to the forget set nearly as much as SGD does. This result challenges the expectation that flatter-minima methods would inherently excel at unlearning. (2) We establish formal guidelines for balancing retain vs. forget objectives: in particular, we derive the minimum retain-weighting factor $\alpha$ needed to prevent catastrophic forgetting of the kept data. Our theory shows that SAM can accomplish successful unlearning with a significantly smaller retain weight $\alpha$ than SGD, meaning SAM tolerates a stronger forgetting signal without sacrificing retained accuracy. In the regime of benign overfitting (where the model fits even noisy data without large generalization error), we quantify the gap in required $\alpha$ between SAM and SGD and prove it scales on the order of $O(\sqrt{d/n})$ (with $d$ the model dimension and $n$ the training set size). (3) Perhaps most surprisingly, our findings call for a re-examination of overfitting in unlearning. Contrary to conventional wisdom, we show that deliberate overfitting – in a controlled way that limits its impact on the rest of the data – can enhance the complete removal of those samples. This insight is especially relevant in stringent privacy or copyright scenarios, suggesting that the strict avoidance of overfitting may not always be optimal.

Our contributions can be summarized as follows:

**Theoretical Framework:** We introduce a rigorous analytical framework for machine unlearning based on signal-noise decomposition. This framework explicitly models the interplay between retain and forget signals. Using this lens, we analyze the behaviors of SGD versus SAM and prove that SAM's denoising advantage "shuts off" on forget data: when SAM is asked to unlearn labeled noise, it ends up overfitting to the forget set almost as much as SGD.

**Balancing Retain vs. Forget Objectives:** We derive provable guidelines for balancing the retain/forget trade-off. In particular, we identify the minimal value of the weighting ratio parameter $\alpha$ that guarantees sufficient retention of knowledge. We show that SAM requires a strictly smaller $\alpha$ than SGD to achieve effective unlearning. In the regime of benign overfitting for both the optimizers, we analytically bound the difference in required $\alpha$ on the order of $O(\sqrt{d/n})$.

**Empirical Validation:** Through extensive experiments on CIFAR-100 and ImageNet datasets, we validate our theoretical insights. We demonstrate that incorporating SAM into state-of-the-art unlearning methods consistently boosts forgetting efficacy while better preserving accuracy on the remaining data. Models optimized with SAM yield flatter loss landscapes and reduced entanglement between retained and forgotten samples, corroborating our theory that SAM distinguishes signal from noise better. We also observe that SAM-trained models are less vulnerable to membership inference attacks to forget set, indicating improved unlearning.

**Novel Unlearning Algorithm:** Finally, inspired by our analysis, we propose Sharp MinMax, a new unlearning approach that decouples the retain and forget objectives. Sharp MinMax splits the model into two cooperative parts: one is trained with SAM on the retained data, while the other performs sharpness maximization on the forget data to intentionally overfit those samples to ensure forgottenness. This design mitigates interference between retain and forget signals. Sharp Min-Max achieves state-of-the-art unlearning performance in our experiments, especially on challenging

high-memorization forget sets, where it significantly outperforms existing techniques in completely erasing the target data's influence.

## 2 PRELIMINARIES

### 2.1 DATA AND MODEL CONSTRUCTION

We construct a practical learning scenario which distinguishes between useful and unrelated signals from inputs. Similar constructions have been adopted in previous work (Kou et al., 2023; Chen et al., 2023) with rich notation. For convenience, we summarize a table of notation in App. C. Consider learning binary classification with label $y \in \{\pm 1\}$ using a two-layer CNN on image training data set $\mathcal{S} = \{(\mathbf{x}_i, y_i)\}_{i \in [n]} \sim \mathcal{D}$. Each image consists of $P$ patches and assign randomly one of them as the signal $y_i \boldsymbol{\varphi}$ for label $y_i$ and the universal signal vector $\boldsymbol{\varphi} \in \mathbb{R}^d$, and represent other patches by the noise vector $\boldsymbol{\xi}_i \in \mathbb{R}^d \sim \mathcal{N}(\mathbf{0}, \sigma_p^2 \mathbf{I})$. Thus, each input image is vectorized as $\mathbf{x}_i = [\boldsymbol{\xi}_i, ..., y_i \boldsymbol{\varphi}, ..., \boldsymbol{\xi}_i] \in \mathbb{R}^{P \times d}$, where $y_i \boldsymbol{\varphi}$ can appear at any position.

The second layer of CNN is fixed as $\pm 1/m$ respectively for $m$ convolutional filters. The two-classes network can be expressed as $f(\mathbf{W}, \mathbf{x}) = f_{+1}(\mathbf{W}_{+1}, \mathbf{x}) - f_{-1}(\mathbf{W}_{-1}, \mathbf{x})$, where

$$f_j(\mathbf{W}_j, \mathbf{x}) = \frac{1}{m} \sum_{r=1}^{m} \sum_{p=1}^{P} \sigma(\langle \mathbf{w}_{j,r}, \mathbf{x} \rangle) = \frac{1}{m} \sum_{r=1}^{m} \sigma(\langle \mathbf{w}_{j,r}, y\boldsymbol{\varphi} \rangle) + (P-1)\sigma(\langle \mathbf{w}_{j,r}, \boldsymbol{\xi} \rangle). \quad (1)$$

Here $\sigma$ denotes ReLU activation, $\mathbf{w}_{j,r} \in \mathbb{R}^d$ denotes the weight for the $r$-th filter, and $\mathbf{W}_j$ is the collection of model weights for $j = \pm 1$. We train this CNN with cross-entropy loss $\mathcal{L}(\mathbf{W}, \mathcal{S})$. Denote $\mathbf{w}_{j,r}^{(t,b)}$ for $j \in \{\pm 1\}, r \in [m]$ the convolutional filter at the $b$-batch of $t$-th epoch of SGD. We decompose the weight update into learning signal and noise coefficients $\kappa_{j,r}^{(t,b)}, \zeta_{j,r,i}^{(t,b)}$ for learning the signal and the noise respectively, such that

$$\mathbf{w}_{j,r}^{(t,b)} = \mathbf{w}_{j,r}^{(0,0)} + j \cdot \kappa_{j,r}^{(t,b)} \cdot \boldsymbol{\varphi} \|\boldsymbol{\varphi}\|_2^{-2} + (P-1)^{-1} \sum_{i=1}^{n} \zeta_{j,r,i}^{(t,b)} \cdot \boldsymbol{\xi}_i \|\boldsymbol{\xi}_i\|_2^{-2}, \quad (2)$$

where the learning goal is to increase $\kappa_{j,r}^{(t,b)}$ and decrease $\zeta_{j,r,i}^{(t,b)}$. This construction also extends to multiclass classification considering one vs. all setting with $K$ binary classification problems. For readability, we abbreviate subscript $j, r$ and replace superscript $(t, b)$ with time vector $\mathbf{t}$ in following sections, and leave full notation to proofs in the Appendix.

### 2.2 SIGNAL-TO-NOISE UNLEARNING

Given a pretrained model $f_{\mathcal{A}}^{T_1}$ by algorithm $\mathcal{A}$ for $T_1$ epochs on $\mathcal{S}$, machine unlearning aims to eliminate the influence of forget set $\mathcal{F} \subseteq \mathcal{S}$ to the model training, while maintain generalizability to unseen data without compromising performance on the remaining retain set $\mathcal{R} = \mathcal{S} \setminus \mathcal{F}$. Denote the unlearned model as $f_{\mathcal{U}}^{T_2}$ by unlearning algorithm $\mathcal{U}$, which is initialized as $f_{\mathcal{A}}^{T_1}$ and unlearned for $T_2$ epochs. We consider unlearning a small portion of $\mathcal{S}$ with much less expense than retraining the model from scratch on $\mathcal{R}$, so $|\mathcal{F}| < |\mathcal{R}|$ and $T_2 < T_1$.

**Random Label** (RL) (Graves et al., 2021) aims to unlearn by finetuning on $\mathcal{S}$ but with $\mathcal{F}$'s labels randomly flipped in each epoch. It naturally fits into our setup as label-flipped $\mathcal{F}$ become the noise, and motivates us to investigate unlearning algorithms under the same theoretical framework. The gradient update of $\kappa^{\mathbf{t}}$ and $\zeta_i^{\mathbf{t}}$ of class $j$ can be expressed as

$$\kappa^{\mathbf{t+1}} = \kappa^{\mathbf{t}} - \frac{\eta \|\boldsymbol{\varphi}\|_2^2}{Bm} \left[ \sum_{i \in \mathcal{I}_{\mathbf{t}}^{\mathcal{R}}} \ell_i'^{\mathbf{t}} \sigma'(\langle \mathbf{w}^{\mathbf{t}}, \widehat{y_i}\boldsymbol{\varphi} \rangle) - \sum_{i \in \mathcal{I}_{\mathbf{t}}^{\mathcal{F}}} \ell_i'^{\mathbf{t}} \sigma'(\langle \mathbf{w}^{\mathbf{t}}, \widehat{y_i}\boldsymbol{\varphi} \rangle) \right],$$

$$\zeta_i^{\mathbf{t+1}} = \zeta_i^{\mathbf{t}} - \frac{\eta (P-1)^2 \|\boldsymbol{\xi}_i\|_2^2}{Bm} \cdot \ell_i'^{\mathbf{t}} \sigma'(\langle \mathbf{w}^{\mathbf{t}}, \boldsymbol{\xi}_i \rangle) \cdot \mathrm{sgn}(y_i = j), \quad (3)$$

where $B, \eta$ denote the batch size and learning rate, $\mathrm{sgn}(\cdot)$ denotes $\pm 1$ sign function, $\mathcal{I}_{\mathbf{t}}^{\mathcal{R}}$ and $\mathcal{I}_{\mathbf{t}}^{\mathcal{F}}$ denote batch samples from $\mathcal{R}$ and $\mathcal{F}$ at $\mathbf{t}$, respectively. In each iteration, $\mathcal{I}_{\mathbf{t}}^{\mathcal{F}}$ aims to erase its signal in $\kappa^{\mathbf{t}}$, while $\boldsymbol{\xi}_i$ reinforces or decreases $\zeta_i^{\mathbf{t}}$ update depending on label agreement.

**Negative Gradient** (NegGrad) (Kurmanji et al., 2023) unlearns $\mathcal{F}$ using gradient ascent while gradient-descending on $\mathcal{R}$. Unlike RL or other $\mathcal{U}$ that aim at random guessing, ascent-based unlearning encourages misclassification by its objective:

$$\mathcal{L}_{\text{NegGrad}}(\mathbf{W}, \mathcal{R}, \mathcal{F}) = \frac{1}{|\mathcal{R}|} \sum_{i \in \mathcal{R}} \alpha \ell \left(y_i f\left(\mathbf{W}, \mathbf{x}_i\right)\right) - \frac{1}{|\mathcal{F}|} \sum_{i \in \mathcal{F}} (1 - \alpha) \ell\left(y_i f\left(\mathbf{W}, \mathbf{x}_i\right)\right). \quad (4)$$

Minimizing $\mathcal{L}_{\text{NegGrad}}$ induces competing gradients, canceling each other during $\kappa, \zeta$ update. $\alpha$ serves as a weight coefficient that accounts for the size imbalance between $\mathcal{R}$ and $\mathcal{F}$. To synchronously optimize the model with retain and forget samples, we draw $B$ samples from both subsets each batch and train for $|\mathcal{R}|/B$ batches. Thus, forget samples' signals are relatively enlarged by a fraction of $|\mathcal{R}|/|\mathcal{F}|$ due to repetition. Heuristically, $\alpha \propto |\mathcal{R}|/(|\mathcal{F}| + |\mathcal{R}|)$.

## 2.3 DENOISING PROPERTY OF SAM

Sharpness-Aware Minimization (SAM) (Foret et al., 2020) aims to minimize a perturbed empirical loss at the worst point in the neighborhood of $\mathbf{W}$, solving the following optimization problem:

$$\min_{\mathbf{W}} \mathcal{L}(\mathbf{W}, \mathcal{S}) + \left[\max_{\widehat{\epsilon}} \mathcal{L}(\mathbf{W} + \widehat{\epsilon}, \mathcal{S}) - \mathcal{L}(\mathbf{W}, \mathcal{S})\right], \quad (5)$$

for a controlled perturbation $\widehat{\epsilon}$. It ensures a uniformly low training loss and avoids sharp landscape. While both SGD and SAM learn a sufficient signal with $\kappa^{T_1} = \Omega(1)$ after $T_1$ epochs, Chen et al. (2023) prove that SAM outperforms SGD by noise suppression and SAM upper bounds $\zeta_i^{T_1}$ by $O(1)$ while SGD is dimension dependent $O(\log d)$. The key difference stems from the noise memorization prevention of SAM. Given the perturbation term $\widehat{\epsilon}^{\mathbf{t}}$ in SAM for class $j$:

$$\widehat{\epsilon}^{\mathbf{t}} = \frac{\tau}{m} \sum_{i \in \mathcal{I}_{\mathbf{t}}} \sum_{p \in [P]} \ell_i'^{\mathbf{t}} j \cdot y_i \sigma'(\langle \mathbf{w}^{\mathbf{t}}, \mathbf{x}_{i,p} \rangle) \mathbf{x}_{i,p} \cdot \left\|\nabla_{\mathbf{W}} \mathcal{L}(\mathbf{W}^{\mathbf{t}}, \mathcal{I}_{\mathbf{t}})\right\|_F^{-1}, \quad (6)$$

consider ReLU activation at any fixed iterate $\mathbf{w}^{\mathbf{t}}$ for SGD: $\langle \mathbf{w}^{\mathbf{t}}, \boldsymbol{\xi}_k \rangle \geq 0$ vs. SAM: $\langle \mathbf{w}^{\mathbf{t}} + \widehat{\epsilon}^{\mathbf{t}}, \boldsymbol{\xi}_k \rangle$ for $k \in \mathcal{I}_{\mathbf{t}}, j = y_k$. SAM's $\langle \mathbf{w}^{\mathbf{t}} + \widehat{\epsilon}^{\mathbf{t}}, \boldsymbol{\xi}_k \rangle$ expands to $\langle \mathbf{w}^{\mathbf{t}}, \boldsymbol{\xi}_k \rangle + \langle \widehat{\epsilon}^{\mathbf{t}}, \boldsymbol{\xi}_k \rangle$, where $\langle \widehat{\epsilon}^{\mathbf{t}}, \boldsymbol{\xi}_k \rangle$ is proven to be sufficiently negative to cancel $\langle \mathbf{w}^{\mathbf{t}}, \boldsymbol{\xi}_k \rangle$ by selecting a proper $\tau$, thus deactivating the noise (Chen et al., 2023). This effectively prevents SAM from learning from the noise which would lead to harmful overfitting for SGD. We are curious about whether SAM improves unlearning: a flatter landscape can make learning easier, then it should make unlearning easier too despite a reverse sign. But is it a simple adaptation, and can we straightforwardly extend previous theories and findings to develop unlearning algorithms?

## 3 SHARPNESS-AWARE UNLEARNING

We first show that the SAM's noise memorization prevention in Sec. 2.3 does not fully hold when SAM is used with NegGrad for gradient ascent on $\mathcal{F}$. Specifically, SAM overfits to forget signals as much as SGD, while maintaining its denoising property on $\mathcal{R}$. Based on this result, we derive refined test error bounds for SGD and SAM under NegGrad and characterize the different $\alpha$ thresholding between SGD and SAM for unlearning. Although SAM continues to improve unlearning and maintain generalizability, the altered activation patterns and unlearning behaviors are not captured by previous works, as SAM is forced to fit forget signals (viewed as noise) by NegGrad objective. This leads to divergent behaviors on $\mathcal{R}$ and $\mathcal{F}$, which can be of independent interest.

### 3.1 NEGGRAD REVISITED

Unlike RL, the mutual interference between $\mathcal{F}$ and $\mathcal{R}$ under NegGrad additionally affects $\zeta$ update. The update rules for $\kappa^{\mathbf{t}}$ and $\zeta^{\mathbf{t}}$ under NegGrad now become:

$$\kappa^{\mathbf{t+1}} = \kappa^{\mathbf{t}} - \frac{\eta \|\boldsymbol{\varphi}\|_2^2}{Bm} \left[\alpha \sum_{i \in \mathcal{I}_{\mathbf{t}}^{\mathcal{R}}} \nabla_{\boldsymbol{\varphi}_i} - (1 - \alpha) \sum_{i \in \mathcal{I}_{\mathbf{t}}^{\mathcal{F}}} \nabla_{\boldsymbol{\varphi}_i}\right],$$

$$\zeta^{\mathbf{t+1}} = \zeta^{\mathbf{t}} - \frac{\eta (P - 1)^2}{Bm} \left[\alpha \sum_{i \in \mathcal{I}_{\mathbf{t}}^{\mathcal{R}}} \nabla_{\boldsymbol{\xi}_i} - (1 - \alpha) \sum_{i \in \mathcal{I}_{\mathbf{t}}^{\mathcal{F}}} \nabla_{\boldsymbol{\xi}_i}\right], \quad (7)$$

where $\nabla_{\boldsymbol{\varphi}_i} = \ell_i'^{\mathbf{t}}\sigma'(\langle \mathbf{w^t} + \delta, y_i\boldsymbol{\varphi}\rangle), \nabla_{\boldsymbol{\xi}_i} = \mathrm{sgn}(y_i = j)\|\boldsymbol{\xi}_i\|_2^2\ell_i'^{\mathbf{t}}\sigma'(\langle \mathbf{w^t} + \delta, \boldsymbol{\xi}_i\rangle)$, and $\delta = \widehat{\boldsymbol{\epsilon}}^{\mathbf{t}}$ for SAM and 0 for SGD. In plain words, a retain sample of class $j$ causes a decrease in $\zeta_j$, discouraging memorizing noise for the correct class, while another retain sample of class $-j$ causes an increase in $\zeta_j$, encouraging $w_j$ to use $\boldsymbol{\xi}_i$ to distinguish class $j$ from $-j$. Conversely, a sample $i \in \mathcal{F}$ of class $j$, which we want to predict $-j$ in ascent-based unlearning, will increase $\zeta_j$ and encourage $w_j$ to use noise $\boldsymbol{\xi}_i$ in a way that harms class $j$, and vice versa. Similar intuition also applies to $\kappa$. The interference in $\zeta$ update will alter SAM's behaviors towards forget signals as summarized in Lemma 3.1.

**Lemma 3.1** *(Noise memorization of $\mathcal{F}$ by SAM under NegGrad). Under the NegGrad scheme and the Assumption D.1 holds, for class $j$ we have that if for SGD: $\langle \mathbf{w^t}, \boldsymbol{\xi}_k\rangle \geq 0, k \in \mathcal{I}_{\mathbf{t}}^{\mathcal{R}}$ and $j = y_k$, then for SAM: $\langle \mathbf{w^t} + \widehat{\boldsymbol{\epsilon}}^{\mathbf{t}}, \boldsymbol{\xi}_k\rangle < 0$. However, if for SGD: $\langle \mathbf{w^t}, \boldsymbol{\xi}_k\rangle \geq 0, k \in \mathcal{I}_{\mathbf{t}}^{\mathcal{F}}$ and $j = y_k$, then for SAM: $\langle \mathbf{w^t} + \widehat{\boldsymbol{\epsilon}}^{\mathbf{t}}, \boldsymbol{\xi}_k\rangle > 0$.*

See proof in App. D.2. Because the activation patterns on $\mathcal{I}_{\mathbf{t}}^{\mathcal{R}}$ and $\mathcal{I}_{\mathbf{t}}^{\mathcal{F}}$ diverge, SAM continues to suppress noise memorization and leverage its sharpness-aware updates when fitting $\mathcal{R}$, but "falls back" to SGD-like behavior on $\mathcal{F}$. This split yields two distinct sets of bounds on $\kappa$ and $\zeta$ for $\mathcal{R}$ and $\mathcal{F}$, which lead to separate test errors shown in App. D.1 and D.2. However, given a pretrained model $f_{\mathcal{A}}^{T_1}$ with $\kappa^{T_1} > 0$ to start unlearning, **as long as retain signals weighted by $\alpha$ dominate, the signal strength will remain sufficient and continue to grow**. This is shown in Chen et al. (2023) when the signal strength is saturated at $T < T_1$. We can thus choose $\alpha$ threshold based on this principle. With proper forget-retain size ratio, results in Chen et al. (2023) still hold: SGD's test error converges when signal strength is sufficient, but can't be upper bounded otherwise; SAM's test error converges either way. $\beta$ serves as a knob to control the convergence rate:

**Theorem 3.2** *(SGD test error under NegGrad). Under Assumption D.1, for any $\epsilon > 0$ and $1 > \alpha \geq |\mathcal{R}|/(|\mathcal{F}| + |\mathcal{R}|) := \beta > 0.5$, then with probability at least $1 - \delta$, the training loss converges: $\mathcal{L}(\mathbf{W}^T, \mathcal{D}) \leq \epsilon$. Moreover, if $\|\boldsymbol{\varphi}\|_2 \geq C_1 d^{1/4}n^{-1/4}P\sigma_p$, we have the test error $\mathcal{L}^{test}(\mathbf{W}^T, \mathcal{D}) \leq \epsilon$. If $\|\boldsymbol{\varphi}\|_2 \leq C_3 d^{1/4}n^{-1/4}P\sigma_p$, we have $\lim_{\beta \to 1} \mathcal{L}^{test}(\mathbf{W}^{T_2}, \mathcal{D}) \geq 0.1$, and $\lim_{\beta \to 0.5} \mathcal{L}^{test}(\mathbf{W}^{T_2}, \mathcal{D}) \geq 0.05$.*

**Theorem 3.3** *(SAM test error under NegGrad). Under Assumption D.1, for any $\epsilon > 0$ and $1 > \alpha \geq |\mathcal{R}|/(|\mathcal{F}| + |\mathcal{R}|) := \beta > 0.5$, choose $\tau = \Theta(\frac{m\sqrt{B}}{P\sigma_p\sqrt{d}})$. Then with probability at least $1 - \delta$, the training loss converges: $\mathcal{L}(\mathbf{W}^T, \mathcal{D}) \leq \epsilon$. Moreover, if $\|\boldsymbol{\varphi}\|_2 \geq C_1 d^{1/4}n^{-1/4}P\sigma_p$, we have $\lim_{\beta \to 1} \mathcal{L}^{test}(\mathbf{W}^T, \mathcal{D}) \leq \epsilon$. If $\Omega(1) \leq \|\boldsymbol{\varphi}\|_2 \leq C_3 d^{1/4}n^{-1/4}P\sigma_p$: we still have $\lim_{\beta \to 1} \mathcal{L}^{test}(\mathbf{W}^T, \mathcal{D}) \leq \epsilon$.*

See proofs in App. D.1 and D.2. Together, these theorems describe how SGD and SAM behave when retain signals dominate. For SAM, if $\|\boldsymbol{\varphi}\|_2 \leq C_3 d^{1/4}n^{-1/4}P\sigma_p$, it will suffer harmful overfitting to $\mathcal{F}$. However, as long as $\alpha \geq |\mathcal{R}|/(|\mathcal{F}| + |\mathcal{R}|)$ and $\|\boldsymbol{\varphi}\|_2 \geq \Omega(1)$, learning on $\mathcal{R}$ guarantees overall benign training and yields a bounded test error. Under the same condition, Corollary 3.3.1 concludes that while the signal coefficient continues to grow for both SGD and SAM, SGD's noise accumulation is loosely bounded by model dimension, while SAM's by $O(1)$:

**Corollary 3.3.1** *($\kappa, \zeta$ update under NegGrad). Under the NegGrad, if $\alpha \geq |\mathcal{R}|/(|\mathcal{F}| + |\mathcal{R}|)$, since $\kappa^{T_1} = \Omega(1)$, both SGD and SAM continue to grow. Given the learned $\zeta^{T_1}$, SGD continues to overfit the noise with $O(\log d)$, while SAM overfit the noise from $\mathcal{F}$ with $O(\log d)$ and from $\mathcal{R}$ with $O(1)$.*

See proof in App. D.3. Finally, we characterize the differed choice of $\alpha$ for SGD and SAM as SAM learns signal more efficiently. We also reveal that $\alpha$ depends not only on forget-retain size ratio as commonly conjectured, but also on the signal strength, and thus the dimensionality of the problem:

**Lemma 3.4** *(Signal-surplus of SAM under NegGrad). Under the NegGrad, for any $\boldsymbol{\varphi}$ where $\|\boldsymbol{\varphi}\|_2 \geq \Omega(1)$, SAM exhibits faster signal learning on $\mathcal{R}$: $\Delta_{epoch}^{SAM}\kappa/\Delta_{epoch}^{SGD}\kappa = \Theta(\|\boldsymbol{\varphi}\|_2^2)$.*

See proof in App. D.4. As a result, SAM relies on a more relaxed $\alpha$ threshold than SGD due to faster signal learning. For SGD to achieve the same signal learning performance as SAM, we need to scale up $\alpha^{\mathrm{SGD}}$ to satisfy $\alpha^{\mathrm{SGD}}/\alpha^{\mathrm{SAM}} = \Theta(\|\boldsymbol{\varphi}\|_2^2)$. If $\|\boldsymbol{\varphi}\|_2 \geq C_1 d^{1/4}n^{-1/4}P\sigma_p$ and both SGD

and SAM achieve benign overfitting, then given the extra signal learning from $\mathcal{R}$, SAM results in faster $\kappa$ update and a surplus signal of $\Theta(d^{1/2}|\mathcal{R}|^{-1/2}P^2\sigma_p^2)$ in each unlearning epoch.

## 3.2 SHARP MINMAX

In Sec. 3.1, we showed that SAM is provably better on out of sample test errors under NegGrad, and we empirically verify that SAM achieves better unlearning performance in Sec. 4. But how does the refined characterization matter, given maintained test error conclusions? Jointly with empirical observations, the altered behaviors of SAM on $\mathcal{F}$ motivates new unlearning algorithms. Our experiments show that SAM+NegGrad attains higher forget accuracy than SGD+NegGrad, forgetting less effectively. This finding forces us to reconsider the conventional view that overfitting is always detrimental: while overfitting indeed harms generalization, it may be beneficial when the goal is to remove specific samples from a model. Consequently, for abstract concept forgetting we continue to demand strong generalization; but for stringent scenarios—where exact sample removal is mandated by privacy or compliance constraints—a model's tendency to overfit can actually enhance its unlearning of those exact points. The divergent behaviors under SAM+NegGrad motivates the following new algorithm: we can split a portion of model parameters to purposefully overfit to $\mathcal{F}$, denoted as the forget model $\mathbf{W}_\mathcal{F}$, while leaving the rest as the retain model $\mathbf{W}_\mathcal{R}$ to maximally maintain the model utility by leveraging SAM purely on $\mathcal{R}$. Motivated by how SGD with sharper minima tends to forget better, we propose Sharp MinMax to intentionally optimize for sharper-than-SGD minima with the purpose of overfitting to forget signals for unlearning. Inspired by Kim et al. (2023), we leverage sharpness maximization on $\mathbf{W}_\mathcal{F}$:

$$\min_{\mathbf{W}_\mathcal{F}} \mathcal{L}(\mathbf{W}_\mathcal{F}, \mathcal{F}) - \left[\max_{\widehat{\boldsymbol{\epsilon}}} \mathcal{L}(\mathbf{W}_\mathcal{F} + \widehat{\boldsymbol{\epsilon}}, \mathcal{F}) - \mathcal{L}(\mathbf{W}_\mathcal{F}, \mathcal{F})\right], \tag{8}$$

resulting in a sharper landscape that harms the generalization by overfitting. We apply weight masking based on gradient magnitudes (Fan et al., 2023) to divide our model into $\mathbf{W}_\mathcal{R}, \mathbf{W}_\mathcal{F}$ during optimization. Specifically, we pass $\mathcal{F}$ to $f_\mathcal{A}$ once, accumulate gradients for each parameter, and check top parameters with smallest magnitudes cut off by a given percentage. We then apply SAM on $\mathbf{W}_\mathcal{R}$ and sharpness maximization on $\mathbf{W}_\mathcal{F}$. The retain model with SAM is already characterized by Chen et al. (2023), while $\mathbf{W}_\mathcal{F}$ requires a stronger signal strength than SGD to avoid harmful overfitting. See implementation details in App. E.2.

## 3.3 QUANTIFYING UNLEARNING DIFFICULTY WITH MEMORIZATION

We examine the effectiveness of unlearning $\mathcal{U}$ based on memorization, which sufficiently reveals the difficulty of unlearning (Zhao et al., 2024). Feldman & Zhang (2020) define the degree to which a sample is memorized by a pretraining $\mathcal{A}$ on example $(\mathbf{x}_i, y_i)$ from $\mathcal{S}$ as the memorization score:

$$\text{mem}(\mathcal{A}, \mathcal{S}, i) := \Pr_{f \leftarrow \mathcal{A}(\mathcal{S})}\left[f(\mathbf{W}, \mathbf{x}_i) = y_i\right] - \Pr_{f \leftarrow \mathcal{A}(\mathcal{S}\setminus i)}\left[f(\mathbf{W}, \mathbf{x}_i) = y_i\right], \tag{9}$$

where $\mathcal{S} \setminus i$ denotes $\mathcal{S}$ with the sample $(\mathbf{x}_i, y_i)$ removed. Samples of high-memorization scores can be atypical samples which model usually learns later in the training process after more updates to the model than typical ones. Thus unlearning them would be harder and may require more iterations of unlearning steps which may impact the model performance on the retain distribution. The converse is true for samples of low-memorization scores. We can hence construct $\mathcal{F}$ of varying unlearning difficulties based on memorization scores to comprehensively evaluate $\mathcal{U}$.

## 4 EMPIRICAL STUDY

We conduct major experiments on CIFAR-100 (Krizhevsky et al., 2009) and ImageNet-1K (Russakovsky et al., 2015) using ResNet-50 (He et al., 2016), and adopt pre-computed memorization scores for from Feldman & Zhang (2020) to generate $\mathcal{F}$ of different difficulties with $|\mathcal{F}| \approx 5\%|\mathcal{S}|$, denoted as $[\mathcal{F}_{\text{high}}, \mathcal{F}_{\text{mid}}, \mathcal{F}_{\text{low}}]$. For both pretraining and unlearning, we adopt SAM (Foret et al., 2020) with $\rho = 0.1$ and Adaptive SAM (ASAM) (Kwon et al., 2021) with $\rho = [0.1, 1.0]$. We ensure same optimal hyper-paprameters for each comparable [SGD,SAM] pair. See details in App. E.

**Evaluation.** We follow previous work (Triantafillou et al., 2024; Zhao et al., 2024) to measure the tug-of-war tradeoff between forgetting and retaining of $f_\mathcal{U}$ based on accuracy $\text{Acc}(\theta, \mathcal{D})$, with the retrained model $f_{\mathcal{A}(\mathcal{R})}$ as reference:

$$\begin{aligned}\text{ToW}(f_\mathcal{U}) =&(1 - (\text{Acc}(f_{\mathcal{A}(\mathcal{R})}, \mathcal{R}) - \text{Acc}(f_\mathcal{U}, \mathcal{R}))) \cdot (1 - (\text{Acc}(f_\mathcal{U}, \mathcal{F}) - \text{Acc}(f_{\mathcal{A}(\mathcal{R})}, \mathcal{F}))) \\ &\cdot (1 - (\text{Acc}(f_{\mathcal{A}(\mathcal{R})}, \mathcal{D}_{\text{test}}) - \text{Acc}(f_\mathcal{U}, \mathcal{D}_{\text{test}}))), \text{ with test transforms on } \mathcal{R}, \mathcal{F}.\end{aligned} \tag{10}$$

Table 1: ToW(%) $\uparrow$ of unlearning on ImageNet-1K and CIFAR-100. For each $(\mathcal{U}, \mathcal{A})$ pair, we report ToW of each $\mathcal{F}$ and compute averages. SAM consistently improves current unlearning methods.

| ImageNet | $\mathcal{A}$=SGD | | | | $\mathcal{A}$=ASAM 0.1 | | | | $\mathcal{A}$=ASAM 1.0 | | | | $\mathcal{A}$=SAM 0.1 | | | |
|---|---|---|---|---|---|---|---|---|---|---|---|---|---|---|---|---|
| Unlearn $\mathcal{U}$ | High | Mid | Low | AVG | High | Mid | Low | AVG | High | Mid | Low | AVG | High | Mid | Low | AVG |
| NegGrad | 78.764 | 84.199 | 88.515 | 83.826 | 78.426 | 83.93 | 86.651 | 83.002 | 78.522 | 83.929 | 89.947 | 84.133 | 78.03 | 84.176 | 88.839 | 83.682 |
| +ASAM 0.1 | 78.52 | 84.113 | 89.188 | 83.94 | 78.366 | 84.07 | 89.098 | 83.845 | 78.762 | 84.267 | 90.579 | 84.536 | 78.083 | 84.062 | 89.973 | 84.039 |
| +ASAM 1.0 | 78.966 | 83.389 | 92.174 | **84.843** | 78.975 | 83.358 | 91.843 | **84.725** | 78.027 | 83.326 | 92.772 | **84.708** | 77.762 | 83.284 | 92.617 | **84.554** |
| +SAM 0.1 | 77.898 | 82.985 | 92.841 | 84.575 | 78.301 | 83.04 | 91.722 | 84.354 | 77.388 | 82.473 | 93.429 | 84.43 | 76.807 | 82.587 | 92.829 | 84.074 |
| RL | 74.598 | 86.617 | 86.714 | 82.643 | 74.857 | 86.462 | 86.192 | 82.504 | 74.317 | 86.813 | 87.630 | 82.92 | 74.055 | 86.715 | 88.594 | 83.121 |
| +ASAM 1.0 | 74.951 | 85.581 | 91.069 | **83.867** | 75.221 | 85.473 | 90.425 | **83.707** | 73.950 | 85.393 | 91.516 | **83.62** | 73.579 | 85.494 | 91.74 | **83.604** |
| SalUn | 44.981 | 71.839 | 95.008 | 70.609 | 46.104 | 71.735 | 94.652 | 70.83 | 45.814 | 72.308 | 95.116 | 71.079 | 46.006 | 72.419 | 95.218 | 71.214 |
| +ASAM 1.0 | 45.998 | 71.554 | 95.628 | **71.06** | 46.938 | 71.268 | 95.224 | **71.143** | 45.856 | 71.695 | 95.924 | **71.158** | 46.358 | 72.034 | 95.791 | **71.394** |

| CIFAR100 | $\mathcal{A}$=SGD | | | | $\mathcal{A}$=ASAM 0.1 | | | | $\mathcal{A}$=ASAM 1.0 | | | | $\mathcal{A}$=SAM 0.1 | | | |
|---|---|---|---|---|---|---|---|---|---|---|---|---|---|---|---|---|
| Unlearn $\mathcal{U}$ | High | Mid | Low | AVG | High | Mid | Low | AVG | High | Mid | Low | AVG | High | Mid | Low | AVG |
| NegGrad | 78.334 | 83.335 | 83.718 | 81.796 | 79.277 | 86.454 | 88.637 | 84.789 | 77.274 | 78.59 | 85.443 | 80.436 | 67.826 | 74.145 | 76.374 | 72.78 |
| +ASAM 0.1 | 78.131 | 82.846 | 86.78 | 82.586 | 80.336 | 87.539 | 87.671 | 85.182 | 77.331 | 79.074 | 88.039 | 81.482 | 70.054 | 74.158 | 78.087 | 74.1 |
| +ASAM 1.0 | 80.806 | 81.465 | 87.052 | 83.108 | 82.196 | 84.391 | 90.502 | **85.696** | 78.731 | 79.264 | 93.249 | **83.748** | 72.518 | 75.653 | 86.759 | **78.31** |
| +SAM 0.1 | 81.331 | 75.059 | 94.151 | **83.514** | 82.86 | 77.94 | 94.179 | 84.993 | 74.704 | 70.898 | 95.898 | 80.5 | 65.080 | 66.089 | 95.078 | 75.416 |
| L1-Sparse | 63.448 | 68.686 | 53.991 | 62.042 | 63.699 | 72.775 | 60.34 | 65.605 | 61.252 | 68.197 | 61.47 | 63.64 | 65.258 | 71.941 | 59.014 | 65.404 |
| +ASAM 1.0 | 66.903 | 75.554 | 58.967 | **67.141** | 66.213 | 77.119 | 66.697 | **70.01** | 65.117 | 73.754 | 62.517 | **67.129** | 63.051 | 74.556 | 65.117 | **67.575** |
| SCRUB | 58.418 | 76.125 | 12.708 | 49.084 | 67.163 | 79.09 | 10.823 | 52.359 | 57.816 | 73.176 | 58.483 | 63.158 | 43.246 | 68.433 | 17.368 | 43.016 |
| +ASAM 1.0 | 50.313 | 73.353 | 97.631 | **73.766** | 60.515 | 80.204 | 97.508 | **79.409** | 48.569 | 73.09 | 97.776 | **73.145** | 18.137 | 61.618 | 97.933 | **59.229** |
| RL | 68.464 | 84.395 | 72.4 | 75.086 | 64.518 | 80.615 | 69.711 | 71.481 | 66.689 | 86.411 | 69.677 | 74.259 | 64.391 | 85.481 | 70.55 | 73.474 |
| +ASAM 1.0 | 69.952 | 86.779 | 74.409 | **77.047** | 66.909 | 86.557 | 69.375 | **74.280** | 69.73 | 91.124 | 80.321 | **80.392** | 72.884 | 88.633 | 78.066 | **79.861** |
| SalUn | 69.926 | 83.056 | 71.73 | 74.904 | 66.541 | 83.377 | 71.95 | 73.956 | 67.355 | 89.768 | 79.095 | 78.739 | 69.671 | 90.495 | 75.281 | 78.482 |
| +ASAM 1.0 | 73.268 | 92.225 | 88.175 | **84.556** | 71.426 | 89.182 | 86.13 | **82.246** | 67.715 | 93.401 | 89.289 | **83.468** | 70.933 | 92.914 | 86.477 | **83.441** |

Table 2: MIA (%) $\downarrow$ correctness to $\mathcal{F}$ on CIFAR-100. We enhance each $\mathcal{U}$ with ASAM 1.0 and observe consistent improvement.

| | $\mathcal{A}$=SGD | | | | $\mathcal{A}$=ASAM 0.1 | | | | $\mathcal{A}$=ASAM 1.0 | | | | $\mathcal{A}$=SAM 0.1 | | | |
|---|---|---|---|---|---|---|---|---|---|---|---|---|---|---|---|---|
| Unlearn $\mathcal{U}$ | High | Mid | Low | AVG | High | Mid | Low | AVG | High | Mid | Low | AVG | High | Mid | Low | AVG |
| L1-Sparse | 94.733 | 63.233 | 8.6 | 55.522 | 94.933 | 61.367 | 4.0 | 53.433 | 93.833 | 62.067 | 5.8 | 53.9 | 92.867 | 60.033 | 5.033 | 52.644 |
| +ASAM 1.0 | 94.267 | 58.5 | 5.5 | **52.756** | 94.3 | 57.3 | 3.6 | **51.733** | 93.633 | 56.033 | 3.9 | **51.189** | 93.8 | 59.333 | 3.8 | **52.311** |
| SCRUB | 55.433 | 18.6 | 32.6 | 35.544 | 64.733 | 23.1 | 71.633 | 53.155 | 54.767 | 16.133 | 9.833 | 26.911 | 39.3 | 9.833 | 56.3 | 35.144 |
| +ASAM 1.0 | 46.467 | 14.867 | 0.1 | **20.478** | 57.367 | 22.633 | 0.167 | **26.722** | 44.7 | 14.567 | 0.2 | **19.822** | 14.433 | 2.333 | 0.2 | **5.655** |
| RL | 90.767 | 62.933 | 10.767 | 54.822 | 91.6 | 68.267 | 13.5 | 57.789 | 89.067 | 63.567 | 15.8 | 56.145 | 89.167 | 61.967 | 8.267 | 53.134 |
| +ASAM 1.0 | 90.3 | 61.3 | 9.467 | **53.689** | 91.6 | 62.667 | 12.7 | **55.656** | 88.0 | 61.3 | 10.667 | **53.322** | 86.3 | 59.833 | 5.833 | **50.655** |
| SalUn | 83.433 | 59.233 | 7.333 | 50.0 | 84.533 | 59.1 | 11.167 | 51.6 | 79.3 | 54.667 | 8.8 | 47.589 | 81.467 | 53.133 | 6.867 | 47.156 |
| +ASAM 1.0 | 79.1 | 51.833 | 4.5 | **45.144** | 81.7 | 54.167 | 6.633 | **47.50** | 74.967 | 49.5 | 4.2 | **42.889** | 75.633 | 47.667 | 4.067 | **42.456** |
| NegGrad | 86.933 | 37.233 | 2.167 | 42.111 | 88.867 | 40.2 | 1.733 | 43.60 | 82.167 | 32.1 | 1.8 | 38.689 | 74.667 | 36.967 | 3.433 | 38.356 |
| +ASAM 1.0 | 84.5 | 30.1 | 0.733 | **38.444** | 85.6 | 30.1 | 0.7 | **38.8** | 81.233 | 24.533 | 0.533 | **35.433** | 73.967 | 20.733 | 0.366 | **31.689** |

Thus, we encourage high retain/test accuracies and low forget accuracy. Note that our ToW differs from that in previous work as we measure the raw accuracy difference instead of the absolute difference, because new unlearning methods that continue to fine-tune on $\mathcal{R}$ can outperform $f_{\mathcal{A}(\mathcal{R})}$ within a conventional unlearning time $T_2$. If using the absolute ToW, a higher test accuracy than $f_{\mathcal{A}(\mathcal{R})}$ will be penalized and the model performance cannot be properly measured.

## 4.1 SAM CONSISTENTLY OUTPERFORMS WITH BETTER TRADEOFF

We conduct unlearning with various unlearning algorithms $\mathcal{U}$ given different pretrained $f_{\mathcal{A}}$. Tab. 1 reports ToW scores of $\mathcal{U}$ on CIFAR-100 and ImageNet. We observe that SAM consistently improves all unlearning methods under different initializations $f_{\mathcal{A}}^{T_1}$, suggesting that **SAM can universally enhance prevailing $\mathcal{U}$**. While different $\mathcal{U}$ exhibit varied effectiveness to $[\mathcal{F}_{\text{high}}, \mathcal{F}_{\text{mid}}, \mathcal{F}_{\text{low}}]$, we observe that NegGrad achieves a better balance between three forget sets than other methods. We include detailed [retain, forget, test] accuracies, further analysis and demonstration of statistical significance in App. F. Upon close examination on those accuraices, we observe that despite SAM outperforms SGD by better retain and test accuracies and thus better ToW, SGD can oftentimes achieve lower forget accuracies. This aligns with our theoretical analysis where SGD overfits more to $\mathcal{F}$, and it also sparks our Sharp MinMax. Smaller experiments on CIFAR-10 and Tiny-ImageNet in App. G yield aligned conclusions.

**MIA correctness.** We report correctness rates of membership inference attack (MIA) to $\mathcal{F}$ on CIFAR-100 in Tab. 2. Lower correctness means better unlearning: forget samples behave more like samples that were never in $\mathcal{S}$. We find that **SAM consistently improves data privacy while unlearning more effectively**. Note that NegGrad achieves better MIA correctness than RL; this is because gradient ascent actively erases gradient signatures of $\mathcal{F}$ in the model. SCRUB (Kurmanji et al., 2023) with SAM achieves best MIA performance.

**Relearning attacks.** We also present relearning attack experiments to demonstrate SAM's unlearning robustness in App. G.5. We observe that SAM enhanced $\mathcal{U}$ are more resilient to relearning attacks with smaller increases. While not our main focus, these experiments highlight the robustness of our approach and encourage future works for deeper investigation into the role of loss landscape geometry for robust unlearning.

**KL on margins.** While ToW measures performance closeness between unlearned model and retrained model, Georgiev et al. (2024) propose to measure distribution closeness in output space with KL divergence on margins (KLoM). We evaluate NegGrad w/ SGD vs. w/ SAM on CIFAR-100 using KLoM means and 95%-percentiles (tails) in App. G.7, and observe similar conclusions: on KLoM means, SGD can outperform SAM on $\mathcal{F}$, but SAM achieves better closeness on $\mathcal{R}, \mathcal{D}_{\text{test}}$ and hence better KLoMs overall; on 95%-percentiles, SAM outperforms SGD for all settings, suggesting lower variance and better stability at tails. While SAM is not targeted to resolve data dependency issues in unlearning (Georgiev et al., 2024), it ameliorates them by its geometric properties as suggested by lower entanglement and better ToW and KLoM.

**Our observations further generalize.** We consider structured noise unlearning, where another source of noise is introduced during unlearning. We adopt the glass blur and snow effect from ImageNet-C (Hendrycks & Dietterich, 2019) to corrupt $\mathcal{R}$ and $\mathcal{F}$ of CIFAR-100, and unlearn with NegGrad and Sharp MinMax. We record experiment results in App. G.3, and observe consistent conclusions where SAM outperforms under both corruptions. We also experiment on ViT-Small (Dosovitskiy et al., 2020) with AdamW (Loshchilov & Hutter, 2017) on CIFAR-100 in App. G.4, with NegGrad and Sharp MinMax. We continue to observe promising improvement by adding SAM, with significant increase of ToW on Sharp MinMax. While we focus on studying the geometric properties of SAM rather than efficiency, we are the first to demonstrate how recent efficient SAM variants (specifically Momentum SAM (Becker et al., 2024)) can perform equivalently well as the original SAM with much less computation overhead in App. G.6.

## 4.2 Constrained Overfitting Benefits Unlearning

Table 3: ToW(%) ↑ of Sharp MinMax on ImageNet-1K and CIFAR-100. Comparing with Tab. 1, Sharp MinMax achieves new best ToW performance.

| ImageNet | $\mathcal{A}$ =SGD | | | | $\mathcal{A}$ =ASAM 0.1 | | | | $\mathcal{A}$ =ASAM 1.0 | | | | $\mathcal{A}$ =SAM 0.1 | | | |
|---|---|---|---|---|---|---|---|---|---|---|---|---|---|---|---|---|
| Unlearn $\mathcal{U}$ | High | Mid | Low | AVG | High | Mid | Low | AVG | High | Mid | Low | AVG | High | Mid | Low | AVG |
| SGD | 73.357 | 80.881 | 86.334 | 80.191 | 73.418 | 80.784 | 84.378 | 79.527 | 73.103 | 81.105 | 86.402 | 80.204 | 73.052 | 80.913 | 85.517 | 79.827 |
| ASAM 0.1 | 78.066 | 87.914 | 87.338 | 84.44 | 79.077 | 87.4 | 86.953 | 84.476 | 70.148 | 88.039 | 87.554 | 81.914 | 78.529 | 87.642 | 86.668 | 84.28 |
| ASAM 1.0 | 86.658 | 87.345 | 89.694 | **87.899** | 86.166 | 87.192 | 89.138 | **87.498** | 86.915 | 87.27 | 90.142 | 88.109 | 86.272 | 87.076 | 90.064 | **87.804** |
| SAM 0.1 | 86.463 | 86.755 | 90.005 | 87.741 | 85.511 | 86.635 | 89.852 | 87.333 | 86.849 | 86.722 | 91.111 | **88.227** | 85.712 | 86.486 | 90.207 | 87.468 |

| CIFAR100 | $\mathcal{A}$ =SGD | | | | $\mathcal{A}$ =ASAM 0.1 | | | | $\mathcal{A}$ =ASAM 1.0 | | | | $\mathcal{A}$ =SAM 0.1 | | | |
|---|---|---|---|---|---|---|---|---|---|---|---|---|---|---|---|---|
| Unlearn $\mathcal{U}$ | High | Mid | Low | AVG | High | Mid | Low | AVG | High | Mid | Low | AVG | High | Mid | Low | AVG |
| SGD | 70.7668 | 76.692 | 82.853 | 76.771 | 72.137 | 77.864 | 81.847 | 77.282 | 65.925 | 74.526 | 80.127 | 73.526 | 60.478 | 71.931 | 73.843 | 68.751 |
| ASAM 0.1 | 78.895 | 96.027 | 83.473 | 86.132 | 84.968 | 96.451 | 82.883 | 88.101 | 81.825 | 93.786 | 87.151 | 87.587 | 72.897 | 80.104 | 86.659 | 79.887 |
| ASAM 1.0 | 82.27 | 94.913 | 86.504 | 87.896 | 77.576 | 99.422 | 85.894 | 87.631 | 84.521 | 87.761 | 84.381 | 85.554 | 76.037 | 83.633 | 77.461 | 79.044 |
| SAM 0.1 | 90.578 | 90.960 | 92.494 | **91.344** | 91.695 | 95.543 | 91.508 | **92.915** | 88.664 | 88.646 | 93.163 | **90.158** | 85.195 | 78.286 | 90.963 | **84.814** |

We present ToW of Sharp MinMax and compare to Tab. 1. Compared with NegGrad and other methods, Sharp MinMax further improves the unlearning capabilities across all settings by a noticeable margin, especially on $\mathcal{F}_{\text{high}}$, and SAM 0.1 achieves ToW $> 0.9$ for most settings on CIFAR-100. The effectiveness of Sharp MinMax assures our assumptions about overfitting for sample-specific unlearning, providing new insights for designing future unlearning algorithms. By constraining overfitting to only a small portion of model parameters which are most salient to $\mathcal{F}$, Sharp MinMax effectively boosts unlearning performance. In App. G.5, the impact of relearning attacks to Sharp MinMax is effectively limited to the sharp terrains as it makes retain and forget models geometrically distinct, so the robustness against relearning attacks as well as the model performance is retained.

## 4.3 Quantitative Analysis and Visualizations

**Measuring entanglement.** We measure the entanglement between $\mathcal{R}$ and $\mathcal{F}$ before and after unlearning. At a coarse level, we implement variance-based entanglement from Goldblum et al. (2020); Zhao et al. (2024): $E_{\text{Var}}^{\text{All}}(\mathcal{R}, \mathcal{F}, f) = (\frac{1}{|\mathcal{R}|} \sum_{i \in \mathcal{R}} (\phi_i - \mu_{\mathcal{R}})^2 + \frac{1}{|\mathcal{F}|} \sum_{j \in \mathcal{F}} (\phi_j - \mu_{\mathcal{F}})^2) / ((\mu_{\mathcal{R}} - \mu)^2 + (\mu_{\mathcal{F}} - \mu)^2)$, where $\phi_i, \phi_j$ denote sample embedding, $\mu_{\mathcal{R}}, \mu_{\mathcal{F}}$ denote mean embedding of $\mathcal{R}, \mathcal{F}$, and $\mu$ denotes mean embedding over $\mathcal{R} \cup \mathcal{F}$. We also compute the class-wise entanglement and report weighted averaged $E_{\text{Var}}^{\text{Cls}}$. However, $E_{\text{Var}}$ assumes good/convex shapes of clusters and relies heavily on cluster means. Inspired by Optimal Transport literature, we propose a refined geometry-

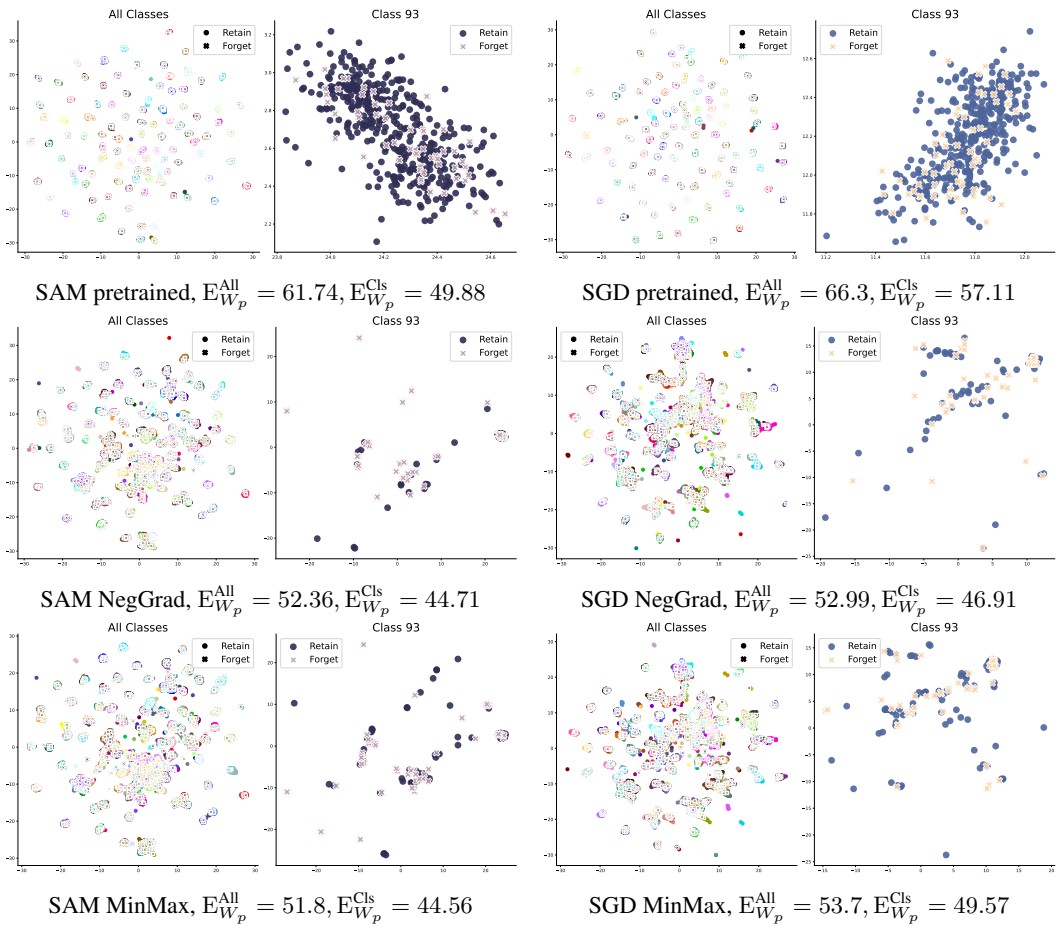

Figure 1: UMAP (McInnes et al., 2018) feature analysis on Mid Mem $\mathcal{F}_{\text{mid}}$. At all-class level, we observe that SAM better maintains class clusters after unlearning while SGD is forming a more evident clump of features; at classwise level, we observe that while both push away forget features, SGD also scatters retain features further, suggesting overfitting. This also explains the larger clump of SGD at all-class level. We observe that SAM further pushes away forget features on $\mathcal{F}_{\text{high}}$ and SGD scatters more retain features on $\mathcal{F}_{\text{low}}$, see App. H.2 for full visualizations.

Table 4: Entanglement ↓ between $\mathcal{F}$ and $\mathcal{R}$ of different memorization levels given models based on SGD and ASAM 1.0. While $\text{E}_{\text{Var}}$ is hard to conclude a comparison between SGD and SAM across different $\mathcal{U}$, SAM shows less entanglement both before and after unlearning than SGD by $\text{E}_{W_p}$.

| SGD | Variance $\text{E}_{\text{Var}}$ | | | | Wasserstein $\text{E}_{W_p}$ | | | | SAM | Variance $\text{E}_{\text{Var}}$ | | | | Wasserstein $\text{E}_{W_p}$ | | | |
|---|---|---|---|---|---|---|---|---|---|---|---|---|---|---|---|---|---|
| Model | High | Mid | Low | AVG | High | Mid | Low | AVG | Model | High | Mid | Low | AVG | High | Mid | Low | AVG |
| Pretrained | 30.5 | 95.28 | 32.39 | 52.72 | 59.58 | 66.3 | 63.13 | 63.0 | Pretrained | 29.56 | 88.43 | 28.91 | **48.97** | 55.86 | 61.74 | 59.84 | **59.15** |
| -per class | 2.5 | 6.71 | 2.51 | **3.91** | 51.21 | 57.11 | 59.64 | 55.99 | -per class | 2.88 | 6.66 | 2.71 | 4.08 | 45.45 | 49.88 | 52.46 | **49.26** |
| NegGrad | 18.87 | 37.16 | 22.12 | **26.05** | 51.24 | 52.99 | 56.12 | 53.45 | NegGrad | 17.78 | 37.49 | 24.47 | 26.58 | 49.87 | 52.36 | 54.93 | **52.39** |
| -per class | 0.56 | 1.8 | 2.69 | **1.68** | 35.22 | 46.91 | 55.93 | 46.02 | -per class | 0.66 | 2.03 | 2.88 | 1.86 | 36.42 | 44.71 | 50.83 | **43.99** |
| MinMax | 17.7 | 38.03 | 21.51 | 25.75 | 51.12 | 53.7 | 56.77 | 53.86 | MinMax | 16.35 | 32.07 | 20.75 | **23.06** | 51.26 | 51.8 | 55.08 | **52.71** |
| -per class | 0.69 | 2.41 | 2.27 | 1.79 | 38.41 | 49.57 | 57.15 | 48.38 | -per class | 0.49 | 1.52 | 2.97 | **1.66** | 33.65 | 44.56 | 52.55 | **43.59** |

aware entanglement based on Wasserstein distance to measure the separation of retain and forget features, $\text{E}_{W_p}^{\text{All}}$ and $\text{E}_{W_p}^{\text{Cls}}$, which computes the cost of transferring one shaped distribution to another point-wisely. From Tab. 4, we observe that both SGD and SAM unlearning have decreased entanglement with $\text{E}^{\text{Cls}} < \text{E}^{\text{All}}$. While $\text{E}_{\text{Var}}$ cannot further differentiate, we observe that SAM achieves better $\text{E}_{W_p}$ than SGD at all levels. Fig. 1 visualizes the feature space of $\mathcal{A}, \mathcal{U} = \text{ASAM } 1.0$ and $\mathcal{A}, \mathcal{U} = \text{SGD}$ on $\mathcal{F}_{\text{mid}}$. For all classes, we observe forget samples are assigned to wrong class clusters after unlearning, where SAM better maintains class clusters. For class-wise, we visualize the largest class in $\mathcal{F}_{\text{mid}}$ and observe that SGD unlearning scatters more retain samples than its SAM counterpart, suggesting overfitting. See App. H.2 for complete visualizations.

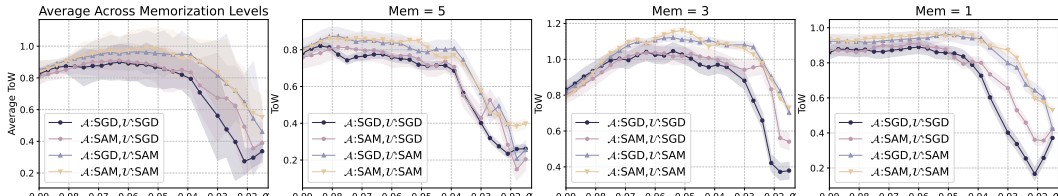

Figure 2: As $\alpha$ decreases, NegGrad puts less weight on retain signals and learns more from $\mathcal{F}$, leading to harmful overfitting. SAM exhibits more tolerance to insufficient retain signals, while $\mathcal{A}, \mathcal{U} = $ SGD collapses the fastest. Note that ToW starts failing before $\alpha = |\mathcal{R}|/(|\mathcal{F}| + |\mathcal{R}|)$, implying more factors affecting $\alpha$ threshold as we point out.

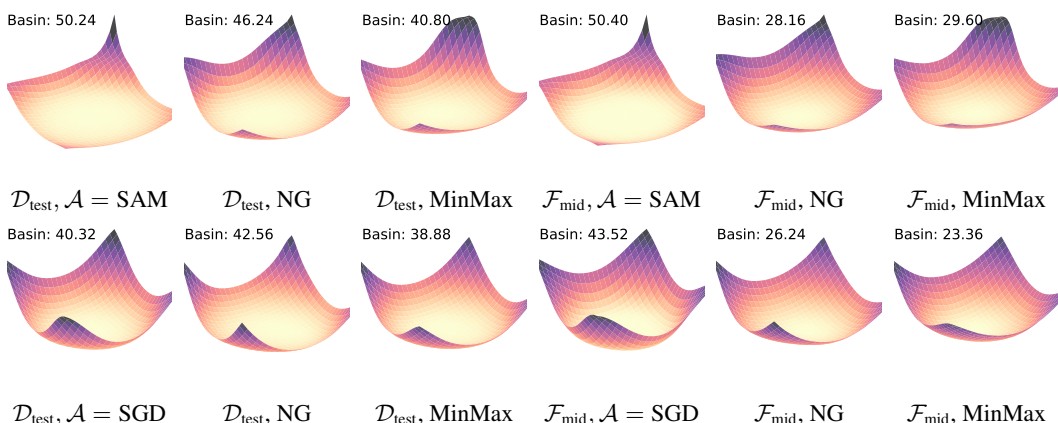

Figure 3: Loss landscapes on $\mathcal{D}_{\text{test}}$ and $\mathcal{F}_{\text{mid}}$, where first row shows a SAM pretrained model and SAM unlearned models, and second row shows SGD counterparts. While unlearning increases sharpness as suggested by reduced basin ratios, we observe SAM unlearned models still maintain flatter landscapes than SGD models do.

**Reducing retain signal.** We verify Lemma 3.4 by reducing $\alpha$ in NegGrad. Fig. 2 shows ToW changes as $\alpha$ decreases for various $\mathcal{A}, \mathcal{U}$ pairs at different memorization levels on CIFAR-100. We observe that $\mathcal{A}, \mathcal{U} = $ SGD fails the fastest and hardest, while $\mathcal{A}, \mathcal{U} = $ ASAM 1.0 exhibits the best resilience. Also note that for CIFAR-100, $|\mathcal{R}|/(|\mathcal{F}| + |\mathcal{R}|) \approx 0.93$, but unlearning starts to fail at a higher $\alpha$. This supports our claim that $\alpha$ depends more than retain-forget ratio.

**Loss landscape.** We visualize loss landscapes of SGD and ASAM 1.0 by perturbing original model along two directions with filter normalization (Li et al., 2018), and quantify more sharpness by smaller basin ratio. Fig. 3 shows loss landscapes on $\mathcal{D}_{\text{test}}$ and $\mathcal{F}_{\text{mid}}$, where SAM unlearning generally keeps flatter landscapes. Same observations apply to different $\mathcal{F}$ except that we observe SGD+NegGrad on $\mathcal{F}_{\text{high}}$ to achieve flatter landscape, which might indicate that unlearning can be an implicit regularizer, we will leave it to future work. See full visualizations and more details in App. H.1.

## 5 CONCLUSION

In this paper, we provide a refined characterization of SAM under NegGrad unlearning, and theoretical insights on bounding and choosing the weight factor to balance retain and forget signals. Extensive studies verify our analysis and reveals more underlying properties of SAM that are desired for unlearning. Based on our rethinking of overfitting, we also propose a new algorithm which further pushes the boundary of sample-specific unlearning. Our theoretical and empirical findings shed light on future design of unlearning algorithms.

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

# A    RELATED WORKS

## A.1    MACHINE UNLEARNING

A wide variety of unlearning algorithms have been proposed to erase the influence of specific data in the pre-trained model. Basic approaches involve finetuning on retain set to unlearn the forget samples with catastrophic forgetting, randomly labeling forget set to force the model to ignore the noisy forget samples, and explicitly "learning to unlearn" from the forget set via gradient ascent (Golatkar et al., 2020; Graves et al., 2021; Warnecke et al., 2021). Recent work pushes the boundaries of each genre with more advanced tools. L1-Sparse (Jia et al., 2023) finetunes on retain set with L1 penalty to improve unlearning with sparsification, NegGrad and SCRUB (Kurmanji et al., 2023) combines gradient descent on retain set and gradient ascent on forget set to jointly update the model, Influence Unlearning and Saliency Unlearning (Izzo et al., 2021; Fan et al., 2023) aim to find model parameters which are important to the forget set for more effective unlearning while preserving model performance. Theoretical work in unlearning draws insights from differential privacy and characterizes distributional closeness in $(\epsilon, \delta)$-language. Sekhari et al. (2021) studies unlearning with second-order update which computes Hessian inverse. Langevin Unlearning (Chien et al., 2024) studies approximate unlearning with privacy and efficiency guarantees based on projected noisy gradient descent. Unlearning also extends to generative vision and language tasks, addressing privacy and safety concerns, erasing concepts, and aligning with human preference (Ko et al., 2024; Wang et al., 2024; Zhang et al., 2024; Scholten et al., 2025).

## A.2    SHARPNESS AWARE MINIMIZATION

Sharpness-aware minimization (SAM) perturbs the model within a ball neighborhood to maximize the loss. Since perturbations in sharp regions result in higher penalties, SAM learns to avoid sharp landscapes and improve generalization with flatness. Recent work improves SAM's flexibility and efficiency. Adaptive SAM (Kwon et al., 2021) introduces scale-invariant adaptive sharpness to address parameter re-scaling sensitivity. GA-SAM (Zhang & Lan, 2022) adapts the perturbation based on gradient strength to improve generalization performance. Sparse SAM (Mi et al., 2022) shows that adding sparsity in perturbations can preserve or even improve performance while accelerating training. LookSAM (Liu et al., 2022) efficiently scales up SAM by only periodically computing the inner gradient ascent. Theoretical studies of SAM focus both on the convergence analysis (Khanh et al., 2024) and its dynamics (Bartlett et al., 2022). Chen et al. (2023) reveal the fundamental mechanism of SAM that prevents memorizing noisy signals by deactivating neurons based on a practical signal-to-noise analytical framework. This inspires us to investigate the intriguing properties of SAM in machine unlearning, where signals from the forget set can be naturally modeled as the noise from the perspective of maintaining model performance with remaining samples.

## A.3    DATA MEMORIZATION

Recent work aims to identify key factors that affect the difficulty of an unlearning task. Fan et al. (2024) define and seek the "worst-case" forget set using a gradient-based adversarial approach. Carlini et al. (2019) investigates and quantifies the atypical-ness of data samples under a differential privacy setting. Zhao et al. (2024) discovers that the more memorized the forget examples are, the harder unlearning becomes. We agree with the empirical studies in Zhao et al. (2024) and study the unlearning effectiveness under different levels of data memorization. Memorization literature provides fundamental understanding and interpretation of learning dynamics and model behaviors, characterizing generalization bounds and the interplay with data (Feldman & Zhang, 2020; Attias et al., 2024). Recent studies also investigate the effects of memorization in large-scale scenarios such as language models (Biderman et al., 2023; Prashanth et al., 2024; Li et al., 2025). Specifically, the memorization and influence scores in Feldman (2020); Feldman & Zhang (2020) provide insights into evaluating unlearning algorithms and designing new approaches. In our study, we have observed varied effectiveness of each unlearning method with respect to forget sets of different memorization levels, and aim at designing unlearning methods which perform well on forgets sets of all difficulties.

# B STATEMENTS

## B.1 REPRODUCIBILITY STATEMENT

**Experiment environment.** Our code is built upon several open-source code bases [1] and will be released. We perform all experiments on single NVIDIA A100/H100. We fix random seed for all data processing, saved precomputation (e.g., indices for data subsetting, weight masks), model splitting, pretraining and retraining for reproducible observations. For unlearning parameters and settings, we run experiments with multiple seeds to evaluate statistical significance, see App. F.1.

**Theoretical Assumptions.** Our theoretical analysis follows standard, existing assumptions of model size, data size, effective information in the data (signal) and Gaussian noise in data, which were previously stated in Kou et al. (2023); Chen et al. (2023). In addition to mentioned common assumptions, our Assumption D.1 also assumes conventional unlearning schemes: cross-entropy loss, ReLU activation, clean labels and reasonable size of forget set ($< 1/2$ trainset size).

## B.2 LLM USAGE STATEMENT

We use GPT to fix grammar and polish short phrases to sharpen our expression. We also use GPT as a smart search engine to gather recent work of interest and summarize existing bug fixes. Zero LLM usage for any core component of our work, including data processing, implementation and experiment, theory, etc., and LLM does not guide the development of any module. No "vibe coding" and mathematical derivation from LLM.

## B.3 LIMITATIONS AND FUTURE WORK

There are a few limitations based on the signal-to-noise framework, which on the other hand inspire us for future studies. First, there are more interference which can be modeled as noise in machine unlearning, such as the overlap between retain set and forget set. Using hard-cutoff or random sampling to build $\mathcal{F}$ might split two similar samples into two opposite subsets, causing interference and impacting unlearning effectiveness. We hypothesize that less overlap between $\mathcal{R}$ and $\mathcal{F}$ results in more effective unlearning, and vice versa. With more identified and modeled noise sources, another limitation comes from the uncharacterized behaviors when retain signals are weak for some upper bound. Will SAM fail into harmful overfitting under this circumstance? Theoretical and empirical studies under this situation might leverage the interplay between all signals, including different noisy signals. Another limitation comes from the design of ascent-based unlearning like NegGrad as discussed by a concurrent work Mavrothalassitis et al. (2025): while the objective encourages misclassification, the targeted retrained model should treat forget samples as never seen (generalizing or guessing). This misaligned objective might cause potential imprecision and leakage. While we show that SAM's benefits trivially apply to randomness-based unlearning (e.g. RL in Eqn. 3), and ascent-based unlearning is widely adopted in frontiers (LLM, diffusions), deeper studies are expected for more robust unlearning. Last, we observe an intriguing "regularizing" effect of unlearning using SGD via loss landscape visualization, which demands deeper investigation in future work.

---

[1] https://github.com/kairanzhao/RUM, https://github.com/davda54/sam, https://github.com/OPTML-Group/Unlearn-Saliency, https://pluskid.github.io/influence-memorization/

## C    TABLE OF NOTATIONS

Table 5:

| Symbol | Meaning / Notes | Symbol | Meaning / Notes |
|---|---|---|---|
| $\mathbf{x}_i \in \mathbb{R}^{P \times d}$ | Input image of sample $i$, vectorized into $P$ patches of dimension $d$ (one patch holds the signal $y_i \boldsymbol{\varphi}$ and $P-1$ patches contain noise) | $y_i \in \{\pm 1\}$ | Binary class label for sample $i$ |
| $\boldsymbol{\varphi} \in \mathbb{R}^d$ | Universal signal vector shared across samples | $\boldsymbol{\xi}_i \in \mathbb{R}^d$ | Noise vector for sample $i$, often drawn from $\mathcal{N}(\mathbf{0}, \sigma_p^2 \mathbf{I})$ |
| $P$ | Number of patches per input image | $d$ | Dimensionality of each patch and each convolutional filter |
| $m$ | Number of convolutional filters per class | $\mathbf{w}_{j,r} \in \mathbb{R}^d$ | Weight vector for the $r$-th filter of class $j \in \{\pm 1\}$ |
| $\mathbf{W}_j$ | Collection of filters $\{\mathbf{w}_{j,r}\}_{r=1}^m$ for class $j$ | $\mathbf{W}$ | Complete set of model parameters |
| $f(\mathbf{W}, \mathbf{x})$ | Two-class CNN output: $f_{+1}(\mathbf{W}_{+1}, \mathbf{x}) - f_{-1}(\mathbf{W}_{-1}, \mathbf{x})$ | $f_j(\mathbf{W}_j, \mathbf{x})$ | Class-$j$ output: $\frac{1}{m} \sum_{r=1}^m \sum_{p=1}^P \sigma(\langle \mathbf{w}_{j,r}, \mathbf{x}_p \rangle)$ |
| $\sigma(\cdot)$ | ReLU activation function | $\sigma'(\cdot)$ | Derivative of ReLU used in gradients |
| $\mathcal{L}(\mathbf{W}, \mathcal{S})$ | Cross-entropy loss over training set $\mathcal{S}$ | $\ell_i'^{(t,b)}$ | Gradient of the loss for sample $i$ at epoch $t$, batch $b$ |
| $\mathbf{w}_{j,r}^{(t,b)}$ | $r$-th filter of class $j \in \{\pm 1\}$ after $t$ epochs and $b$ batches | $\kappa_{j,r}^{(t,b)}$ | Learned signal coefficient for filter $(j, r)$ at step $(t, b)$ |
| $\zeta_{j,r,i}^{(t,b)}$ | Learned noise coefficient from sample $i$ on filter $(j, r)$ at step $(t, b)$ | $\|\boldsymbol{\varphi}\|_2, \|\boldsymbol{\xi}_i\|_2$ | Euclidean norms of the signal and noise vectors |
| $\mathcal{F} \subseteq \mathcal{S}$ | Forget set whose influence is to be removed | $\mathcal{R} = \mathcal{S} \setminus \mathcal{F}$ | Retain set used for continued training |
| $f_{\mathcal{A}}^{T_1}$ | Model after $T_1$ epochs of training by algorithm $\mathcal{A}$ | $f_{\mathcal{U}}^{T_2}$ | Model after $T_2$ epochs of unlearning by algorithm $\mathcal{U}$ |
| $T_1, T_2$ | Numbers of epochs for pretraining and unlearning | $\mathcal{I}_{t,b}^{\mathcal{R}}, \mathcal{I}_{t,b}^{\mathcal{F}}$ | Mini-batch indices from $\mathcal{R}$ and $\mathcal{F}$ at step $(t, b)$ |
| $B$ | Batch size | $\eta$ | Learning rate |
| $\mathrm{sgn}(\cdot)$ | Sign function returning $\pm 1$ | $\alpha$ | Weight in NegGrad balancing retain and forget contributions |
| $\widehat{\boldsymbol{\epsilon}}_{j,r}^{(t,b)}$ | SAM perturbation applied to $\mathbf{w}_{j,r}^{(t,b)}$ | $\tau, \rho$ | Perturbation radius in theory and in practice used in SAM/ASAM |
| $\delta$ | Perturbation term: $\delta = \widehat{\boldsymbol{\epsilon}}_{j,r}^{(t,b)}$ for SAM and 0 for SGD | $\nabla_{\boldsymbol{\varphi}_i}, \nabla_{\boldsymbol{\xi}_i}$ | Gradient contributions for the signal and noise in NegGrad updates |
| $\Delta_{\mathrm{epoch}}^{\mathrm{SAM}} \kappa_{j,r}$ | Per-epoch change of $\kappa_{j,r}$ under SAM | $\Delta_{\mathrm{epoch}}^{\mathrm{SGD}} \kappa_{j,r}$ | Per-epoch change of $\kappa_{j,r}$ under SGD |
| $\mathrm{Acc}(\theta, \mathcal{D})$ | Classification accuracy model on dataset; $\theta, \mathcal{D}$ are abbreviated terms in Acc() | $\mathrm{ToW}(f_{\mathcal{U}})$ | "Tug-of-war" metric combining retain, forget and test accuracies |
| $\mathcal{D}$ | data distribution | $\mathcal{F}_{\mathrm{high}}$ | Forget sets of high memorization difficulty; same for mid, low |
| $\mathrm{mem}(\mathcal{A}, \mathcal{S}, i)$ | Memorization score: $\Pr[f(\mathcal{S}) = y_i] - \Pr[f(\mathcal{S} \setminus i) = y_i]$ | $\mathcal{S} \setminus i$ | Training set $\mathcal{S}$ with sample $i$ removed |
| $\phi_i$ | Feature embedding of sample $i$ used in entanglement analysis | $\boldsymbol{\mu}_{\mathcal{R}}, \boldsymbol{\mu}_{\mathcal{F}}, \boldsymbol{\mu}$ | Mean embeddings of retain set, forget set and all data |

Table 5: (Continued)

| Symbol | Meaning / Notes | Symbol | Meaning / Notes |
|---|---|---|---|
| $\mathrm{E}_{\mathrm{Var}}^{\mathrm{All}}(\mathcal{R}, \mathcal{F}, f)$ | Variance-based entanglement measure between $\mathcal{R}$ and $\mathcal{F}$, given model $f$ | $\mathrm{E}_{\mathrm{Var}}^{\mathrm{Cls}}$ | Class-wise version of the variance-based entanglement |
| $\mathrm{E}_{W_p}^{\mathrm{All}}$, $\mathrm{E}_{W_p}^{\mathrm{Cls}}$ | Geometry-aware entanglement measures based on Wasserstein distance (all/class-wise) | $\mathcal{A}, \mathcal{U}$ | Training algorithm (e.g. SGD, SAM) and unlearning algorithm (e.g. NegGrad, RL, with default SGD optimization, can be used with SAM) |
| $\kappa_{j,r}^{(0,0)}$ | Initial signal coefficient for filter $(j, r)$ | $\zeta_{j,r,i}^{(0,0)}$ | Initial noise coefficient for sample $i$ on filter $(j, r)$ |
| $\|\mathcal{F}\|, \|\mathcal{R}\|$ | Cardinalities of the forget and retain sets, which is size in our work | $n$ | Total number of samples ($|\mathcal{S}|$) |
| $\mathcal{D}_{\mathrm{test}}$ | Test dataset used for evaluation | $\alpha^{\mathrm{SGD}}, \alpha^{\mathrm{SAM}}$ | $\alpha$ weight coeff for SGD and SAM, respectively |

## D  DETAILED FORMULATIONS AND PROOFS

We prove our theorems and lemmas based on previous theoretical results in Kou et al. (2023); Chen et al. (2023). Specifically, we prove that with additional yet necessary conditions for effective unlearning, the final test errors can be preserved, while we identify and characterize the changed internal dynamics. We begin by expanding and restating $\kappa, \zeta$ update rule for NegGrad in Eq. 7:

$$\kappa_{j,r}^{(t,b+1)} - \kappa_{j,r}^{(t,b)} = -\frac{\eta \|\boldsymbol{\varphi}\|_2^2}{Bm} \left[ \alpha \sum_{i \in \mathcal{I}_{t,b}^{\mathcal{R}}} \ell_i'^{(t,b)} \sigma'(\langle \mathbf{w}_{j,r}^{(t,b)} + \Delta, y_i \boldsymbol{\varphi} \rangle) \right.$$

$$\left. -(1-\alpha) \sum_{i \in \mathcal{I}_{t,b}^{\mathcal{F}}} \ell_i'^{(t,b)} \sigma'(\langle \mathbf{w}_{j,r}^{(t,b)} + \Delta, y_i \boldsymbol{\varphi} \rangle) \right],$$

$$\overline{\zeta}_{j,r}^{(t,b+1)} - \overline{\zeta}_{j,r}^{(t,b)} = -\frac{\eta (P-1)^2}{Bm} \left[ \alpha \sum_{i \in \mathcal{I}_{t,b}^{\mathcal{R}}} \|\boldsymbol{\xi}_i\|_2^2 \ell_i'^{(t,b)} \sigma'(\langle \mathbf{w}_{j,r}^{(t,b)} + \Delta, \boldsymbol{\xi}_i \rangle) \cdot \mathbb{1}(y_i = j) \right.$$

$$\left. -(1-\alpha) \sum_{i \in \mathcal{I}_{t,b}^{\mathcal{F}}} \|\boldsymbol{\xi}_i\|_2^2 \ell_i'^{(t,b)} \sigma'(\langle \mathbf{w}_{j,r}^{(t,b)} + \Delta, \boldsymbol{\xi}_i \rangle) \cdot \mathbb{1}(y_i = j) \right],$$

$$\underline{\zeta}_{j,r}^{(t,b+1)} - \underline{\zeta}_{j,r}^{(t,b)} = +\frac{\eta (P-1)^2}{Bm} \left[ \alpha \sum_{i \in \mathcal{I}_{t,b}^{\mathcal{R}}} \|\boldsymbol{\xi}_i\|_2^2 \ell_i'^{(t,b)} \sigma'(\langle \mathbf{w}_{j,r}^{(t,b)} + \Delta, \boldsymbol{\xi}_i \rangle) \cdot \mathbb{1}(y_i \neq j) \right.$$

$$\left. -(1-\alpha) \sum_{i \in \mathcal{I}_{t,b}^{\mathcal{F}}} \|\boldsymbol{\xi}_i\|_2^2 \ell_i'^{(t,b)} \sigma'(\langle \mathbf{w}_{j,r}^{(t,b)} + \Delta, \boldsymbol{\xi}_i \rangle) \cdot \mathbb{1}(y_i \neq j) \right],$$

$$(11)$$

where $\Delta = \widehat{\boldsymbol{\epsilon}}_{j,r}^{(t,b)}$ for SAM and 0 for SGD, $\zeta_{j,r}^{(t,b)}$ is split into $\overline{\zeta}_{j,r}^{(t,b)} := \zeta_{j,r}^{(t,b)} \mathbb{1}(\zeta_{j,r}^{(t,b)} \geq 0)$ and $\underline{\zeta}_{j,r}^{(t,b)} := \zeta_{j,r}^{(t,b)} \mathbb{1}(\zeta_{j,r}^{(t,b)} \leq 0)$ based on label agreement. We summarize several reasonable assumptions from previous work in addition to our conditions which ensure unlearning to progress:

**Assumption D.1** *Suppose there exists a sufficiently large constant C, such that the following hold:*

1. *Sufficiently large dimension d:* $d \geq C \max\{n\sigma_p^{-2} \|\boldsymbol{\varphi}\|_2^2 \log(T^*), n^2 \log(nm/\delta)(\log(T^*))^2\}$, *for some* $T^* = \Omega(\eta^{-1} Bmd^{-1} P^{-2} \sigma_p^{-2})$.

2. *The size of $\mathcal{S}$ and the CNN width satisfy $n \geq C \log(m/\delta), m \geq C \log(n/\delta)$.*

3. *The signal strength satisfies $\|\boldsymbol{\varphi}\|_2^2 \geq C\sigma_p^2 \log(n/\delta)$.*

4. *For the Gaussian noise initialization, $\sigma_0 \leq (C \max\{\sigma_p d/\sqrt{n}, \sqrt{\log(m/\delta)} \cdot \|\boldsymbol{\varphi}\|_2\})^{-1}$.*

5. *The learning rate $\eta$ satisfies $\eta \leq (C \max\{\sigma_p^2 d^{3/2}/(n^2 m\sqrt{\log(n/\delta)}), \sigma_p^2 d/n\})^{-1}$.*

6. *Assume cross-entropy loss: $\ell(z) = \log(1 + \exp(-z)) \Longrightarrow \ell' = -1/(1 + \exp(z))$.*

7. *Assume ReLU activation.*

8. *Assume all clean labels and $\mathcal{F}$ signals do not dominate: $\alpha \geq |\mathcal{R}|/(|\mathcal{F}| + |\mathcal{R}|) := \beta > 0.5$.*

We then obtain several proven quantities from previous work, which are achieved during pretraining and can be leveraged at the start of unlearning:

- $\sum_{i=1}^{n} \overline{\zeta}_{j,r,i}^{(t)}/\kappa_{j',r'}^{(t)} = \Theta(\mathrm{SNR}^{-2})$, for the signal-to-noise ratio $\mathrm{SNR} = \frac{\|\boldsymbol{\varphi}\|_2}{(P-1)\sigma_p\sqrt{d}}$.

- $\sum_{i=1}^{n} \overline{\zeta}_{j,r,i}^{(t)} = \Omega(n) = O(n \log(T^*)) = \widetilde{\Theta}(n)$, for some $T^* = \Omega(\eta^{-1}Bmd^{-1}P^{-2}\sigma_p^{-2})$.

- $\max_{j,r,i} |\underline{\zeta}_{j,r,i}^{(t)}| = \max\{O(\sqrt{\log(mn/\delta)} \cdot \sigma_0\sigma_p\sqrt{d}), O(\sqrt{\log(n/\delta)} \log(T^*) \cdot n/\sqrt{d})\}$.

- $\kappa_{j,r}^{(T^*)} = \Theta(\widehat{\kappa})$, where $\widehat{\kappa} = n \cdot \mathrm{SNR}^2$.

### D.1 PROOF TO THEOREM 3.2

Under NegGrad, we want to predict retain samples in $\mathcal{R}$ correctly while we count correct predictions in $\mathcal{F}$ as errors, yielding same bounds for $\mathbb{P}_{(\mathbf{x},y)\sim\mathcal{R}}(yf(\mathbf{W}^{(t)}, \mathbf{x}) \leq 0)$ and $\mathbb{P}_{(\mathbf{x},y)\sim\mathcal{F}}(yf(\mathbf{W}^{(t)}, \mathbf{x}) > 0)$ based on inverse objectives. However, when considering the test error on the model that is jointly updated by gradient descent on $\mathcal{R}$ and gradient ascent on $\mathcal{F}$, we still measure the error rate by wrong predictions. In other words, fitting forget samples will reduce the generalization performance. We can decompose the test error as follows:

$$
\begin{aligned}
&\mathbb{P}_{(\mathbf{x},y)\sim\mathcal{D}} \left( y \neq \mathrm{sign}\left( f\left(\mathbf{W}^{(t)}, \mathbf{x}\right)\right)\right) = \mathbb{P}_{(\mathbf{x},y)\sim\mathcal{D}} \left( yf\left(\mathbf{W}^{(t)}, \mathbf{x}\right) \leq 0\right) \\
=&\mathbb{P}_{(\mathbf{x},y)\sim\mathcal{D}} \left( yf\left(\mathbf{W}^{(t)}, \mathbf{x}\right) \leq 0, (\mathbf{x}, y) \in \mathcal{R}\right) + \mathbb{P}_{(\mathbf{x},y)\sim\mathcal{D}} \left( yf\left(\mathbf{W}^{(t)}, \mathbf{x}\right) \leq 0, (\mathbf{x}, y) \in \mathcal{F}\right) \\
=&\beta \cdot \mathbb{P}_{(\mathbf{x},y)\sim\mathcal{R}} \left( yf\left(\mathbf{W}^{(t)}, \mathbf{x}\right) \leq 0\right) + (1 - \beta) \cdot \mathbb{P}_{(\mathbf{x},y)\sim\mathcal{F}} \left( yf\left(\mathbf{W}^{(t)}, \mathbf{x}\right) \leq 0\right) \\
=&\beta \cdot \mathbb{P}_{(\mathbf{x},y)\sim\mathcal{R}} \left( yf\left(\mathbf{W}^{(t)}, \mathbf{x}\right) \leq 0\right) + (1 - \beta) \cdot \left(1 - \mathbb{P}_{(\mathbf{x},y)\sim\mathcal{F}} \left( yf\left(\mathbf{W}^{(t)}, \mathbf{x}\right) > 0\right)\right).
\end{aligned}
\tag{12}
$$

Note that in practice, $\mathcal{R}$ and $\mathcal{F}$ come from training set $\mathcal{S}$. During inference and evaluation, we convert the data augmentations of $\mathcal{R}, \mathcal{F}$ to test transforms, thus measuring proxy-test errors on $\mathcal{R}$-like and $\mathcal{F}$-like samples. To bound the test error, first decompose $yf(\mathbf{W}^{(t)}, \mathbf{x})$ into signal and noise learning of both positive and negative classes, considering $\Delta = 0$ for SGD:

$$
\begin{aligned}
yf\left(\mathbf{W}^{(t)}, \mathbf{x}\right) &= \frac{1}{m} \sum_{j,r} yj \left[\sigma\left(\left\langle \mathbf{w}_{j,r}^{(t)}, y\boldsymbol{\varphi}\right\rangle\right) + \sigma\left(\left\langle \mathbf{w}_{j,r}^{(t)}, \boldsymbol{\xi}\right\rangle\right)\right] \\
&= \frac{1}{m} \sum_{r} \left[\sigma\left(\left\langle \mathbf{w}_{y,r}^{(t)}, y\boldsymbol{\varphi}\right\rangle\right) + (P-1)\sigma\left(\left\langle \mathbf{w}_{y,r}^{(t)}, \boldsymbol{\xi}\right\rangle\right)\right] \\
&\quad - \frac{1}{m} \sum_{r} \left[\sigma\left(\left\langle \mathbf{w}_{-y,r}^{(t)}, y\boldsymbol{\varphi}\right\rangle\right) + (P-1)\sigma\left(\left\langle \mathbf{w}_{-y,r}^{(t)}, \boldsymbol{\xi}\right\rangle\right)\right].
\end{aligned}
\tag{13}
$$

**Remark D.2** *The following proof process for bounding $\mathbb{P}_{(\mathbf{x},y)\sim\mathcal{R}}(yf(\mathbf{W}^{(t)}, \mathbf{x})$ comes from Kou et al. (2023). We include it here for readability, since we will leverage the results when combining $\mathcal{R}$ and $\mathcal{F}$ in the end, as well as make adaptations for proving Theorem 3.3. Our results benefit from previous work as we consider the unlearning process as an extension of the second stage in Chen et al. (2023).*

We begin by two lemmas that bound the signal, noise norm, and the related inner products:

**Lemma D.3** *(Lemma B.4 in Kou et al. (2023)). Suppose that $\delta > 0$ and $d = \Omega(\log(6n/\delta))$. Then with probability at least $1 - \delta$,*

$$\sigma_p^2 d/2 \leq \|\boldsymbol{\xi}_i\|_2^2 \leq 3\sigma_p^2 d/2,$$

$$|\langle \boldsymbol{\xi}_i, \boldsymbol{\xi}_{i'} \rangle| \leq 2\sigma_p^2 \cdot \sqrt{d \log(6n^2/\delta)},$$

$$|\langle \boldsymbol{\xi}_i, \boldsymbol{\varphi} \rangle| \leq \|\boldsymbol{\varphi}\|_2 \sigma_p \cdot \sqrt{2 \log(6n/\delta)},$$

*for all $i, i' \in [n]$.*

**Lemma D.4** *(Lemma B.5 in Kou et al. (2023)). Suppose that $d = \Omega(\log(mn/\delta)), m = \Omega(\log(1/\delta))$. Then with probability at least $1 - \delta$,*

$$\sigma_0^2 d/2 \leq \left\| \mathbf{w}_{j,r}^{(0,0)} \right\|_2^2 \leq 3\sigma_0^2 d/2,$$

$$\left| \left\langle \mathbf{w}_{j,r}^{(0,0)}, \boldsymbol{\varphi} \right\rangle \right| \leq \sqrt{2 \log(12m/\delta)} \cdot \sigma_0 \|\boldsymbol{\varphi}\|_2,$$

$$\left| \left\langle \mathbf{w}_{j,r}^{(0,0)}, \boldsymbol{\xi}_i \right\rangle \right| \leq 2\sqrt{\log(12mn/\delta)} \cdot \sigma_0 \sigma_p \sqrt{d},$$

*for all $r \in [m], j \in \{\pm 1\}$ and $i \in [n]$. Moreover,*

$$\sigma_0 \|\boldsymbol{\varphi}\|_2/2 \leq \max_{r \in [m]} j \cdot \left\langle \mathbf{w}_{j,r}^{(0,0)}, \boldsymbol{\varphi} \right\rangle \leq \sqrt{2 \log(12m/\delta)} \cdot \sigma_0 \|\boldsymbol{\varphi}\|_2,$$

$$\sigma_0 \sigma_p \sqrt{d}/4 \leq \max_{r \in [m]} j \cdot \left\langle \mathbf{w}_{j,r}^{(0,0)}, \boldsymbol{\xi}_i \right\rangle \leq 2\sqrt{\log(12mn/\delta)} \cdot \sigma_0 \sigma_p \sqrt{d},$$

*for all $j \in \{\pm 1\}$ and $i \in [n]$.*

Plug in the weight update decomposition in Eq. 2, we can first bound the inner product for $j = y$:

$$\left\langle \mathbf{w}_{y,r}^{(t)}, y\boldsymbol{\varphi} \right\rangle = \left\langle \mathbf{w}_{y,r}^{(0)}, y\boldsymbol{\varphi} \right\rangle + \kappa_{y,r}^{(t)}$$

$$+ \frac{1}{P-1} \sum_{i=1}^{n} \overline{\zeta}_{y,r,i}^{(t)} \|\boldsymbol{\xi}_i\|_2^{-2} \langle \boldsymbol{\xi}_i, y\boldsymbol{\varphi} \rangle + \frac{1}{P-1} \sum_{i=1}^{n} \underline{\zeta}_{y,r,i}^{(t)} \|\boldsymbol{\xi}_i\|_2^{-2} \langle \boldsymbol{\xi}_i, y\boldsymbol{\varphi} \rangle$$

$$\geq -\sqrt{2 \log(12m/\delta)} \cdot \sigma_0 \|\boldsymbol{\varphi}\|_2 + \kappa_{y,r}^{(t)}$$

$$- \frac{\sqrt{2 \log(6n/\delta)}}{P-1} \cdot \sigma_p \|\boldsymbol{\varphi}\|_2 \cdot \left( \sigma_p^2 d/2 \right)^{-1} \left[ \sum_{i=1}^{n} \overline{\zeta}_{y,r,i}^{(t)} + \sum_{i=1}^{n} |\underline{\zeta}_{y,r,i}^{(t)}| \right]$$

$$= -\Theta\left( \sqrt{\log(m/\delta)} \sigma_0 \|\boldsymbol{\varphi}\|_2 \right) + \kappa_{y,r}^{(t)} - \Theta\left( \sqrt{\log(n/\delta)} \left( P\sigma_p d \right)^{-1} \|\boldsymbol{\varphi}\|_2 \right) \cdot \Theta\left( \text{SNR}^{-2} \right) \cdot \kappa_{y,r}^{(t)}$$

$$= -\Theta\left( \sqrt{\log(m/\delta)} \left( \sigma_p d \right)^{-1} \sqrt{n} \|\boldsymbol{\varphi}\|_2 \right) + \left[ 1 - \Theta\left( \sqrt{\log(n/\delta)} \cdot P\sigma_p/\|\boldsymbol{\varphi}\|_2 \right) \right] \kappa_{y,r}^{(t)}$$

$$= \Theta\left( \kappa_{y,r}^{(t)} \right),$$

$$(14)$$

where the inequality is by Lemma D.3 and Lemma D.4; the second equality is obtained by plugging in the coefficient orders we summarized at the beginning of the section; the third equality is by $\sigma_0 \leq C^{-1}(\sigma_p d)^{-1}\sqrt{n}$ in Assumption D.1 and $\text{SNR} = \|\boldsymbol{\varphi}\|_2/((P-1)\sigma_p\sqrt{d})$. The fourth equality is by $\kappa_{j,r}^{(t)} = \Theta(\widehat{\kappa})$, where $\widehat{\kappa} = n \cdot \text{SNR}^2$. Also $\sqrt{\log(n/\delta)} \cdot \sigma_p/\|\boldsymbol{\varphi}\|_2 \leq 1/\sqrt{C}$ and $\sqrt{\log(m/\delta)}(\sigma_p d)^{-1}\sqrt{n}\|\boldsymbol{\varphi}\|_2/\widehat{\kappa} = \sqrt{\log(m/\delta)}\sigma_p/(\sqrt{n}\|\boldsymbol{\varphi}\|_2) \leq \sqrt{\log(m/\delta)/n} \cdot 1/(\sqrt{C \log(n/\delta)}) \leq 1/(C\sqrt{\log(n/\delta)})$ holds by $\|\boldsymbol{\varphi}\|_2^2 \geq C \cdot \sigma_p^2 \log(n/\delta)$ and $n \geq C \log(m/\delta)$ in Assumption D.1, so for sufficiently large constant $C$ the equality holds. Similarly, we can show that $\langle \mathbf{w}_{-y,r}^{(t)}, y\boldsymbol{\varphi} \rangle = -\Theta(\kappa_{y,r}^{(t)}) < 0$ for $j \neq y$.

Next denote $g(\boldsymbol{\xi})$ as $\sum_r \sigma(\langle \mathbf{w}_{-y,r}^{(t)}, \boldsymbol{\xi} \rangle)$. Since $\boldsymbol{\xi} \sim \mathcal{N}(\mathbf{0}, \sigma_p^2 \mathbf{I})$, we can leverage the Gaussian concentration bound for $x \geq 0$:

$$\mathbb{P}(g(\boldsymbol{\xi}) - \mathbb{E}g(\boldsymbol{\xi}) \geq x) \leq \exp\left( -\frac{cx^2}{\sigma_p^2 \|g\|_{\text{Lip}}^2} \right),$$

$$(15)$$

where $c$ is a constant. To calculate the Lipschitz norm, we have

$$
|g(\boldsymbol{\xi}) - g(\boldsymbol{\xi}')| = \left| \sum_{r=1}^{m} \sigma\left(\left\langle \mathbf{w}_{-y,r}^{(t)}, \boldsymbol{\xi} \right\rangle\right) - \sum_{r=1}^{m} \sigma\left(\left\langle \mathbf{w}_{-y,r}^{(t)}, \boldsymbol{\xi}' \right\rangle\right) \right|
$$

$$
\leq \sum_{r=1}^{m} \left| \sigma\left(\left\langle \mathbf{w}_{-y,r}^{(t)}, \boldsymbol{\xi} \right\rangle\right) - \sigma\left(\left\langle \mathbf{w}_{-y,r}^{(t)}, \boldsymbol{\xi}' \right\rangle\right) \right| \tag{16}
$$

$$
\leq \sum_{r=1}^{m} \left| \left\langle \mathbf{w}_{-y,r}^{(t)}, \boldsymbol{\xi} - \boldsymbol{\xi}' \right\rangle \right| \leq \sum_{r=1}^{m} \left\| \mathbf{w}_{-y,r}^{(t)} \right\|_2 \cdot \| \boldsymbol{\xi} - \boldsymbol{\xi}' \|_2 .
$$

The first inequality is by triangle inequality; the second inequality is by the property of ReLU; the last inequality is by Cauchy-Schwartz inequality. Therefore, we have $\|g\|_{\mathrm{Lip}} \leq \sum_{r=1}^{m} \|\mathbf{w}_{-y,r}^{(t)}\|_2$, and since $\langle \mathbf{w}_{-y,r}^{(t)}, \boldsymbol{\xi} \rangle \sim \mathcal{N}(0, \|\mathbf{w}_{-y,r}^{(t)}\|_2^2 \sigma_p^2)$, we can get

$$
\mathbb{E} g(\boldsymbol{\xi}) = \sum_{r=1}^{m} \mathbb{E} \sigma\left(\left\langle \mathbf{w}_{-y,r}^{(t)}, \boldsymbol{\xi} \right\rangle\right) = \sum_{r=1}^{m} \frac{\left\| \mathbf{w}_{-y,r}^{(t)} \right\|_2 \sigma_p}{\sqrt{2\pi}} = \frac{\sigma_p}{\sqrt{2\pi}} \sum_{r=1}^{m} \left\| \mathbf{w}_{-y,r}^{(t)} \right\|_2 . \tag{17}
$$

Then, we seek to upper bound the 2-norm of $\mathbf{w}_{j,r}^{(t)}$. First we have

$$
\left\| \sum_{i=1}^{n} \zeta_{j,r,i}^{(t)} \cdot \|\boldsymbol{\xi}_i\|_2^{-2} \cdot \boldsymbol{\xi}_i \right\|_2^2
$$

$$
= \underbrace{\sum_{i=1}^{n} \zeta_{j,r,i}^{(t)}{}^2 \cdot \|\boldsymbol{\xi}_i\|_2^{-2}}_{\text{diagonal}} + \underbrace{2 \sum_{1 \leq i_1 < i_2 \leq n} \zeta_{j,r,i_1}^{(t)} \zeta_{j,r,i_2}^{(t)} \cdot \|\boldsymbol{\xi}_{i_1}\|_2^{-2} \|\boldsymbol{\xi}_{i_2}\|_2^{-2} \cdot \langle \boldsymbol{\xi}_{i_1}, \boldsymbol{\xi}_{i_2} \rangle}_{\text{off-diagonal}}
$$

$$
\leq 4\sigma_p^{-2} d^{-1} \sum_{i=1}^{n} \zeta_{j,r,i}^{(t)}{}^2 + 2 \sum_{1 \leq i_1 < i_2 \leq n} \left| \zeta_{j,r,i_1}^{(t)} \zeta_{j,r,i_2}^{(t)} \right| \cdot \left(16\sigma_p^{-4} d^{-2}\right) \cdot \left(2\sigma_p^2 \sqrt{d \log\left(6n^2/\delta\right)}\right)
$$

$$
= 4\sigma_p^{-2} d^{-1} \sum_{i=1}^{n} \zeta_{j,r,i}^{(t)}{}^2 + 32\sigma_p^{-2} d^{-3/2} \sqrt{\log\left(6n^2/\delta\right)} \left[ \left( \sum_{i=1}^{n} \left| \zeta_{j,r,i}^{(t)} \right| \right)^2 - \sum_{i=1}^{n} \zeta_{j,r,i}^{(t)}{}^2 \right] \tag{18}
$$

$$
= \Theta\left(\sigma_p^{-2} d^{-1}\right) \sum_{i=1}^{n} \zeta_{j,r,i}^{(t)}{}^2 + \widetilde{\Theta}\left(\sigma_p^{-2} d^{-3/2}\right) \left( \sum_{i=1}^{n} \left| \zeta_{j,r,i}^{(t)} \right| \right)^2
$$

$$
\leq \left[ \Theta\left(\sigma_p^{-2} d^{-1} n^{-1}\right) + \widetilde{\Theta}\left(\sigma_p^{-2} d^{-3/2}\right) \right] \left( \sum_{i=1}^{n} \left| \overline{\zeta}_{j,r,i}^{(t)} \right| + \sum_{i=1}^{n} \left| \underline{\zeta}_{j,r,i}^{(t)} \right| \right)^2
$$

$$
\leq \Theta\left(\sigma_p^{-2} d^{-1} n^{-1}\right) \left( \sum_{i=1}^{n} \overline{\zeta}_{j,r,i}^{(t)} \right)^2 .
$$

The first inequality is by Lemma D.3; for the second inequality we used the definition of $\overline{\zeta}, \underline{\zeta}$; for the second to last equation we plugged in coefficient orders. We can thus upper bound the 2-norm of $\mathbf{w}_{j,r}^{(t)}$ as:

$$
\left\| \mathbf{w}_{j,r}^{(t)} \right\|_2 \leq \left\| \mathbf{w}_{j,r}^{(0)} \right\|_2 + \kappa_{j,r}^{(t)} \cdot \|\boldsymbol{\varphi}\|_2^{-1} + \frac{1}{P-1} \left\| \sum_{i=1}^{n} \zeta_{j,r,i}^{(t)} \cdot \|\boldsymbol{\xi}_i\|_2^{-2} \cdot \boldsymbol{\xi}_i \right\|_2
$$

$$
\leq \left\| \mathbf{w}_{j,r}^{(0)} \right\|_2 + \kappa_{j,r}^{(t)} \cdot \|\boldsymbol{\varphi}\|_2^{-1} + \Theta\left(P^{-1} \sigma_p^{-1} d^{-1/2} n^{-1/2}\right) \cdot \sum_{i=1}^{n} \overline{\zeta}_{j,r,i}^{(t)} \tag{19}
$$

$$
= \Theta\left(P^{-1} \sigma_p^{-1} d^{-1/2} n^{-1/2}\right) \cdot \sum_{i=1}^{n} \overline{\zeta}_{j,r,i}^{(t)},
$$

where the first inequality is due to the triangle inequality, and the equality is due to the following:

$$\frac{\kappa_{j,r}^{(t)} \cdot \|\boldsymbol{\varphi}\|_2^{-1}}{\Theta\left(P^{-1}\sigma_p^{-1}d^{-1/2}n^{-1/2}\right) \cdot \sum_{i=1}^n \overline{\zeta}_{j,r,i}^{(t)}} = \Theta\left(P^{-1}\sigma_p d^{1/2}n^{1/2}\|\boldsymbol{\varphi}\|_2^{-1}\mathrm{SNR}^2\right)$$
$$=\Theta\left(P^{-1}\sigma_p^{-1}d^{-1/2}n^{1/2}\|\boldsymbol{\varphi}\|_2\right) = O(1), \tag{20}$$

based on the coefficient order $\sum_{i=1}^n \overline{\zeta}_{j,r,i}^{(t)}/\kappa_{j,r}^{(t)} = \Theta(\mathrm{SNR}^{-2})$, the definition of SNR, and the condition for $d$ in Assumption D.1. Similarly,

$$\frac{\left\|\mathbf{w}_{j,r}^{(0)}\right\|_2}{\Theta\left(P^{-1}\sigma_p^{-1}d^{-1/2}n^{-1/2}\right) \cdot \sum_{i=1}^n \overline{\zeta}_{j,r,i}^{(t)}} = \frac{\Theta\left(\sigma_0\sqrt{d}\right)}{\Theta\left(P^{-1}\sigma_p^{-1}d^{-1/2}n^{-1/2}\right) \cdot \sum_{i=1}^n \overline{\zeta}_{j,r,i}^{(t)}}$$
$$=O\left(P\sigma_0\sigma_p dn^{-1/2}\right) = O(1), \tag{21}$$

based on Lemma D.4, the coefficient order $\sum_{i=1}^n \overline{\zeta}_{j,r,i}^{(t)} = \Omega(n)$, and the condition for $\sigma_0$ in Assumption D.1. Then we can give an analysis of the following key component:

$$\frac{\sum_r \sigma\left(\left\langle\mathbf{w}_{y,r}^{(t)}, y\boldsymbol{\varphi}\right\rangle\right)}{(P-1)\sigma_p \sum_{r=1}^m \left\|\mathbf{w}_{-y,r}^{(t)}\right\|_2} \geq \frac{\Theta\left(\sum_r \kappa_{y,r}^{(t)}\right)}{\Theta\left(d^{-1/2}n^{-1/2}\right) \cdot \sum_{r,i} \overline{\zeta}_{-y,r,i}^{(t)}}$$
$$=\Theta\left(d^{1/2}n^{1/2}\mathrm{SNR}^2\right) = \Theta\left(n^{1/2}\|\boldsymbol{\varphi}\|_2^2/(P^2\sigma_p^2 d^{1/2})\right). \tag{22}$$

Then for $\|\boldsymbol{\varphi}\|_2 \geq C_1^{1/4}n^{-1/4}P\sigma_p d^{1/4}$ for some large constant $C_1$, we have

$$\sum_r \sigma\left(\left\langle\mathbf{w}_{y,r}^{(t)}, y\boldsymbol{\varphi}\right\rangle\right) - \frac{(P-1)\sigma_p}{\sqrt{2\pi}}\sum_{r=1}^m \left\|\mathbf{w}_{-y,r}^{(t)}\right\|_2 > 0. \tag{23}$$

**Upper bound.** Now plug in previous results to obtain

$$\mathbb{P}_{(\mathbf{x},y)\sim\mathcal{R}}\left(yf\left(\mathbf{W}^{(t)},\mathbf{x}\right) \leq 0\right) \leq \mathbb{P}_{(\mathbf{x},y)\sim\mathcal{R}}\left((P-1)\sum_r \sigma\left(\left\langle\mathbf{w}_{-y,r}^{(t)}, \boldsymbol{\xi}\right\rangle\right) \geq \sum_r \sigma\left(\left\langle\mathbf{w}_{y,r}^{(t)}, y\boldsymbol{\varphi}\right\rangle\right)\right)$$

$$=\mathbb{P}_{(\mathbf{x},y)\sim\mathcal{R}}\left(g(\boldsymbol{\xi}) - \mathbb{E}g(\boldsymbol{\xi}) \geq 1/(P-1)\sum_r \sigma\left(\left\langle\mathbf{w}_{y,r}^{(t)}, y\boldsymbol{\varphi}\right\rangle\right) - \frac{\sigma_p}{\sqrt{2\pi}}\sum_{r=1}^m \left\|\mathbf{w}_{-y,r}^{(t)}\right\|_2\right)$$

$$\leq \exp\left[-\frac{c\left(1/(P-1)\sum_r \sigma\left(\left\langle\mathbf{w}_{y,r}^{(t)}, y\boldsymbol{\varphi}\right\rangle\right) - (\sigma_p/\sqrt{2\pi})\sum_{r=1}^m \left\|\mathbf{w}_{-y,r}^{(t)}\right\|_2\right)^2}{\sigma_p^2\left(\sum_{r=1}^m \left\|\mathbf{w}_{-y,r}^{(t)}\right\|_2\right)^2}\right]$$

$$=\exp\left[-c\left(\frac{\sum_r \sigma\left(\left\langle\mathbf{w}_{y,r}^{(t)}, y\boldsymbol{\varphi}\right\rangle\right)}{(P-1)\sigma_p \sum_{r=1}^m \left\|\mathbf{w}_{-y,r}^{(t)}\right\|_2} - 1/\sqrt{2\pi}\right)^2\right]$$

$$\leq \exp(c/2\pi)\exp\left(-0.5c\left(\frac{\sum_r \sigma\left(\left\langle\mathbf{w}_{y,r}^{(t)}, y\boldsymbol{\varphi}\right\rangle\right)}{(P-1)\sigma_p \sum_{r=1}^m \left\|\mathbf{w}_{-y,r}^{(t)}\right\|_2}\right)^2\right).$$
$$\tag{24}$$

The second inequality is by Eq. 23 and plugging $\|g\|_{\mathrm{Lip}} \leq \sum_{r=1}^{m} \|\mathbf{w}_{-y,r}^{(t)}\|_2$ into Eq. 15; the third inequality is due to $(s-t)^2 \geq s^2/2 - t^2, \forall s, t \geq 0$. And from Eq. 22 and Eq. 24 we have

$$
\begin{aligned}
\mathbb{P}_{(\mathbf{x},y)\sim\mathcal{R}} \left( yf\left(\mathbf{W}^{(t)}, \mathbf{x}\right) \leq 0 \right) &\leq \exp(c/2\pi) \exp\left( -0.5c \left( \frac{\sum_r \sigma\left(\left\langle \mathbf{w}_{y,r}^{(t)}, y\boldsymbol{\varphi} \right\rangle\right)}{(P-1)\sigma_p \sum_{r=1}^{m} \left\| \mathbf{w}_{-y,r}^{(t)} \right\|_2} \right)^2 \right) \\
&= \exp\left( \frac{c}{2\pi} - \frac{n\|\boldsymbol{\varphi}\|_2^4}{C(P-1)^4 \sigma_p^4 d} \right) \\
&\leq \exp\left( -\frac{n\|\boldsymbol{\varphi}\|_2^4}{2C_1(P-1)^4 \sigma_p^4 d} \right) \\
&= \exp\left( -\frac{n\|\boldsymbol{\varphi}\|_2^4}{C_2(P-1)^4 \sigma_p^4 d} \right) = \epsilon,
\end{aligned}
\tag{25}
$$

where $C = O(1)$; the last inequality holds if we choose $C_1 \geq cC/\pi$; the last equality holds if we choose $C_2$ as $2C$.

For the forget set $\mathcal{F}$, we thus have

$$
\mathbb{P}_{(\mathbf{x},y)\sim\mathcal{F}} \left( yf\left(\mathbf{W}^{(t)}, \mathbf{x}\right) > 0 \right) \leq \epsilon.
\tag{26}
$$

**Lower bound.** Without loss of generality, let $\sum_r \kappa_{1,r}^{(t)} = \max\left\{ \sum_r \kappa_{1,r}^{(t)}, \sum_r \kappa_{-1,r}^{(t)} \right\}$. Denote $\mathbf{v} = \lambda \cdot \sum_i \mathbb{1}(y_i = 1)\boldsymbol{\xi}_i$, where $\lambda = C_7 \mathrm{SNR}^2 = C_7 \|\boldsymbol{\varphi}\|_2^2 / \left((P-1)^2 \sigma_p^2 d\right)$ and $C_7$ is a sufficiently large constant. Since ReLU is convex, we have

$$
\begin{aligned}
\sigma\left(\left\langle \mathbf{w}_{1,r}^{(t)}, \boldsymbol{\xi} + \mathbf{v} \right\rangle\right) - \sigma\left(\left\langle \mathbf{w}_{1,r}^{(t)}, \boldsymbol{\xi} \right\rangle\right) &\geq \sigma'\left(\left\langle \mathbf{w}_{1,r}^{(t)}, \boldsymbol{\xi} \right\rangle\right) \left\langle \mathbf{w}_{1,r}^{(t)}, \mathbf{v} \right\rangle, \\
\sigma\left(\left\langle \mathbf{w}_{1,r}^{(t)}, -\boldsymbol{\xi} + \mathbf{v} \right\rangle\right) - \sigma\left(\left\langle \mathbf{w}_{1,r}^{(t)}, -\boldsymbol{\xi} \right\rangle\right) &\geq \sigma'\left(\left\langle \mathbf{w}_{1,r}^{(t)}, -\boldsymbol{\xi} \right\rangle\right) \left\langle \mathbf{w}_{1,r}^{(t)}, \mathbf{v} \right\rangle.
\end{aligned}
\tag{27}
$$

Summing the above two, we have that almost surely for all $\boldsymbol{\xi}$

$$
\begin{aligned}
&\sigma\left(\left\langle \mathbf{w}_{1,r}^{(t)}, \boldsymbol{\xi} + \mathbf{v} \right\rangle\right) - \sigma\left(\left\langle \mathbf{w}_{1,r}^{(t)}, \boldsymbol{\xi} \right\rangle\right) + \sigma\left(\left\langle \mathbf{w}_{1,r}^{(t)}, -\boldsymbol{\xi} + \mathbf{v} \right\rangle\right) - \sigma\left(\left\langle \mathbf{w}_{1,r}^{(t)}, -\boldsymbol{\xi} \right\rangle\right) \\
&\geq \left\langle \mathbf{w}_{1,r}^{(t)}, \mathbf{v} \right\rangle \\
&\geq \lambda \left[ \sum_{y_i=1} \overline{\zeta}_{1,r,i}^{(t)} - 2n\sqrt{\log(12mn/\delta)} \cdot \sigma_0 \sigma_p \sqrt{d} - 5n^2\alpha\sqrt{\log\left(6n^2/\delta\right)/d} \right],
\end{aligned}
\tag{28}
$$

where the last inequality is by Lemma C.3 in Kou et al. (2023) and Lemma D.4. Additionally, since ReLU is a Liptchitz, we also have that

$$
\begin{aligned}
&\sigma\left(\left\langle \mathbf{w}_{-1,r}^{(t)}, \boldsymbol{\xi} + \mathbf{v} \right\rangle\right) - \sigma\left(\left\langle \mathbf{w}_{-1,r}^{(t)}, \boldsymbol{\xi} \right\rangle\right) + \sigma\left(\left\langle \mathbf{w}_{-1,r}^{(t)}, -\boldsymbol{\xi} + \mathbf{v} \right\rangle\right) - \sigma\left(\left\langle \mathbf{w}_{-1,r}^{(t)}, -\boldsymbol{\xi} \right\rangle\right) \\
&\leq 2\left|\left\langle \mathbf{w}_{-1,r}^{(t)}, \mathbf{v} \right\rangle\right| \\
&\leq 2\lambda \left[ \sum_{y_i=1} \zeta_{-1,r,i}^{(t)} + 2n\sqrt{\log(12mn/\delta)} \cdot \sigma_0 \sigma_p \sqrt{d} + 5n^2\alpha\sqrt{\log\left(6n^2/\delta\right)/d} \right].
\end{aligned}
\tag{29}
$$

Therefore, by plugging Eq. 28 and Eq. 29, we have that

$$
\begin{aligned}
& g(\boldsymbol{\xi} + \mathbf{v}) - g(\boldsymbol{\xi}) + g(-\boldsymbol{\xi} + \mathbf{v}) - g(-\boldsymbol{\xi}) \\
& \geq \lambda \left[ \sum_r \sum_{y_i=1} \overline{\zeta}_{1,r,i}^{(t)} - 6nm\sqrt{\log(12mn/\delta)} \cdot \sigma_0 \sigma_p \sqrt{d} - 15mn^2\alpha\sqrt{\log\left(6n^2/\delta\right)/d} \right] \\
& \geq (\lambda/2) \cdot \sum_r \sum_{y_i=1} \overline{\zeta}_{1,r,i}^{(t)} \\
& \geq \lambda/2 \cdot \Theta\left(\mathrm{SNR}^{-2}\right) \sum_r \kappa_{1,r}^{(t)} \\
& \geq 4C_6 \sum_r \kappa_{1,r}^{(t)},
\end{aligned}
\tag{30}
$$

where the second inequality is by Lemma D.1 in Kou et al. (2023) and Assumption D.1; the third inequality is by $\sum_{i=1}^n \overline{\zeta}_{j,r,i}^{(t)}/\kappa_{j',r'}^{(t)} = \Theta(\mathrm{SNR}^{-2})$. Finally, it is worth noting that the norm

$$
\|\mathbf{v}\|_2 = \left\| \lambda \cdot \sum_i \mathbb{1}\left(y_i = 1\right) \boldsymbol{\xi}_i \right\|_2 = \Theta\left( \sqrt{\frac{n\|\boldsymbol{\varphi}\|_2^4}{P^4 \sigma_p^4 d}} \right) \leq 0.06\sigma_p.
\tag{31}
$$

where the last inequality is by condition $\|\boldsymbol{\varphi}\|_2 \leq C_3 d^{1/4} n^{-1/4} P\sigma_p$ with sufficiently large $C_3$. Then we present a Lemma which bounds the Total Variation (TV) distance between two Gaussian with the same covariance matrix.

**Lemma D.5** *(Proposition 2.1 by Devroye et al. (2018)). The TV distance between $\mathcal{N}\left(0, \sigma_p^2 \mathbf{I}_d\right)$ and $\mathcal{N}\left(\mathbf{v}, \sigma_p^2 \mathbf{I}_d\right)$ is smaller than $\|\mathbf{v}\|_2/2\sigma_p$.*

Finally, we can prove the lower bound for $\mathcal{R}$:

$$
\begin{aligned}
& \mathbb{P}_{(\mathbf{x},y)\sim\mathcal{R}} \left( yf\left(\mathbf{W}^{(t)}, \mathbf{x}\right) \leq 0 \right) \\
& = \mathbb{P}_{(\mathbf{x},y)\sim\mathcal{R}} \left( \sum_r \sigma\left(\left\langle \mathbf{w}_{-y,r}^{(t)}, \boldsymbol{\xi} \right\rangle\right) - \sum_r \sigma\left(\left\langle \mathbf{w}_{y,r}^{(t)}, \boldsymbol{\xi} \right\rangle\right) \geq \sum_r \sigma\left(\left\langle \mathbf{w}_{y,r}^{(t)}, y\boldsymbol{\varphi} \right\rangle\right) - \sum_r \sigma\left(\left\langle \mathbf{w}_{-y,r}^{(t)}, y\boldsymbol{\varphi} \right\rangle\right) \right) \\
& \geq 0.5\mathbb{P}_{(\mathbf{x},y)\sim\mathcal{R}} \left( \left| \sum_r \sigma\left(\left\langle \mathbf{w}_{-y,r}^{(t)}, \boldsymbol{\xi} \right\rangle\right) - \sum_r \sigma\left(\left\langle \mathbf{w}_{y,r}^{(t)}, \boldsymbol{\xi} \right\rangle\right) \right| \geq C_6 \max\left\{ \sum_r \kappa_{1,r}^{(t)}, \sum_r \kappa_{-1,r}^{(t)} \right\} \right),
\end{aligned}
\tag{32}
$$

where $C_6$ is a constant, the inequality holds since if $|\sum_r \sigma(\langle \mathbf{w}_{1,r}^{(t)}, \boldsymbol{\xi} \rangle) - \sum_r \sigma(\langle \mathbf{w}_{-1,r}^{(t)}, \boldsymbol{\xi} \rangle)|$ is too large, we can always pick a corresponding $y$ given $\boldsymbol{\xi}$ to make a wrong prediction.

Let $g(\boldsymbol{\xi}) = \sum_r \sigma(\langle \mathbf{w}_{1,r}^{(t)}, \boldsymbol{\xi} \rangle) - \sum_r \sigma(\langle \mathbf{w}_{-1,r}^{(t)}, \boldsymbol{\xi} \rangle)$, and denote the set $\Omega := \{\boldsymbol{\xi} \mid |g(\boldsymbol{\xi})| \geq C_6 \max\{\sum_r \kappa_{1,r}^{(t)}, \sum_r \kappa_{-1,r}^{(t)}\}\}$. Thus we have

$$
\mathbb{P}_{(\mathbf{x},y)\sim\mathcal{R}} \left( yf\left(\boldsymbol{W}^{(t)}, \mathbf{x}\right) \leq 0 \right) \geq 0.5\mathbb{P}(\Omega).
\tag{33}
$$

By Lemma 5.8 of Kou et al. (2023), we have that $\sum_j [g(j\boldsymbol{\xi} + \mathbf{v}) - g(j\boldsymbol{\xi})] \geq 4C_6 \max_j \left\{ \sum_r \kappa_{j,r}^{(t)} \right\}$. Therefore, by pigeonhole principle, one of $[\boldsymbol{\xi}, -\boldsymbol{\xi}, \boldsymbol{\xi}+\mathbf{v}, -\boldsymbol{\xi}+\mathbf{v}]$ must belong to $\Omega$, thus $\Omega \cup -\Omega \cup \Omega - \{\mathbf{v}\} \cup -\Omega - \{\mathbf{v}\} = \mathbb{R}^d$. Therefore, at least one of $\mathbb{P}(\Omega), \mathbb{P}(-\Omega), \mathbb{P}(\Omega - \{\mathbf{v}\}), \mathbb{P}(-\Omega - \{\mathbf{v}\})$ is greater than $\frac{1}{4}$. Note that $\mathbb{P}(-\Omega) = \mathbb{P}(\Omega)$ and

$$
\begin{aligned}
|\mathbb{P}(\Omega) - \mathbb{P}(\Omega - \mathbf{v})| &= \left| \mathbb{P}_{\boldsymbol{\xi}\sim\mathcal{N}\left(0,\sigma_p^2\mathbf{I}_d\right)}(\boldsymbol{\xi} \in \Omega) - \mathbb{P}_{\boldsymbol{\xi}\sim\mathcal{N}\left(\mathbf{v},\sigma_p^2\mathbf{I}_d\right)}(\boldsymbol{\xi} \in \Omega) \right| \\
&\leq \mathrm{TV}\left( \mathcal{N}\left(0,\sigma_p^2\mathbf{I}_d\right), \mathcal{N}\left(\mathbf{v},\sigma_p^2\mathbf{I}_d\right) \right) \\
&\leq \frac{\|\mathbf{v}\|_2}{2\sigma_p} \leq 0.03,
\end{aligned}
\tag{34}
$$

where the first inequality is by the definition of TV distance, the second inequality is by Lemma D.5. Hence, we have that $\mathbb{P}(\Omega) \geq \frac{1}{4} - 0.03 = 0.22$, and plugging this into Eq. 33, we get

$$\mathbb{P}_{(\mathbf{x},y)\sim\mathcal{R}}\left(yf\left(\mathbf{W}^{(t)},\mathbf{x}\right) \leq 0\right) \geq 0.5\mathbb{P}(\Omega) = 0.11 \geq 0.1. \tag{35}$$

Like the upper bound, the derived lower bounds also applies to $\mathbb{P}_{(\mathbf{x},y)\sim\mathcal{F}}(yf(\mathbf{W}^{(t)},\mathbf{x}) > 0)$. Hence, if $\|\boldsymbol{\varphi}\|_2 \geq C_1 d^{1/4} n^{-1/4} P\sigma_p$,

$$\mathcal{L}^{\text{test}}(\mathbf{W}^{T_2}, \mathcal{D}) = \mathbb{P}_{(\mathbf{x},y)\sim\mathcal{D}}\left(y \neq \text{sign}\left(f\left(\mathbf{W}^{T_2}, \mathbf{x}\right)\right)\right)$$

$$= \beta \cdot \underbrace{\mathbb{P}_{(\mathbf{x},y)\sim\mathcal{R}}\left(yf\left(\mathbf{W}^{T_2}, \mathbf{x}\right) \leq 0\right)}_{\leq \epsilon_{\mathcal{R}}} + (1-\beta) \cdot \left(1 - \underbrace{\mathbb{P}_{(\mathbf{x},y)\sim\mathcal{F}}\left(yf\left(\mathbf{W}^{T_2}, \mathbf{x}\right) > 0\right)}_{\leq \epsilon_{\mathcal{F}}}\right) \tag{36}$$

$$\implies \lim_{\beta \to 1} \mathcal{L}^{\text{test}}(\mathbf{W}^{T_2}, \mathcal{D}) \leq \epsilon_{\mathcal{R}} = \epsilon.$$

On the other hand, when $\beta \to 0.5$, we have $\lim_{\beta \to 0.5} \mathcal{L}^{\text{test}}(\mathbf{W}^{T_2}, \mathcal{D}) \leq 0.5 + 0.5\epsilon_{\mathcal{R}} - 0.5\epsilon_{\mathcal{F}} = \epsilon$. Depending on the size ratio of $\mathcal{R}$ and $\mathcal{F}$, $\epsilon$ ranges from a very small constant to a minimally PAC-learnable threshold.

For harmful overfitting where $\|\boldsymbol{\varphi}\|_2 \leq C_3 d^{1/4} n^{-1/4} P\sigma_p$,

$$\mathcal{L}^{\text{test}}(\mathbf{W}^{T_2}, \mathcal{D}) = \mathbb{P}_{(\mathbf{x},y)\sim\mathcal{D}}\left(y \neq \text{sign}\left(f\left(\mathbf{W}^{T_2}, \mathbf{x}\right)\right)\right)$$

$$= \beta \cdot \underbrace{\mathbb{P}_{(\mathbf{x},y)\sim\mathcal{R}}\left(yf\left(\mathbf{W}^{T_2}, \mathbf{x}\right) \leq 0\right)}_{\geq 0.1} + (1-\beta) \cdot \left(1 - \underbrace{\mathbb{P}_{(\mathbf{x},y)\sim\mathcal{F}}\left(yf\left(\mathbf{W}^{T_2}, \mathbf{x}\right) > 0\right)}_{\geq 0.1}\right) \tag{37}$$

$$\implies \lim_{\beta \to 1} \mathcal{L}^{\text{test}}(\mathbf{W}^{T_2}, \mathcal{D}) \geq 0.1.$$

On the other hand, when $\beta \to 0.5$, we have $\lim_{\beta \to 0.5} \mathcal{L}^{\text{test}}(\mathbf{W}^{T_2}, \mathcal{D}) \geq 0.05$.

## D.2 PROOF TO THEOREM 3.3

First we have the same decomposition for NegGrad:

$$\mathcal{L}^{\text{test}}(\mathbf{W}^{T_2}, \mathcal{D}) = \mathbb{P}_{(\mathbf{x},y)\sim\mathcal{D}}\left(y \neq \text{sign}\left(f\left(\mathbf{W}^{(t)}, \mathbf{x}\right)\right)\right)$$

$$= \beta \cdot \mathbb{P}_{(\mathbf{x},y)\sim\mathcal{R}}\left(yf\left(\mathbf{W}^{(t)}, \mathbf{x}\right) \leq 0\right) + (1-\beta) \cdot \left(1 - \mathbb{P}_{(\mathbf{x},y)\sim\mathcal{F}}\left(yf\left(\mathbf{W}^{(t)}, \mathbf{x}\right) > 0\right)\right);$$

$$yf\left(\mathbf{W}^{(t)}, \mathbf{x}\right) = \frac{1}{m} \sum_{j,r} yj\left[\sigma\left(\left\langle \mathbf{w}_{j,r}^{(t)}, y\boldsymbol{\varphi}\right\rangle\right) + \sigma\left(\left\langle \mathbf{w}_{j,r}^{(t)}, \boldsymbol{\xi}\right\rangle\right)\right]$$

$$= \frac{1}{m} \sum_{r}\left[\sigma\left(\left\langle \mathbf{w}_{y,r}^{(t)}, y\boldsymbol{\varphi}\right\rangle\right) + (P-1)\sigma\left(\left\langle \mathbf{w}_{y,r}^{(t)}, \boldsymbol{\xi}\right\rangle\right)\right]$$

$$- \frac{1}{m} \sum_{r}\left[\sigma\left(\left\langle \mathbf{w}_{-y,r}^{(t)}, y\boldsymbol{\varphi}\right\rangle\right) + (P-1)\sigma\left(\left\langle \mathbf{w}_{-y,r}^{(t)}, \boldsymbol{\xi}\right\rangle\right)\right]. \tag{38}$$

However, note that for $(\mathbf{x}, y) \sim \mathcal{F}$, SAM gives up its denoising property. We first show this by proving Lemma 3.1.

### D.2.1 PROOF TO LEMMA 3.1

*Proof.* Consider extending Lemma D.5 in Chen et al. (2023) to the NegGrad setting by rewriting $\left\langle \widehat{\boldsymbol{\epsilon}}_{j,r}^{(t,b)}, \boldsymbol{\xi}_k \right\rangle$. First we have the Frobenius norm upper bounded by the same quantity:

$$\|\nabla_{\mathbf{W}}\mathcal{L}_{\mathcal{I}_{t,b}}(\mathbf{W}^{(t,b)})\|_F = \|\alpha\nabla_{\mathbf{W}}\mathcal{L}_{\mathcal{I}_{t,b}^{\mathcal{R}}}(\mathbf{W}^{(t,b)}) - (1-\alpha)\nabla_{\mathbf{W}}\mathcal{L}_{\mathcal{I}_{t,b}^{\mathcal{F}}}(\mathbf{W}^{(t,b)})\|_F$$

$$\leq \alpha\|\nabla_{\mathbf{W}}\mathcal{L}_{\mathcal{I}_{t,b}^{\mathcal{R}}}(\mathbf{W}^{(t,b)})\|_F + (1-\alpha)\|\nabla_{\mathbf{W}}\mathcal{L}_{\mathcal{I}_{t,b}^{\mathcal{F}}}(\mathbf{W}^{(t,b)})\|_F \tag{39}$$

$$= \|\nabla_{\mathbf{W}}\mathcal{L}_{\mathcal{I}_{t,b}}(\mathbf{W}^{(t,b)})\|_F \leq 2\sqrt{2}P\sigma_p\sqrt{d/Bm},$$

where the first inequality comes from triangle inequality; the second equality holds because $\mathcal{R}, \mathcal{F}$ are split from $\mathcal{S}$ and come from the same $\mathcal{D}$, thus having the same gradient norm; the second inequality comes from the original bounds in Chen et al. (2023). Next we expand $\left\langle \widehat{\boldsymbol{\epsilon}}_{j,r}^{(t,b)}, \boldsymbol{\xi}_k \right\rangle$ under NegGrad:

$$
\begin{aligned}
\left\langle \widehat{\boldsymbol{\epsilon}}_{j,r}^{(t,b)}, \boldsymbol{\xi}_k \right\rangle =& \frac{\tau}{mB} \left\| \nabla_{\mathbf{W}} \mathcal{L}_{\mathcal{I}_{t,b}}(\mathbf{W}^{(t,b)}) \right\|_F^{-1} \sum_{i \in \mathcal{I}_{t,b}} \sum_{p \in [P]} \ell_i'^{(t)} j \cdot y_i \sigma'(\langle \mathbf{w}_{j,r}^{(t)}, \mathbf{x}_{i,p} \rangle) \langle \mathbf{x}_{i,p}, \boldsymbol{\xi}_k \rangle \\
=& \frac{\tau}{mB} \left\| \nabla_{\mathbf{W}} \mathcal{L}_{\mathcal{I}_{t,b}}(\mathbf{W}^{(t,b)}) \right\|_F^{-1} \left[ \alpha \sum_{i \in \mathcal{I}_{t,b}^{\mathcal{R}}} \sum_{p \in [P]} \ell_i'^{(t)} j \cdot y_i \sigma'(\langle \mathbf{w}_{j,r}^{(t)}, \mathbf{x}_{i,p} \rangle) \langle \mathbf{x}_{i,p}, \boldsymbol{\xi}_k \rangle \right. \\
& \left. -(1-\alpha) \sum_{i \in \mathcal{I}_{t,b}^{\mathcal{F}}} \sum_{p \in [P]} \ell_i'^{(t)} j \cdot y_i \sigma'(\langle \mathbf{w}_{j,r}^{(t)}, \mathbf{x}_{i,p} \rangle) \langle \mathbf{x}_{i,p}, \boldsymbol{\xi}_k \rangle \right].
\end{aligned}
\tag{40}
$$

Note that $\langle \mathbf{x}_{i,p}, \boldsymbol{\xi}_k \rangle$ can be divided into three different terms:

$$
|\langle \mathbf{x}_{i,p}, \boldsymbol{\xi}_k \rangle| = \begin{cases} \|\boldsymbol{\xi}_k\|_2^2 \leq 3\sigma_p^2 d/2, & \text{if } i = k, x_{k,p} = \boldsymbol{\xi}_k \\ |\langle \boldsymbol{\xi}_i, \boldsymbol{\xi}_k \rangle| \leq 2\sigma_p^2 \sqrt{d \log(6n^2/\delta)}, & \text{if } i \neq k, x_{i,p} = \boldsymbol{\xi}_i \\ |\langle y_i \boldsymbol{\varphi}, \boldsymbol{\xi}_k \rangle| \leq \|\boldsymbol{\varphi}\|_2 \sigma_p \sqrt{2 \log(6n^2/\delta)}, & \text{if } x_{i,p} = y_i \boldsymbol{\varphi} \end{cases}
\tag{41}
$$

The upper bounds come from Lemma D.3. Based on Assumption D.1 and Lemma D.4 of Chen et al. (2023), the $i = k$ term will dominate the upper bound and we can write

$$
\begin{aligned}
\left\langle \widehat{\boldsymbol{\epsilon}}_{j,r}^{(t,b)}, \boldsymbol{\xi}_k \right\rangle \leq & \frac{\tau}{mB \cdot 2\sqrt{2} P \sigma_p \sqrt{d/Bm}} \left[ -0.15\alpha(P-1)C_1\sigma_p^2 d\mathbb{1}[k \in \mathcal{I}_{t,b}^{\mathcal{R}}] \right. \\
& \left. +0.15(1-\alpha)(P-1)C_1\sigma_p^2 d\mathbb{1}[k \in \mathcal{I}_{t,b}^{\mathcal{F}}] \right]
\end{aligned}
\tag{42}
$$

Thus, when $k \in \mathcal{I}_{t,b}^{\mathcal{R}}$, we can preserve the original bound with additional $\alpha$:

$$
\left\langle \widehat{\boldsymbol{\epsilon}}_{j,r}^{(t,b)}, \boldsymbol{\xi}_k \right\rangle < -C \frac{\alpha \tau \sigma_p \sqrt{d}}{m\sqrt{B}}.
\tag{43}
$$

Choosing $\tau = \frac{m\sqrt{B}}{C_3 \alpha P \sigma_p \sqrt{d}}$ will cancel with $\left\langle \mathbf{w}_{j,r}^{(t)}, \boldsymbol{\xi}_k \right\rangle$ to deactivate the neuron. When $k \in \mathcal{I}_{t,b}^{\mathcal{F}}$, the entire $\langle \mathbf{w}_{j,r}^{(t,b)} + \widehat{\boldsymbol{\epsilon}}_{j,r}^{(t,b)}, \boldsymbol{\xi}_k \rangle$ will remain activated:

$$
0 \leq \left\langle \widehat{\boldsymbol{\epsilon}}_{j,r}^{(t,b)}, \boldsymbol{\xi}_k \right\rangle < C \frac{(1-\alpha)\tau \sigma_p \sqrt{d}}{m\sqrt{B}} \implies \left\langle \mathbf{w}_{j,r}^{(t,b)} + \widehat{\boldsymbol{\epsilon}}_{j,r}^{(t,b)}, \boldsymbol{\xi}_k \right\rangle \geq \left\langle \mathbf{w}_{j,r}^{(t,b)}, \boldsymbol{\xi}_k \right\rangle \geq 0.
\tag{44}
$$

This fundamentally differs SAM's behaviors towards unlearning $\mathcal{F}$ from behaviors towards learning $\mathcal{R}$ as how SGD differs from SAM. For gradient ascent on $\mathcal{F}$ under NegGrad, we now know SAM learns from activated noise products as much as SGD. The activation patterns are further utilized to bound products and norms of the weight, signal and noise, which characterize the final test errors.

Our task is reduced to bounding $\mathbb{P}_{(\mathbf{x},y)\sim\mathcal{R}}(yf\left(\mathbf{W}^{(t)},\mathbf{x}\right)\le 0)$, then use previous error bounds for SGD in App. D.1 for $\mathbb{P}_{(\mathbf{x},y)\sim\mathcal{F}}(yf(\mathbf{W}^{(t)},\mathbf{x})>0)$. The inner product with $j=y$ can be bounded as

$$
\begin{aligned}
\left\langle \mathbf{w}_{y,r}^{(t)}, y\boldsymbol{\varphi}\right\rangle &= \left\langle \mathbf{w}_{y,r}^{(0)}, y\boldsymbol{\varphi}\right\rangle + \kappa_{y,r}^{(t)} + \frac{1}{(P-1)}\sum_{i=1}^{n}\overline{\zeta}_{y,r,i}^{(t)}\cdot\|\boldsymbol{\xi}_i\|_2^{-2}\cdot\langle\boldsymbol{\xi}_i, y\boldsymbol{\varphi}\rangle\\
&\quad + \frac{1}{(P-1)}\sum_{i=1}^{n}\underline{\varsigma}_{y,r,i}^{(t)}\cdot\|\boldsymbol{\xi}_i\|_2^{-2}\cdot\langle\boldsymbol{\xi}_i, y\boldsymbol{\varphi}\rangle\\
&\ge \left\langle \mathbf{w}_{y,r}^{(0)}, y\boldsymbol{\varphi}\right\rangle + \kappa_{y,r}^{(t)}\\
&\quad - \frac{\sqrt{2\log(6n/\delta)}}{P-1}\cdot\sigma_p\|\boldsymbol{\varphi}\|_2\cdot\left(\sigma_p^2 d/2\right)^{-1}\left[\sum_{i=1}^{n}\overline{\zeta}_{y,r,i}^{(t)} + \sum_{i=1}^{n}\left|\underline{\varsigma}_{y,r,i}^{(t)}\right|\right]\\
&= \left\langle \mathbf{w}_{y,r}^{(0)}, y\boldsymbol{\varphi}\right\rangle + \kappa_{y,r}^{(t)} - \Theta\left(\sqrt{\log(n/\delta)}\cdot(P\sigma_p d)^{-1}\|\boldsymbol{\varphi}\|_2\right)\cdot\Theta\left(\mathrm{SNR}^{-2}\right)\cdot\kappa_{y,r}^{(t)}\\
&= \left\langle \mathbf{w}_{y,r}^{(0)}, y\boldsymbol{\varphi}\right\rangle + \left[1 - \Theta\left(\sqrt{\log(n/\delta)}\cdot P\sigma_p/\|\boldsymbol{\varphi}\|_2\right)\right]\kappa_{y,r}^{(t)}\\
&= \left\langle \mathbf{w}_{y,r}^{(0)}, y\boldsymbol{\varphi}\right\rangle + \Theta\left(\kappa_{y,r}^{(t)}\right) = \Theta(1),
\end{aligned}
$$

(45)

where the inequality is by Lemma D.3; the second equality is obtained by plugging in the coefficient orders we summarized; the third equality is by $\mathrm{SNR} = \|\boldsymbol{\varphi}\|_2/(P\sigma_p\sqrt{d})$; the fourth equality is by $\|\boldsymbol{\varphi}\|_2^2 \ge C\cdot P^2\sigma_p^2\log(n/\delta)$ in Assumption D.1 for sufficiently large constant $C$; the last equality is by Lemma D.7 of Chen et al. (2023). We similarly have $\langle\mathbf{w}_{y,r}^{(t)}, y\boldsymbol{\varphi}\rangle = -\Theta(1) < 0$.

Denote $g(\boldsymbol{\xi})$ as $\sum_r\sigma(\langle\mathbf{w}_{-y,r}^{(t)}, \boldsymbol{\xi}\rangle)$. The results for noise learning from SGD in App. D.1 still apply:

$$
\begin{aligned}
|g(\boldsymbol{\xi}) - g\left(\boldsymbol{\xi}'\right)| &\le \sum_{r=1}^{m}\left\|\mathbf{w}_{-y,r}^{(t)}\right\|_2\cdot\|\boldsymbol{\xi} - \boldsymbol{\xi}'\|_2;\\
\mathbb{E}g(\boldsymbol{\xi}) &= \frac{\sigma_p}{\sqrt{2\pi}}\sum_{r=1}^{m}\left\|\mathbf{w}_{-y,r}^{(t)}\right\|_2;\\
\left\|\sum_{i=1}^{n}\varsigma_{j,r,i}^{(t)}\cdot\|\boldsymbol{\xi}_i\|_2^{-2}\cdot\boldsymbol{\xi}_i\right\|_2^2 &\le \Theta\left(\sigma_p^{-2}d^{-1}n^{-1}\right)\left(\sum_{i=1}^{n}\overline{\zeta}_{j,r,i}^{(t)}\right)^2.
\end{aligned}
$$

(46)

We can thus upper bound the 2-norm of $\mathbf{w}_{j,r}^{(t)}$ as:

$$
\begin{aligned}
\left\|\mathbf{w}_{j,r}^{(t)}\right\|_2 &\le \left\|\mathbf{w}_{j,r}^{(0)}\right\|_2 + \kappa_{j,r}^{(t)}\cdot\|\boldsymbol{\varphi}\|_2^{-1} + \frac{1}{P-1}\left\|\sum_{i=1}^{n}\varsigma_{j,r,i}^{(t)}\cdot\|\boldsymbol{\xi}_i\|_2^{-2}\cdot\boldsymbol{\xi}_i\right\|_2\\
&\le \left\|\mathbf{w}_{j,r}^{(0)}\right\|_2 + \kappa_{j,r}^{(t)}\cdot\|\boldsymbol{\varphi}\|_2^{-1} + \Theta\left(P^{-1}\sigma_p^{-1}d^{-1/2}n^{-1/2}\right)\cdot\sum_{i=1}^{n}\overline{\zeta}_{j,r,i}^{(t)}\\
&= \Theta(\sigma_0\sqrt{d}) + \Theta\left(P^{-1}\sigma_p^{-1}d^{-1/2}n^{-1/2}\right)\cdot\sum_{i=1}^{n}\overline{\zeta}_{j,r,i}^{(t)},
\end{aligned}
$$

(47)

based on $\mathrm{SNR} = \|\boldsymbol{\varphi}\|_2/(P\sigma_p\sqrt{d})$ and $\sum_{i=1}^{n}\overline{\zeta}_{j,r,i}^{(t)}/\kappa_{j,r}^{(t)} = \Theta\left(\mathrm{SNR}^{-2}\right)$, and the condition for $d$ in Assumption D.1, and also $\left\|\mathbf{w}_{j,r}^{(0)}\right\|_2 = \Theta\left(\sigma_0\sqrt{d}\right)$ based on Lemma D.7 of Chen et al. (2023). Then

we have

$$
\begin{aligned}
\frac{\sum_r \sigma\left(\left\langle \mathbf{w}_{y,r}^{(t)}, y\boldsymbol{\varphi}\right\rangle\right)}{(P-1)\sigma_p \sum_{r=1}^m \left\|\mathbf{w}_{-y,r}^{(t)}\right\|_2} &\geq \frac{\Theta(1)}{\Theta\left(\sigma_0\sqrt{d}\right) + \Theta\left(P^{-1}\sigma_p^{-1}d^{-1/2}n^{-1/2}\right)\cdot\sum_{i=1}^n \bar{\zeta}_{j,r,i}^{(t)}} \\
&\geq \frac{\Theta(1)}{\Theta\left(\sigma_0\sqrt{d}\right) + O\left(P^{-1}\sigma_p^{-1}d^{-1/2}n^{1/2}\alpha\right)} \\
&\geq \min\left\{\Omega\left(\sigma_0^{-1}d^{-1/2}\right), \Omega\left(P\sigma_p d^{1/2}n^{-1/2}\alpha^{-1}\right)\right\} \\
&\geq 1 \\
\implies \sum_r \sigma\left(\left\langle \mathbf{w}_{y,r}^{(t)}, y\boldsymbol{\varphi}\right\rangle\right) &- \frac{(P-1)\sigma_p}{\sqrt{2\pi}}\sum_{r=1}^m \left\|\mathbf{w}_{-y,r}^{(t)}\right\|_2 > 0.
\end{aligned}
\tag{48}
$$

**Upper bound.** Now plug in previous results to obtain

$$
\begin{aligned}
\mathbb{P}_{(\mathbf{x},y)\sim\mathcal{R}}\left(yf\left(\mathbf{W}^{(t)},\mathbf{x}\right)\leq 0\right) &\leq \mathbb{P}_{(\mathbf{x},y)\sim\mathcal{R}}\left((P-1)\sum_r \sigma\left(\left\langle \mathbf{w}_{-y,r}^{(t)}, \boldsymbol{\xi}\right\rangle\right) \geq \sum_r \sigma\left(\left\langle \mathbf{w}_{y,r}^{(t)}, y\boldsymbol{\varphi}\right\rangle\right)\right) \\
&= \mathbb{P}_{(\mathbf{x},y)\sim\mathcal{R}}\left(g(\boldsymbol{\xi}) - \mathbb{E}g(\boldsymbol{\xi}) \geq 1/(P-1)\sum_r \sigma\left(\left\langle \mathbf{w}_{y,r}^{(t)}, y\boldsymbol{\varphi}\right\rangle\right) - \frac{\sigma_p}{\sqrt{2\pi}}\sum_{r=1}^m \left\|\mathbf{w}_{-y,r}^{(t)}\right\|_2\right) \\
&\leq \exp\left[-\frac{c\left(1/(P-1)\sum_r \sigma\left(\left\langle \mathbf{w}_{y,r}^{(t)}, y\boldsymbol{\varphi}\right\rangle\right) - \left(\sigma_p/\sqrt{2\pi}\right)\sum_{r=1}^m \left\|\mathbf{w}_{-y,r}^{(t)}\right\|_2\right)^2}{\sigma_p^2\left(\sum_{r=1}^m \left\|\mathbf{w}_{-y,r}^{(t)}\right\|_2\right)^2}\right] \\
&= \exp\left[-c\left(\frac{\sum_r \sigma\left(\left\langle \mathbf{w}_{y,r}^{(t)}, y\boldsymbol{\varphi}\right\rangle\right)}{(P-1)\sigma_p \sum_{r=1}^m \left\|\mathbf{w}_{-y,r}^{(t)}\right\|_2} - 1/\sqrt{2\pi}\right)^2\right] \\
&\leq \exp(c/2\pi)\exp\left(-0.5c\left(\frac{\sum_r \sigma\left(\left\langle \mathbf{w}_{y,r}^{(t)}, y\boldsymbol{\varphi}\right\rangle\right)}{(P-1)\sigma_p \sum_{r=1}^m \left\|\mathbf{w}_{-y,r}^{(t)}\right\|_2}\right)^2\right).
\end{aligned}
\tag{49}
$$

The second inequality is by Eq. 48 and plugging $\|g\|_{\mathrm{Lip}} \leq \sum_{r=1}^m \|\mathbf{w}_{-y,r}^{(t)}\|_2$ into Eq. 15, the third inequality is because $(s-t)^2 \geq s^2/2 - t^2, \forall s,t \geq 0$. And we can obtain

$$
\begin{aligned}
\mathbb{P}_{(\mathbf{x},y)\sim\mathcal{R}}\left(yf\left(\mathbf{W}^{(t)},\mathbf{x}\right)\leq 0\right) &\leq \exp(c/2\pi)\exp\left(-0.5c\left(\frac{\sum_r \sigma\left(\left\langle \mathbf{w}_{y,r}^{(t)}, y\boldsymbol{\varphi}\right\rangle\right)}{(P-1)\sigma_p \sum_{r=1}^m \left\|\mathbf{w}_{-y,r}^{(t)}\right\|_2}\right)^2\right) \\
&\leq \exp\left(\frac{c}{2\pi} - C\min\left\{\sigma_0^{-2}d^{-1}, P\sigma_p^2 dn^{-1}\alpha^{-2}\right\}\right) \\
&\leq \exp\left(-0.5C\min\left\{\sigma_0^{-2}d^{-1}, P\sigma_p^2 dn^{-1}\alpha^{-2}\right\}\right) = \epsilon,
\end{aligned}
\tag{50}
$$

where $C = O(1)$, the last inequality holds since $\sigma_0^2 \leq 0.5Cd^{-1}\log(1/\epsilon)$ and $d \geq 2C^{-1}P^{-1}\sigma_p^{-2}n\alpha^2\log(1/\epsilon)$. Now we upper bound the test error $\mathcal{L}^{\mathrm{test}}(\mathbf{W}^{T_2},\mathcal{D})$. Depending on the strength of the unified signal vector $\boldsymbol{\varphi}$, the unlearning of $\mathcal{F}$ can exhibit either benign or harmful overfitting following SGD's characterization, dividing error bounds into two cases:

1. If $\|\boldsymbol{\varphi}\|_2 \geq C_1 d^{1/4} n^{-1/4} P \sigma_p$, we have benign overfitting on both $\mathcal{R}$ and $\mathcal{F}$. Thus,

$$\mathcal{L}^{\text{test}}(\mathbf{W}^{T_2}, \mathcal{D}) = \mathbb{P}_{(\mathbf{x},y)\sim\mathcal{D}}\left(y \neq \text{sign}\left(f\left(\mathbf{W}^{T_2}, \mathbf{x}\right)\right)\right)$$

$$= \beta \cdot \underbrace{\mathbb{P}_{(\mathbf{x},y)\sim\mathcal{R}}\left(yf\left(\mathbf{W}^{T_2}, \mathbf{x}\right) \leq 0\right)}_{\leq \epsilon_{\mathcal{R}}} + (1-\beta) \cdot \left(1 - \underbrace{\mathbb{P}_{(\mathbf{x},y)\sim\mathcal{F}}\left(yf\left(\mathbf{W}^{T_2}, \mathbf{x}\right) > 0\right)}_{\leq \epsilon_{\mathcal{F}}}\right)$$

$$\implies \lim_{\beta\to 1} \mathcal{L}^{\text{test}}(\mathbf{W}^{T_2}, \mathcal{D}) \leq \epsilon_{\mathcal{R}} = \epsilon.$$

(51)

As $\beta \to 1$, $|\mathcal{F}|/n$ decreases so the model can better maintain its performance; as $\beta \to 0.5$, $|\mathcal{F}|/n$ increases and more samples are to be unlearned, making the model performance reduce to a minimally PAC-learnable guarantee. Hence, when $\beta \to 0.5$, we have $\lim_{\beta\to 0.5} \mathcal{L}^{\text{test}}(\mathbf{W}^{T_2}, \mathcal{D}) \leq 0.5 + 0.5\epsilon_{\mathcal{R}} - 0.5\epsilon_{\mathcal{F}} = \epsilon$.

2. If $\Omega(1) \leq \|\boldsymbol{\varphi}\|_2 \leq C_1 d^{1/4} n^{-1/4} P \sigma_p$, we have benign overfitting on $\mathcal{R}$ and harmful overfitting on $\mathcal{F}$. Thus,

$$\mathcal{L}^{\text{test}}(\mathbf{W}^{T_2}, \mathcal{D}) = \mathbb{P}_{(\mathbf{x},y)\sim\mathcal{D}}\left(y \neq \text{sign}\left(f\left(\mathbf{W}^{(t)}, \mathbf{x}\right)\right)\right)$$

$$= \beta \cdot \underbrace{\mathbb{P}_{(\mathbf{x},y)\sim\mathcal{R}}\left(yf\left(\mathbf{W}^{(t)}, \mathbf{x}\right) \leq 0\right)}_{\leq \epsilon_{\mathcal{R}}} + (1-\beta) \cdot \left(1 - \underbrace{\mathbb{P}_{(\mathbf{x},y)\sim\mathcal{F}}\left(yf\left(\mathbf{W}^{(t)}, \mathbf{x}\right) > 0\right)}_{\geq 0.1}\right)$$

$$\implies \lim_{\beta\to 1} \mathcal{L}^{\text{test}}(\mathbf{W}^{T_2}, \mathcal{D}) \leq \epsilon_{\mathcal{R}} = \epsilon.$$

(52)

Similarly, we have $\lim_{\beta\to 0.5} \mathcal{L}^{\text{test}}(\mathbf{W}^{T_2}, \mathcal{D}) \leq 0.5\epsilon_{\mathcal{R}} + 0.45 = \epsilon$.

**Remark D.6** ($\beta$-dependence of the $\epsilon$-bound). The overall test error

$$\mathcal{L}^{test}(\mathbf{W}^{T_2}, \mathcal{D}) = \beta \cdot \mathbb{P}_{(\mathbf{x},y)\sim\mathcal{R}}\left(yf\left(\mathbf{W}^{(t)}, \mathbf{x}\right) \leq 0\right) + (1-\beta) \cdot \left(1 - \mathbb{P}_{(\mathbf{x},y)\sim\mathcal{F}}\left(yf\left(\mathbf{W}^{(t)}, \mathbf{x}\right) > 0\right)\right)$$

can be considered as an affine function of the mixing factor $\beta$, and so its achievable range runs from the best-case retain error $\epsilon_{\mathcal{R}}$ (as $\beta \to 1$) up to asymptotically $0.5$ (as $\beta \to 0.5$)—the trivial PAC-learnability threshold. Concretely, by choosing $\beta$ sufficiently close to 1, one drives $\mathcal{L}^{test}(\mathbf{W}^{T_2}, \mathcal{D})$ arbitrarily close to the small "benign" error level $\epsilon$, whereas if $\beta$ remains near $0.5$ then $\mathcal{L}^{test}(\mathbf{W}^{T_2}, \mathcal{D})$ can approach $0.5$, the worst-case "minimally learnable" error. Thus, all our bounds interpolate smoothly between these two extremes via the single parameter $\beta$, and we report the most informative bounds in Theorem 3.2 and Theorem 3.3.

## D.3 PROOF TO COROLLARY 3.3.1

Recall the update rule for $\kappa_{j,r}$. For each epoch, the interference between retain and forget signals can be measured as

$$\sum_b^{|\mathcal{R}|/B} \alpha \sum_{i\in\mathcal{I}_{t,b}^{\mathcal{R}}} \ell_i'^{(t,b)} \sigma'(\langle \mathbf{w}_{j,r}^{(t,b)}, y_i\boldsymbol{\varphi}\rangle) - \sum_b^{|\mathcal{F}|/B} (1-\alpha)\frac{|\mathcal{R}|}{|\mathcal{F}|} \sum_{i\in\mathcal{I}_{t,b}^{\mathcal{F}}} \ell_i'^{(t,b)} \sigma'(\langle \mathbf{w}_{j,r}^{(t,b)}, y_i\boldsymbol{\varphi}\rangle). \quad (53)$$

Similar to Lemma 3.1, the expected gradient values between retain and forget samples should not differ. Since we cycle the forget set to synchronously train with the retain set, updates from $\mathcal{F}$ has been scaled up by $\frac{|\mathcal{R}|}{|\mathcal{F}|}$. Hence,

$$\mathbb{E}\left[\sum_b^{|\mathcal{R}|/B} \sum_{i\in\mathcal{I}_{t,b}^{\mathcal{R}}} \ell_i'^{(t,b)} \sigma'(\langle \mathbf{w}_{j,r}^{(t,b)}, y_i\boldsymbol{\varphi}\rangle)\right] = \mathbb{E}\left[\sum_b^{|\mathcal{F}|/B} \sum_{i\in\mathcal{I}_{t,b}^{\mathcal{F}}} \ell_i'^{(t,b)} \sigma'(\langle \mathbf{w}_{j,r}^{(t,b)}, y_i\boldsymbol{\varphi}\rangle)\right] \quad (54)$$

Combining together, to expect $\kappa_{j,r}$ to increase monotonically every epoch, we want

$$\mathbb{E}\left[\sum_b^{|\mathcal{R}|/B} \alpha \sum_{i\in\mathcal{I}_{t,b}^{\mathcal{R}}} \ell_i'^{(t,b)}\sigma'(\langle\mathbf{w}_{j,r}^{(t,b)}, y_i\boldsymbol{\varphi}\rangle) - \sum_b^{|\mathcal{F}|/B}(1-\alpha)\frac{|\mathcal{R}|}{|\mathcal{F}|}\sum_{i\in\mathcal{I}_{t,b}^{\mathcal{F}}} \ell_i'^{(t,b)}\sigma'(\langle\mathbf{w}_{j,r}^{(t,b)}, y_i\boldsymbol{\varphi}\rangle)\right] \geq 0$$

$$\implies \alpha - (1-\alpha)\frac{|\mathcal{R}|}{|\mathcal{F}|} \geq 0 \implies \alpha \geq \frac{|\mathcal{R}|}{|\mathcal{F}|+|\mathcal{R}|}.$$

(55)

### D.4 PROOF TO LEMMA 3.4

By Theorem 3.3, SAM turns off noise memorization prevention mechanism when fitting $\mathcal{F}$, which leads to the same requirement on signal strength as SGD. The only difference between SAM and SGD under NegGrad is the more effective learning on $\mathcal{R}$. From Eq. 7 we have the per-batch update of $\kappa_{j,r}$ on $\mathcal{R}$ as

$$\Delta\kappa_{j,r} = \frac{\eta\|\boldsymbol{\varphi}\|_2^2}{Bm}\alpha \sum_{i\in\mathcal{I}_{t,b}^{\mathcal{R}}} \ell_i'^{(t,b)}\sigma'(\langle\mathbf{w}_{j,r}^{(t,b)}, y_i\boldsymbol{\varphi}\rangle).$$

(56)

Let $g$ denote the batch-average magnitude of $\ell_i'^{(t,b)}\sigma'(\langle\mathbf{w}_{j,r}^{(t,b)}, y_i\boldsymbol{\varphi}\rangle)$ for convenience. We can then express per-epoch $\kappa$ update as

$$\Delta_{\text{epoch}}\kappa_{j,r} = \frac{\eta\|\boldsymbol{\varphi}\|_2^2}{m}\alpha|\mathcal{R}|g.$$

(57)

Now, consider achieving benign overfitting on $\mathcal{R}$ only, where SGD requires $\|\boldsymbol{\varphi}\|_2 = \Omega(d^{1/4}|\mathcal{R}|^{-1/4}P\sigma_p)$ while SAM only requires $\|\boldsymbol{\varphi}\|_2 = \Omega(1)$. That being said, given a fixed universal $\boldsymbol{\varphi}$ for $\mathcal{D}$ and a choice of $\alpha$, we have SAM learning the retain signals faster than SGD:

$$\frac{\Delta_{\text{epoch}}\kappa_{j,r}^{\text{SAM}}}{\Delta_{\text{epoch}}\kappa_{j,r}^{\text{SGD}}} = \Theta(d^{1/2}|\mathcal{R}|^{-1/2}P^2\sigma_p^2) = \Theta(\|\boldsymbol{\varphi}\|_2^2).$$

(58)

Hence, in order to achieve the same signal learning performance as SAM on $\mathcal{R}$, SGD needs to scale up $\alpha^{\text{SGD}}$. Thus,

$$\frac{\alpha^{\text{SGD}}}{\alpha^{\text{SAM}}} = \Theta(d^{1/2}|\mathcal{R}|^{-1/2}P^2\sigma_p^2) = \Theta(\|\boldsymbol{\varphi}\|_2^2), \text{ or } \alpha^{\text{SGD}} - \alpha^{\text{SAM}} = \Theta(\|\boldsymbol{\varphi}\|_2^2).$$

(59)

In general, since $|\mathcal{R}| = \Theta(n)$, we can characterize the gap between $\alpha^{\text{SGD}}$ and $\alpha^{\text{SAM}}$ by $O(\sqrt{d/n})$.

## E IMPLEMENTATION DETAILS

### E.1 EXPERIMENT SETUP

We conduct major experiments on CIFAR-100 (Krizhevsky et al., 2009) and ImageNet-1K (Russakovsky et al., 2015) using ResNet-50 (He et al., 2016). We adopt pre-computed memorization scores for these two datasets from Feldman & Zhang (2020) to generate $\mathcal{F}$ of different memorization levels with $|\mathcal{F}| \approx 5\%|\mathcal{S}|$. We have $|\mathcal{F}| = 3000$ for CIFAR-100 and $|\mathcal{F}| = 60000$ for ImageNet. We sample high-memorization forget set $\mathcal{F}_{\text{high}}$ by choosing $|\mathcal{F}|$ samples of highest memorization scores from $\mathcal{S}$, $\mathcal{F}_{\text{low}}$ by choosing $|\mathcal{F}|$ samples of lowest memorization scores, and $\mathcal{F}_{\text{mid}}$ by choosing $|\mathcal{F}|$ samples whose memorization scores are closest to 0.5. We also run experiments with randomly sampled $\mathcal{F}_{\text{rand}}$ on Tiny-ImageNet and CIFAR-10 in App. G. We use $\mathrm{RandomResizedCrop}$ and $\mathrm{RandomHorizontalFlip}$ as train transforms.

**Pretraining and retraining.** We pretrain on $\mathcal{S}$ and retrain on $\mathcal{R}$ with the same settings. For CIFAR-100, we train for $T_1 = 200$ epochs, use batch size 256, learning rate $\eta_0 = 0.1$ with cosine annealing, SGD with momentum 0.9 and weight decay $5 \times 10^{-4}$. For ImageNet, we train for $T_1 = 150$ epochs, use batch size 512, learning rate $\eta_0 = 0.25$ with cosine annealing and 5 warm-up epochs, SGD with momentum 0.9 and weight decay $2 \times 10^{-5}$. For CIFAR-10, we train ResNet-18 for $T_1 = 50$ epochs, use batch size 256, learning rate $\eta_0 = 0.1$ with cosine annealing, SGD with momentum 0.9 and weight decay $5 \times 10^{-4}$. We summarize the settings, test performance of different pretrained models, as well as accuracies of retrain models in Tab. 6.

Table 6: Pretraining settings and test accuracies using different $\mathcal{A}$ (top), as well as performance of retrained models w.r.t different $\mathcal{F}$ (bottom) for CIFAR-100 and ImageNet-1K.

| Dataset, Model | lr+warmup | Batch $B$ | Epoch $T$ | W. Decay | SGD | ASAM 0.1 | ASAM 1.0 | SAM 0.1 |
|---|---|---|---|---|---|---|---|---|
| CIFAR100, Res50 | 0.1+0 | 256 | 200 | 5e-4 | 77.23 | 76.0 | 78.05 | 77.85 |
| ImageNet, Res50 | 0.25+5 | 512 | 150 | 2e-5 | 75.04 | 74.94 | 76.53 | 76.18 |

| Retrain | High Mem | | | Mid Mem | | | Low Mem | | |
|---|---|---|---|---|---|---|---|---|---|
| Dataset, Model | Retain | Forget | Test | Retain | Forget | Test | Retain | Forget | Test |
| CIFAR100, Res50 | 99.964 | 3.3 | 74.96 | 99.981 | 57.5 | 74.14 | 99.956 | 100.0 | 75.81 |
| ImageNet, Res50 | 97.134 | 13.828 | 74.826 | 97.388 | 52.27 | 74.832 | 96.671 | 99.858 | 75.018 |

Table 7: Ablation on weight mask cutoff choice for Sharp MinMax on CIFAR-100 with ResNet50. We report ToWs across different $\mathcal{F}$ and the averages. We observe that all choices work well: 10% works as well as 30%, while a larger $\mathbf{W}_{\mathcal{F}}$ as 50% can further improve the performance.

| Cutoff | $\mathcal{A}$=SGD | | | | $\mathcal{A}$=ASAM 1.0 | | | |
|---|---|---|---|---|---|---|---|---|
| | $\mathcal{F}_{\text{high}}$ | $\mathcal{F}_{\text{mid}}$ | $\mathcal{F}_{\text{low}}$ | AVG | $\mathcal{F}_{\text{high}}$ | $\mathcal{F}_{\text{mid}}$ | $\mathcal{F}_{\text{low}}$ | AVG |
| 10% | 82.675 | 92.495 | 87.636 | 87.602 | 83.916 | 90.27 | 81.362 | 85.183 |
| 30% | 82.27 | 94.913 | 86.504 | 87.896 | 84.521 | 87.761 | 84.381 | 85.554 |
| 50% | 82.798 | 98.177 | 87.806 | **89.594** | 83.567 | 95.516 | 90.096 | **89.726** |

**Unlearning.** We conduct all unlearning methods for $T_2 = 10$ epochs with the same batch size and optimizer settings. For NegGrad and Sharp MinMax, we unlearn with constant learning rate 0.02. We use $\alpha = 0.99$ for CIFAR-100 and $\alpha = 0.989$ for ImageNet accounting for its slightly smaller $|\mathcal{F}|/|\mathcal{S}|$ ratio. For model splitting, we empirically find that a small ratio for forget model benefits ImageNet such as 5%, while CIFAR-100 suits a larger ratio such as 30%. For both pretraining and unlearning, we wrap SGD with vanilla SAM (Foret et al., 2020) with $\rho = 0.1$, and Adaptive SAM (ASAM) (Kwon et al., 2021) with $\rho = [0.1, 1.0]$, while keep other hyper-parameters the same for fair comparison.

### E.2  Sharp MinMax Implementation

Inspired by SalUn (Fan et al., 2023), we split the model into retain, forget models $\mathbf{W}_{\mathcal{R}}, \mathbf{W}_{\mathcal{F}}$ and update using two separate optimizers: SAM on $\mathbf{W}_{\mathcal{R}}$ and sharpness maximization on $\mathbf{W}_{\mathcal{F}}$. We split the model by ranking the parameters that are important to $\mathcal{F}$ based on the magnitude of the gradient of the parameters after one pass on $\mathcal{F}$, and choose the highest percentage where we have 5% for ImageNet and 30% for CIFAR-100. The cutoff choice is based on the over-parameterization scheme: since ResNet50 w/ CIFAR-100 is much more over-parameterized than w/ ImageNet, there is less overlap between retain and forget parameters and more freedom to increase size of $\mathbf{W}_{\mathcal{F}}$ for more aggressive unlearning. We have also experimented with 10% and 50% and notice a slight better performance of using 50% cutoff in Tab. 7. Unlike SalUn, which essentially performs RL unlearning on the selected parameters, we update both models using opposite optimization. SalUn also requires a larger part of the model to fine-tune with noisy, label flipped $\mathcal{F}$ (50%). We have summarized our implementation for weight masking in Alg. 1, and Sharp MinMax in Alg. 2.

### E.3  Unlearning Setup for Previous Work

We compare with state-of-the-art unlearning methods with optimized hyper-parameter settings. To our best knowledge, several previous methods are evaluated on ImageNet for the first time. We apply SGD and ASAM 1.0 on each $\mathcal{U}$ and compare the performance between SGD and SAM. For L1-Sparse (Jia et al., 2023), we use unlearn lr= 0.02 and $\alpha = 1 \times 10^{-4}$. For SCRUB (Kurmanji et al., 2023), we use unlearn lr= 0.004, msteps= 8, kd_T= 4, $\beta = 0.01$, and $\gamma = 0.99$. For RL (Graves et al., 2021), we use unlearn lr= 0.06 on CIFAR-100 and 0.02 on ImageNet. For SalUn (Fan et al., 2023), we use the unlearn lr= 0.06, 50% weight to finetune on CIFAR-100, and unlearn lr= 0.04, 30% weight to finetune on ImageNet.

---

**Algorithm 1** WeightMask

---

**Require:** forget_loader, model, criterion, percent
1: **for all** (name, param) in model parameters **do**
2:     gradients[name] ← zeros_like(param)
3: **end for**
4: **for all** (image, target) in forget_loader **do**
5:     loss ← criterion(model(image), target)
6:     optimizer.zero_grad(); loss.backward()
7:     accumulate parameter gradients into gradients
8: **end for**
9: **for all** name in gradients **do**
10:     gradients[name] ← |gradients[name]|
11: **end for**
12: all_vals ← cat $\left(\{\text{flatten}(v) \mid v \in \text{gradients.values}()\}\right)$
13: cutoff ← quantile(all_vals, percent)                    ▷ e.g., 0.1 = bottom 10%
14: **return** { name ↦ (grad < cutoff) | (name, grad) ∈ gradients}

---

**Algorithm 2** SharpMinMax

---

**Require:** x_retain, y_retain, x_forget, y_forget, model, criterion, mask, alpha, optimizer_retain, optimizer_forget
1: $r\_loss1 \leftarrow \alpha \cdot \text{criterion}(\text{model}(\text{x\_retain}), \text{y\_retain})$
2: $r\_loss1.\text{backward}()$
3: optimizer_retain.first_step(zero_grad=True)                    ▷ SAM first step
4: $r\_loss2 \leftarrow \alpha \cdot \text{criterion}(\text{model}(\text{x\_retain}), \text{y\_retain})$
5: $r\_loss2.\text{backward}()$
6: **for all** (name, $p$) in model parameters **do**
7:     **if** $p$.grad **then**
8:         $p.\text{grad} \leftarrow p.\text{grad} \odot \left(1 - \text{mask[name]}\right)$            ▷ mask out forget grads
9:     **end if**
10: **end for**
11: optimizer_retain.second_step(zero_grad=True)                    ▷ sharp min
12: $f\_loss1 \leftarrow -(1-\alpha) \cdot \text{criterion}(\text{model}(\text{x\_forget}), \text{y\_forget})$
13: $f\_loss1.\text{backward}()$
14: optimizer_forget.first_step(zero_grad=True)                    ▷ SAM first step
15: $f\_loss2 \leftarrow -(1-\alpha) \cdot \text{criterion}(\text{model}(\text{x\_forget}), \text{y\_forget})$
16: $f\_loss2.\text{backward}()$
17: **for all** (name, $p$) in model parameters **do**
18:     **if** $p$.grad **then**
19:         $p.\text{grad} \leftarrow p.\text{grad} \odot \text{mask[name]}$            ▷ update forget params only
20:     **end if**
21: **end for**
22: optimizer_forget.second_step(zero_grad=True)                    ▷ sharp max

---

### E.4 EVALUATION DETAILS

**Membership inference attack.** We adopted a MIA based evaluation from Jia et al. (2023). We train a binary classifier using the retain set $\mathcal{R}$ and the test set $\mathcal{D}_{\text{test}}$ to distinguish whether a data sample was involved in the training stage, based on the softmaxed outputs from the unlearned model. Then, we feed the forget set $\mathcal{F}$ to the classifier to evaluate this unlearned model. We expect forget samples to be classified as "non-training" data, and we evaluate the unlearning effectiveness based on MIA correctness. A lower correctness (close to $0.5$) indicates difficulty to distinguish and thus better unlearning. This evaluation examines an unlearned model from a privacy perspective.

**Entanglement computation.** We compute both entanglement scores based on normalized embeddings of retain and forget sets from the penultimate layer of the model. We compute pair-wise entanglement between each retain and forget embedding, either globally or within a class. For variance-based entanglement $E_{\text{Var}}$, we directly follow Zhao et al. (2024) for implementation, and

then rescale the raw scores to $[0, 1]$ based on the value range across global and class-wise scores. For Wasserstein entanglement $E_{W_p}$, we randomly sample an equal number of embeddings from retain and forget embeddings and build two uniform proxy-distributions. We then use existing optimal transport library to compute the transport distance (cost), outputting entanglement scores as $1 - \text{distance}$. No clipping is needed as we observe all scores lie within $[0, 1]$.

# F  DETAILED EMPIRICAL RESULTS

## F.1  STATISTICAL SIGNIFICANCE

We demonstrate the statistical significance of our main empirical results by running each unlearning experiment three times with different seeds. In Fig. 5 and Fig. 4, we report the $95\%$ confidence intervals $(\mu \pm 2\sigma)$ of all unlearning methods on ImageNet and CIFAR-100, which correspond to Tab. 1 and Tab. 3. Each single bar represents the mean over runs and has the mean ToW scores marked on top of its error bar plotted by $\pm 2\sigma$. We observe that SAM consistently improves all unlearning methods with more noticeable results on CIFAR-100. For "All methods" subplots, we highlight the largest improvement by applying SAM to each $\mathcal{U}$. On CIFAR-100, we observe a general larger variance of SGD based unlearning, especially for SCRUB. Despite that $\mathcal{A}$=SAM 0.1 seems to provide a weaker pretrained model, Adaptive SAM settings can improve unlearning performance more steadily with lower variance, which demonstrate that SAM unlearning is more robust. Tab. 8 also records the means and variances of the "All methods" subplots for ImageNet and CIFAR-100. These additional insights further strengthen our findings.

Table 8: Verifying statistical significance $(\mu \pm \sigma)$ of main experiments on ImageNet and CIFAR-100. Given various pretrained model with different $\mathcal{A}$, we observe that SAM consistently improve base unlearn methods $\mathcal{U}$ with higher means across multiple seeds. Moreover, we observe generally more stable performance with SAM based on smaller variance on average.

| **ImageNet** | RL | | SalUn | | NG | | MinMax | |
| Method | SGD | ASAM 1.0 | SGD | ASAM 1.0 | SGD | ASAM 1.0 | SGD | ASAM 1.0 |
|---|---|---|---|---|---|---|---|---|
| $\mathcal{A}$=SGD | 82.9±0.3 | 83.9±0.2 | 70.6±0.1 | 71.0±0.1 | 83.5±0.3 | 84.8±0.0 | 80.2±0.1 | 87.9±0.0 |
| $\mathcal{A}$=ASAM 0.1 | 82.5±0.1 | 83.8±0.1 | 70.7±0.1 | 71.1±0.1 | 83.4±0.3 | 84.7±0.1 | 79.7±0.2 | 87.5±0.1 |
| $\mathcal{A}$=ASAM 1.0 | 83.2±0.4 | 83.8±0.2 | 71.1±0.0 | 71.2±0.0 | 84.1±0.0 | 84.6±0.2 | 80.1±0.2 | 88.0±0.1 |
| $\mathcal{A}$=SAM 0.1 | 82.9±0.2 | 83.7±0.3 | 71.2±0.0 | 71.4±0.1 | 83.6±0.1 | 84.4±0.1 | 79.9±0.1 | 87.8±0.1 |

| **CIFAR100** | L1 Sparse | | Scrub | | RL | | SalUn | | NG | |
| Method | SGD | ASAM 1.0 | SGD | ASAM 1.0 | SGD | ASAM 1.0 | SGD | ASAM 1.0 | SGD | ASAM 1.0 |
|---|---|---|---|---|---|---|---|---|---|---|
| $\mathcal{A}$=SGD | 62.1±1.4 | 67.3±0.1 | 56.5±14.1 | 73.6±0.4 | 74.2±1.0 | 77.2±0.2 | 76.1±1.5 | 83.8±0.9 | 82.8±1.1 | 84.0±0.9 |
| $\mathcal{A}$=ASAM 0.1 | 63.6±1.7 | 69.3±0.6 | 54.3±1.8 | 79.3±0.8 | 72.1±0.9 | 75.8±1.3 | 72.9±1.6 | 82.5±0.4 | 83.9±0.8 | 85.5±0.6 |
| $\mathcal{A}$=ASAM 1.0 | 64.2±0.7 | 68.7±1.7 | 58.4±10.5 | 72.0±2.1 | 75.7±1.5 | 80.3±1.2 | 79.0±0.3 | 83.3±0.2 | 80.2±0.5 | 83.9±0.2 |
| $\mathcal{A}$=SAM 0.1 | 64.9±1.3 | 68.3±0.6 | 41.1±1.7 | 49.7±16.6 | 74.2±0.7 | 80.3±0.9 | 79.4±1.0 | 83.6±0.6 | 71.3±1.8 | 78.7±0.5 |

## F.2  COMPLETE ACCURACIES

In Tab. 9, Tab. 10, and Tab. 11, we report complete results of retain, forget, and test accuracies for all unlearning experiments, which are used to compute ToW scores in Tab. 1 and Tab. 3. As we have mentioned in the main paper, we observe that SGD often achieves lower test accuracies, motivating us to rethink the overfitting under a sample-specific unlearning scheme.

# G  ADDITIONAL EXPERIMENTS

We provide additional experiments on CIFAR-10 and Tiny-ImageNet using randomly sampled forget set $\mathcal{F}_{\text{rand}}$. To diversify our experiment settings, we use ResNet-34 with ImageNet-pretrained weights for our learning and unlearning on Tiny-ImageNet. Similar to our main setup, we pretrain and retrain using the same settings, and we have summarized basic settings and baseline performance in Tab. 12. Since Tiny-ImageNet has 100K samples, we set $|\mathcal{F}_{\text{rand}}| = 6000$ for Tiny-ImageNet. Tab. 13 records detailed accuracies and ToW scores of various unlearning and pretraining settings.

## G.1  CIFAR-10

We summarize detailed unlearning settings on CIFAR-10. For L1-Sparse, we use unlearn lr= 0.02 and $\alpha = 1 \times 10^{-4}$. For SCRUB, we use unlearn lr= 0.004, msteps= 8, kd_T= 3.5, $\beta = 0.01$,

Table 9: Detailed accuracies of NegGrad on ImageNet and CIFAR-100.

| ImageNet | $\mathcal{A}$ =SGD | | | | $\mathcal{A}$ =ASAM 0.1 | | | | $\mathcal{A}$ =ASAM 1.0 | | | | $\mathcal{A}$ =SAM 0.1 | | | |
|---|---|---|---|---|---|---|---|---|---|---|---|---|---|---|---|---|
| **High Mem** | Retain | Forget | Test | ToW | Retain | Forget | Test | ToW | Retain | Forget | Test | ToW | Retain | Forget | Test | ToW |
| +SGD | 88.766 | 25.148 | 71.756 | 78.764 | 88.131 | 24.1 | 70.878 | 78.426 | 89.649 | 26.28 | 71.772 | 78.522 | 89.158 | 26.488 | 71.91 | 78.03 |
| +ASAM 0.1 | 89.487 | 26.407 | 72.08 | 78.52 | 88.640 | 24.77 | 70.988 | 78.366 | 89.767 | 26.542 | 72.236 | 78.762 | 89.816 | 27.422 | 72.328 | 78.083 |
| +ASAM 1.0 | 90.804 | 28.398 | 73.506 | 78.966 | 90.399 | 27.522 | 72.94 | 78.975 | 91.232 | 29.862 | 73.58 | 78.027 | 91.121 | 30.208 | 73.77 | 77.762 |
| +SAM 0.1 | 91.007 | 29.88 | 73.676 | 77.898 | 90.498 | 28.445 | 73.05 | 78.301 | 91.583 | 30.997 | 73.746 | 77.388 | 91.328 | 31.578 | 73.964 | 76.807 |
| **Mid Mem** | Retain | Forget | Test | ToW | Retain | Forget | Test | ToW | Retain | Forget | Test | ToW | Retain | Forget | Test | ToW |
| +SGD | 88.771 | 56.87 | 71.414 | 84.199 | 89.265 | 57.832 | 71.562 | 83.93 | 89.80 | 58.622 | 71.812 | 83.929 | 89.312 | 58.27 | 72.248 | 84.176 |
| +ASAM 0.1 | 89.56 | 58.502 | 72.154 | 84.113 | 89.276 | 57.698 | 71.576 | 84.07 | 90.087 | 59.08 | 72.378 | 84.267 | 89.945 | 59.263 | 72.482 | 84.062 |
| +ASAM 1.0 | 90.969 | 61.998 | 73.544 | 83.389 | 91.064 | 62.023 | 73.434 | 83.358 | 91.427 | 62.757 | 73.82 | 83.326 | 91.505 | 63.078 | 74.046 | 83.284 |
| +SAM 0.1 | 91.396 | 63.015 | 73.734 | 82.985 | 91.015 | 62.308 | 73.422 | 83.04 | 91.984 | 64.367 | 74.014 | 82.473 | 91.823 | 64.258 | 74.198 | 82.587 |
| **Low Mem** | Retain | Forget | Test | ToW | Retain | Forget | Test | ToW | Retain | Forget | Test | ToW | Retain | Forget | Test | ToW |
| +SGD | 87.775 | 99.617 | 71.942 | 88.515 | 86.592 | 99.505 | 71.042 | 86.651 | 88.847 | 99.663 | 72.41 | 89.947 | 87.847 | 99.625 | 72.228 | 88.839 |
| +ASAM 0.1 | 88.251 | 99.643 | 72.198 | 89.188 | 88.296 | 99.635 | 72.044 | 89.098 | 89.293 | 99.7 | 72.658 | 90.579 | 88.553 | 99.69 | 72.776 | 89.973 |
| +ASAM 1.0 | 89.903 | 99.818 | 73.844 | 92.174 | 89.704 | 99.808 | 73.69 | 91.843 | 90.432 | 99.79 | 73.896 | 92.772 | 90.042 | 99.813 | 74.166 | 92.617 |
| +SAM 0.1 | 90.234 | 99.822 | 74.21 | 92.841 | 89.553 | 99.817 | 73.728 | 91.722 | 90.815 | 99.827 | 74.228 | 93.429 | 90.184 | 99.825 | 74.254 | 92.829 |

| CIFAR100 | $\mathcal{A}$ =SGD | | | | $\mathcal{A}$ =ASAM 0.1 | | | | $\mathcal{A}$ =ASAM 1.0 | | | | $\mathcal{A}$ =SAM 0.1 | | | |
|---|---|---|---|---|---|---|---|---|---|---|---|---|---|---|---|---|
| **High Mem** | Retain | Forget | Test | ToW | Retain | Forget | Test | ToW | Retain | Forget | Test | ToW | Retain | Forget | Test | ToW |
| +SGD | 92.929 | 12.9 | 68.17 | 78.334 | 94.05 | 11.433 | 66.68 | 79.277 | 94.533 | 15.267 | 67.78 | 77.274 | 91.814 | 22.4 | 66.23 | 67.82 |
| +ASAM 0.1 | 93.736 | 13.467 | 67.71 | 78.131 | 94.852 | 11.633 | 67.32 | 80.336 | 94.633 | 15.333 | 67.82 | 77.331 | 93.674 | 22.9 | 67.94 | 70.054 |
| +ASAM 1.0 | 96.748 | 15.433 | 69.98 | 80.806 | 96.907 | 13.167 | 69.03 | 82.196 | 96.893 | 17.7 | 69.85 | 78.731 | 96.376 | 24.033 | 69.85 | 72.518 |
| +SAM 0.1 | 98.552 | 19 | 72.82 | 81.331 | 99.193 | 17.4 | 72.17 | 82.86 | 99.4 | 26.467 | 72.74 | 74.704 | 99.24 | 36.767 | 73.49 | 65.08 |
| **Mid Mem** | Retain | Forget | Test | ToW | Retain | Forget | Test | ToW | Retain | Forget | Test | ToW | Retain | Forget | Test | ToW |
| +SGD | 93.162 | 60.3 | 66.15 | 83.335 | 95.024 | 58.433 | 65.96 | 86.454 | 95.519 | 69.2 | 67.3 | 78.59 | 93.714 | 72.233 | 66.91 | 74.145 |
| +ASAM 0.1 | 94.055 | 62.633 | 66.97 | 82.846 | 95.005 | 58.133 | 66.85 | 87.539 | 95.524 | 68.133 | 66.75 | 79.074 | 93.838 | 72.367 | 66.95 | 74.158 |
| +ASAM 1.0 | 96.781 | 69.533 | 69.81 | 81.465 | 97.16 | 65.4 | 68.43 | 84.391 | 97.919 | 72.7 | 69.58 | 79.264 | 97.257 | 76.2 | 69.8 | 75.653 |
| +SAM 0.1 | 98.938 | 80.133 | 72.18 | 75.059 | 99.007 | 76.133 | 70.87 | 77.94 | 99.448 | 85.1 | 72.59 | 70.898 | 99.169 | 90.033 | 72.9 | 66.089 |
| **Low Mem** | Retain | Forget | Test | ToW | Retain | Forget | Test | ToW | Retain | Forget | Test | ToW | Retain | Forget | Test | ToW |
| +SGD | 91.086 | 97.767 | 65.67 | 83.718 | 95.312 | 98.267 | 67.18 | 88.637 | 93.117 | 98.5 | 66.17 | 85.443 | 85.307 | 96.933 | 62.63 | 76.374 |
| +ASAM 0.1 | 92.736 | 97.767 | 67.3 | 86.78 | 94.676 | 98.5 | 67 | 87.671 | 94.298 | 97.967 | 67.27 | 88.039 | 86.902 | 96.9 | 62.92 | 78.087 |
| +ASAM 1.0 | 92.824 | 97.8 | 67.53 | 87.052 | 96.267 | 99.1 | 68.94 | 90.502 | 97.883 | 99.533 | 70.59 | 93.249 | 93.517 | 98.7 | 67.35 | 86.759 |
| +SAM 0.1 | 97.89 | 99.333 | 71.31 | 94.151 | 98.712 | 99.7 | 70.89 | 94.179 | 99.26 | 99.667 | 72.06 | 95.898 | 98.695 | 99.633 | 71.75 | 95.078 |

Table 10: Detailed accuracies of Sharp MinMax on ImageNet and CIFAR-100.

| ImageNet | $\mathcal{A}$ =SGD | | | | $\mathcal{A}$ =ASAM 0.1 | | | | $\mathcal{A}$ =ASAM 1.0 | | | | $\mathcal{A}$ =SAM 0.1 | | | |
|---|---|---|---|---|---|---|---|---|---|---|---|---|---|---|---|---|
| **High Mem** | Retain | Forget | Test | ToW | Retain | Forget | Test | ToW | Retain | Forget | Test | ToW | Retain | Forget | Test | ToW |
| +SGD | 87.513 | 29.79 | 71.408 | 73.357 | 86.802 | 28.42 | 70.692 | 73.418 | 88.411 | 31.423 | 72.016 | 73.103 | 87.879 | 30.953 | 71.964 | 73.052 |
| +ASAM 0.1 | 79.741 | 10.555 | 66.334 | 78.066 | 80.84185 | 11.222 | 66.894 | 79.077 | 73.491 | 8.203 | 61.802 | 70.148 | 80.16741 | 11.032 | 66.828 | 78.529 |
| +ASAM 1.0 | 87.993 | 15.903 | 72.224 | 86.658 | 87.748 | 15.605 | 71.638 | 86.166 | 88.563 | 16.453 | 72.452 | 86.915 | 88.435 | 17.177 | 72.498 | 86.272 |
| +SAM 0.1 | 88.297 | 16.705 | 72.48 | 86.463 | 87.537 | 16.098 | 71.612 | 85.511 | 89.056 | 17.405 | 72.812 | 86.849 | 88.468 | 17.92 | 72.674 | 85.712 |
| **Mid Mem** | Retain | Forget | Test | ToW | Retain | Forget | Test | ToW | Retain | Forget | Test | ToW | Retain | Forget | Test | ToW |
| +SGD | 87.089 | 57.915 | 71.418 | 80.881 | 86.757 | 58.372 | 71.1 | | 87.217 | 59.095 | 71.734 | 81.105 | 87.461 | 59.677 | 71.848 | 80.913 |
| +ASAM 0.1 | 86.936 | 50.585 | 71.38 | 87.914 | 86.281 | 49.833 | 70.814 | 87.40 | 87.561 | 51.3 | 71.528 | 88.039 | 87.529 | 52.043 | 71.84 | 87.642 |
| +ASAM 1.0 | 88.679 | 54.642 | 72.834 | 87.345 | 88.588 | 54.548 | 72.666 | 87.192 | 89.12 | 55.377 | 73.018 | 87.27 | 89.092 | 55.733 | 73.192 | 87.076 |
| +SAM 0.1 | 89.141 | 56.215 | 73.268 | 86.755 | 88.642 | 55.303 | 72.74 | 86.635 | 89.492 | 56.813 | 73.47 | 86.722 | 89.758 | 57.657 | 73.792 | 86.486 |
| **Low Mem** | Retain | Forget | Test | ToW | Retain | Forget | Test | ToW | Retain | Forget | Test | ToW | Retain | Forget | Test | ToW |
| +SGD | 85.798 | 99.61 | 71.644 | 86.334 | 84.348 | 99.482 | 70.894 | 84.378 | 85.863 | 99.568 | 71.61 | 86.402 | 85.098 | 99.57 | 71.45 | 85.517 |
| +ASAM 0.1 | 86.399 | 99.565 | 72.07 | 87.338 | 86.236 | 99.562 | 71.814 | 86.953 | 86.644 | 99.627 | 72.104 | 87.554 | 85.894 | 99.593 | 71.898 | 86.668 |
| +ASAM 1.0 | 87.766 | 99.768 | 73.392 | 89.694 | 87.366 | 99.772 | 73.216 | 89.138 | 88.159 | 99.722 | 73.412 | 90.142 | 87.837 | 99.765 | 73.718 | 90.064 |
| +SAM 0.1 | 87.836 | 99.777 | 73.666 | 90.005 | 87.745 | 99.76 | 73.58 | 89.852 | 88.706 | 99.783 | 73.94 | 91.111 | 87.974 | 99.792 | 73.752 | 90.207 |

| CIFAR100 | $\mathcal{A}$ =SGD | | | | $\mathcal{A}$ =ASAM 0.1 | | | | $\mathcal{A}$ =ASAM 1.0 | | | | $\mathcal{A}$ =SAM 0.1 | | | |
|---|---|---|---|---|---|---|---|---|---|---|---|---|---|---|---|---|
| **High Mem** | Retain | Forget | Test | ToW | Retain | Forget | Test | ToW | Retain | Forget | Test | ToW | Retain | Forget | Test | ToW |
| +SGD | 92.298 | 20.8 | 67.86 | 70.767 | 95.098 | 22.167 | 68.42 | 72.137 | 92.564 | 25.4 | 66.35 | 65.925 | 87.195 | 25.233 | 63.77 | 60.478 |
| +ASAM 0.1 | 89.574 | 6.133 | 65.57 | 78.895 | 93.819 | 5.333 | 67.37 | 84.968 | 92.095 | 6.3 | 66.52 | 81.825 | 86.969 | 9.233 | 64.03 | 72.897 |
| +ASAM 1.0 | 92.121 | 6.467 | 67.15 | 82.27 | 88.976 | 5.067 | 63.68 | 77.576 | 93.895 | 6.567 | 67.98 | 84.521 | 90.448 | 10.7 | 65.71 | 76.037 |
| +SAM 0.1 | 97.383 | 7.1 | 71.61 | 90.578 | 98.183 | 6.133 | 71.04 | 91.695 | 97.619 | 8.467 | 70.7 | 88.664 | 98.198 | 14.167 | 72.26 | 85.195 |
| **Mid Mem** | Retain | Forget | Test | ToW | Retain | Forget | Test | ToW | Retain | Forget | Test | ToW | Retain | Forget | Test | ToW |
| +SGD | 91.433 | 66 | 65.79 | 76.692 | 91.633 | 63.367 | 64.39 | 77.864 | 92.11 | 69.4 | 65.96 | 74.526 | 85.714 | 62.6 | 62.55 | 71.931 |
| +ASAM 0.1 | 91.16 | 42.7 | 65.88 | 96.027 | 91.4 | 40.233 | 64.11 | 96.451 | 95.26 | 51.2 | 66.74 | 93.786 | 88.074 | 55.867 | 63.61 | 80.104 |
| +ASAM 1.0 | 92.586 | 46.9 | 66.81 | 94.913 | 94.074 | 43.133 | 66.53 | 99.422 | 89.36 | 47.433 | 63.35 | 87.761 | 93.119 | 60.067 | 66.3 | 83.633 |
| +SAM 0.1 | 97.433 | 60.867 | 70.73 | 90.96 | 97.874 | 55.033 | 69.39 | 95.543 | 98.6 | 64.333 | 70.62 | 88.646 | 98.824 | 76.433 | 71.84 | 78.286 |
| **Low Mem** | Retain | Forget | Test | ToW | Retain | Forget | Test | ToW | Retain | Forget | Test | ToW | Retain | Forget | Test | ToW |
| +SGD | 89.579 | 97.6 | 66.09 | 82.853 | 89.781 | 97.1 | 64.36 | 81.847 | 88.605 | 97.833 | 64.28 | 80.127 | 81.488 | 94.467 | 61.63 | 73.843 |
| +ASAM 0.1 | 89.026 | 95.067 | 65.12 | 83.473 | 89.874 | 96.033 | 64.47 | 82.883 | 93.748 | 97.167 | 66.17 | 87.151 | 92.967 | 97.433 | 66.65 | 86.659 |
| +ASAM 1.0 | 91.931 | 96.567 | 66.74 | 86.504 | 92.819 | 97.467 | 66.02 | 85.894 | 91.131 | 96.2 | 64.97 | 84.381 | 85.014 | 95.3 | 62.79 | 77.461 |
| +SAM 0.1 | 96.129 | 98.033 | 70.13 | 92.494 | 96.829 | 98.7 | 69.06 | 91.508 | 97.624 | 98.567 | 69.85 | 93.163 | 96.652 | 99.033 | 68.98 | 90.963 |

and $\gamma = 0.99$. For RL and SalUn, we use unlearn lr= 0.08, and use $50\%$ model parameters for SalUn. For NegGrad and Sharp MinMax, we use unlearn lr= 0.02 and $\alpha = 0.99$, and use $30\%$ model parameters for unlearning on $\mathcal{F}$ and the rest for learning on $\mathcal{R}$.

From the results in Tab. 12, we observe consistent improvement by using SAM except only two cases for RL and SalUn with $\mathcal{A} = SGD$. Surprisingly, Sharp MinMax is not the best algorithm on CIFAR-10. By the nature of its design to overfit to forget signals deliberately, we hypothesize that this approach might be aggressive for small-scale unlearning. We again observe SCRUB to be an

Table 11: Detailed accuracies of previous methods on ImageNet and CIFAR-100.

| ImageNet | $\mathcal{A}$ =SGD | | | | $\mathcal{A}$ =ASAM 0.1 | | | | $\mathcal{A}$ =ASAM 1.0 | | | | $\mathcal{A}$ =SAM 0.1 | | | |
|---|---|---|---|---|---|---|---|---|---|---|---|---|---|---|---|---|
| **High Mem** | Retain | Forget | Test | ToW | Retain | Forget | Test | ToW | Retain | Forget | Test | ToW | Retain | Forget | Test | ToW |
| RL | 88.536 | 29.857 | 72.02 | 74.598 | 88.663 | 29.622 | 71.95 | 74.857 | 88.975 | 30.59 | 72.04 | 74.317 | 89.429 | 31.74 | 72.572 | 74.055 |
| +ASAM 1.0 | 90.874 | 33.395 | 74.234 | 74.951 | 90.615 | 32.668 | 73.972 | 75.221 | 91.14 | 34.745 | 74.298 | 73.95 | 91.155 | 35.332 | 74.522 | 73.579 |
| SalUn | 93.248 | 67.118 | 75.04 | 44.981 | 93.016 | 65.807 | 74.976 | 46.104 | 93.124 | 66.372 | 75.418 | 45.814 | 92.911 | 66.333 | 75.982 | 46.006 |
| +ASAM 1.0 | 93.123 | 66.217 | 75.496 | 45.998 | 92.963 | 65.058 | 75.28 | 46.938 | 93.134 | 66.472 | 75.712 | 45.856 | 92.855 | 66.032 | 76.172 | 46.358 |
| **Mid Mem** | Retain | Forget | Test | ToW | Retain | Forget | Test | ToW | Retain | Forget | Test | ToW | Retain | Forget | Test | ToW |
| RL | 88.785 | 54.653 | 71.916 | 86.617 | 88.067 | 53.387 | 71.258 | 86.462 | 89.754 | 56.17 | 72.634 | 86.813 | 88.609 | 54.608 | 72.168 | 86.715 |
| +ASAM 1.0 | 90.597 | 59.53 | 73.836 | 85.581 | 90.457 | 59.337 | 73.654 | 85.473 | 90.993 | 60.35 | 74.078 | 85.393 | 90.902 | 60.402 | 74.348 | 85.494 |
| SalUn | 93.174 | 77.258 | 74.816 | 71.839 | 93.072 | 77.222 | 74.728 | 71.735 | 93.078 | 77.118 | 75.382 | 72.308 | 92.825 | 77.167 | 75.868 | 72.419 |
| +ASAM 1.0 | 93.098 | 77.983 | 75.47 | 71.554 | 92.969 | 77.947 | 75.154 | 71.268 | 93.143 | 78.058 | 75.724 | 71.695 | 92.797 | 77.805 | 76.222 | 72.034 |
| **Low Mem** | Retain | Forget | Test | ToW | Retain | Forget | Test | ToW | Retain | Forget | Test | ToW | Retain | Forget | Test | ToW |
| RL | 85.745 | 98.603 | 71.162 | 86.714 | 85.451 | 98.463 | 70.768 | 86.192 | 86.472 | 98.74 | 71.522 | 87.63 | 86.865 | 98.95 | 72.36 | 88.594 |
| +ASAM 1.0 | 88.517 | 99.408 | 73.728 | 91.069 | 88.218 | 99.377 | 73.32 | 90.425 | 88.985 | 99.457 | 73.758 | 91.516 | 88.963 | 99.507 | 74.072 | 91.74 |
| SalUn | 91.991 | 99.778 | 74.612 | 95.008 | 91.743 | 99.77 | 74.488 | 94.652 | 91.696 | 99.818 | 75.074 | 95.116 | 91.412 | 99.85 | 75.514 | 95.218 |
| +ASAM 1.0 | 92.095 | 99.85 | 75.224 | 95.628 | 91.882 | 99.818 | 74.992 | 95.224 | 91.967 | 99.857 | 75.676 | 95.924 | 91.579 | 99.873 | 75.964 | 95.791 |

| CIFAR100 | $\mathcal{A}$ =SGD | | | | $\mathcal{A}$ =ASAM 0.1 | | | | $\mathcal{A}$ =ASAM 1.0 | | | | $\mathcal{A}$ =SAM 0.1 | | | |
|---|---|---|---|---|---|---|---|---|---|---|---|---|---|---|---|---|
| **High Mem** | Retain | Forget | Test | ToW | Retain | Forget | Test | ToW | Retain | Forget | Test | ToW | Retain | Forget | Test | ToW |
| L1-Sparse | 74.76 | 5.267 | 61.49 | 63.448 | 75.426 | 5.067 | 60.89 | 63.699 | 73.969 | 6.167 | 60.17 | 61.252 | 77.429 | 7.133 | 62.56 | 65.258 |
| +ASAM 1.0 | 77.86 | 5.733 | 62.99 | 66.903 | 77.648 | 5.7 | 62.29 | 66.213 | 77.126 | 6.367 | 62.02 | 65.117 | 75.583 | 6.2 | 60.83 | 63.051 |
| SCRUB | 99.867 | 44.567 | 74.52 | 58.418 | 99.793 | 35.267 | 73.85 | 67.163 | 99.902 | 45.233 | 74.59 | 57.816 | 99.971 | 60.7 | 76.47 | 43.246 |
| +ASAM 1.0 | 99.962 | 53.533 | 76.06 | 50.313 | 99.955 | 42.633 | 74.72 | 60.515 | 99.969 | 55.3 | 76.14 | 48.569 | 99.971 | 85.567 | 77.23 | 18.137 |
| RL | 82.681 | 9.233 | 62.95 | 68.464 | 79.229 | 8.367 | 60.7 | 64.518 | 82.99 | 10.933 | 61.92 | 66.689 | 81.069 | 10.833 | 60.82 | 64.391 |
| +ASAM 1.0 | 84.012 | 9.7 | 63.88 | 69.952 | 81.519 | 8.4 | 61.41 | 66.909 | 86.195 | 12 | 63.53 | 69.73 | 89.324 | 13.7 | 65.99 | 72.884 |
| SalUn | 89.624 | 16.567 | 64.88 | 69.926 | 86.298 | 15.467 | 62.71 | 67.355 | 91.207 | 20.7 | 64.33 | 66.541 | 90.593 | 18.533 | 65.65 | 69.671 |
| +ASAM 1.0 | 94.557 | 20.9 | 68.96 | 73.268 | 92.326 | 18.3 | 65.94 | 71.426 | 94.519 | 25.033 | 66.46 | 67.715 | 95.636 | 24.367 | 68.89 | 70.933 |
| **Mid Mem** | Retain | Forget | Test | ToW | Retain | Forget | Test | ToW | Retain | Forget | Test | ToW | Retain | Forget | Test | ToW |
| L1-Sparse | 67.864 | 36.8 | 57.97 | 68.686 | 71.305 | 38.633 | 59.98 | 72.775 | 68.264 | 37.933 | 57.67 | 68.197 | 71.495 | 39.967 | 59.73 | 71.941 |
| +ASAM 1.0 | 74.148 | 41.5 | 61.96 | 75.554 | 75.836 | 42.7 | 62.7 | 77.119 | 74.267 | 43.967 | 61.59 | 73.754 | 73.857 | 40.667 | 60.52 | 74.556 |
| SCRUB | 99.864 | 81.4 | 74.29 | 76.125 | 99.876 | 76.9 | 72.37 | 79.09 | 99.91 | 83.867 | 73.59 | 73.176 | 99.974 | 90.167 | 75.78 | 68.433 |
| +ASAM 1.0 | 99.974 | 85.133 | 75.51 | 73.353 | 99.969 | 77.367 | 74.24 | 80.204 | 99.981 | 85.433 | 75.56 | 73.09 | 99.974 | 97.667 | 77.13 | 61.618 |
| RL | 79.262 | 37.067 | 62.53 | 84.395 | 75.757 | 31.733 | 58.31 | 80.215 | 81.955 | 36.433 | 61.21 | 86.411 | 81.905 | 38.033 | 61.48 | 85.481 |
| +ASAM 1.0 | 81.688 | 38.7 | 63.54 | 86.779 | 81.686 | 37.333 | 62.3 | 86.557 | 85.674 | 38.7 | 63.65 | 91.124 | 84.914 | 40.167 | 63.08 | 88.633 |
| SalUn | 82.383 | 40.733 | 60.46 | 83.056 | 82.4 | 40.9 | 60.9 | 83.377 | 89.581 | 45.333 | 63.46 | 89.768 | 90.205 | 46.867 | 64.8 | 90.495 |
| +ASAM 1.0 | 91.579 | 48.167 | 66.23 | 92.225 | 88.71 | 45.833 | 64.15 | 89.182 | 94.217 | 50.5 | 66.77 | 93.401 | 94.2 | 52.333 | 67.91 | 92.914 |
| **Low Mem** | Retain | Forget | Test | ToW | Retain | Forget | Test | ToW | Retain | Forget | Test | ToW | Retain | Forget | Test | ToW |
| L1-Sparse | 62.41 | 91.367 | 55.39 | 53.991 | 68.667 | 96 | 60.25 | 60.34 | 68.421 | 94.2 | 60.67 | 61.47 | 67.229 | 94.967 | 59.33 | 59.014 |
| +ASAM 1.0 | 66.95 | 94.5 | 59.24 | 58.967 | 73.457 | 96.4 | 63.4 | 66.697 | 70.207 | 96.1 | 61.46 | 62.517 | 72.355 | 96.2 | 62.46 | 65.117 |
| SCRUB | 17.81 | 32.6 | 18.33 | 12.708 | 15.698 | 28.367 | 15.87 | 10.823 | 66.324 | 90.167 | 56.04 | 58.483 | 23.038 | 43.7 | 23.95 | 17.368 |
| +ASAM 1.0 | 99.683 | 99.9 | 73.61 | 97.631 | 99.869 | 99.833 | 73.24 | 97.508 | 99.64 | 99.8 | 73.7 | 97.776 | 99.729 | 99.8 | 73.77 | 97.933 |
| RL | 76.376 | 89.233 | 61.34 | 72.4 | 73.283 | 86.5 | 59.57 | 69.711 | 73.495 | 84.2 | 57.63 | 69.677 | 76.79 | 91.733 | 60.62 | 70.55 |
| +ASAM 1.0 | 78.286 | 90.533 | 62.59 | 74.409 | 73.881 | 87.3 | 59.08 | 69.375 | 82.695 | 89.333 | 63.53 | 80.321 | 83.483 | 94.167 | 64.12 | 78.066 |
| SalUn | 78.867 | 92.667 | 60.5 | 71.73 | 77.748 | 88.833 | 59.01 | 71.95 | 83.921 | 91.2 | 62.39 | 79.095 | 82.221 | 93.133 | 61.44 | 75.281 |
| +ASAM 1.0 | 91.205 | 95.5 | 68.28 | 88.175 | 90.043 | 93.367 | 65.47 | 86.13 | 93.812 | 95.8 | 67.11 | 89.289 | 91.848 | 95.933 | 66.24 | 86.477 |

Table 12: Differed settings of pretrained models and their test accuracies using different $\mathcal{A}$, as well as performance of retrained models w.r.t $\mathcal{F}_{\text{rand}}$ for CIFAR-10 and Tiny-ImageNet.

| Dataset, Model | lr+warmup | Batch $B$ | Epoch $T$ | W. Decay | SGD | ASAM 0.1 | ASAM 1.0 | SAM 0.1 | Retain | Forget | Test |
|---|---|---|---|---|---|---|---|---|---|---|---|
| CIFAR10, Res18 | 0.1+0 | 256 | 50 | 5e-4 | 93.02 | 93.26 | 93.7 | 93.38 | 99.943 | 92.567 | 92.49 |
| TinyImgNt, Res34 | 0.003+0 | 256 | 200 | 1e-3 | 62.1 | 62.77 | 62.74 | 63.87 | 99.985 | 59.383 | 61.69 |

unstable algorithm which collapses when unlearning with SGD given $\mathcal{A} = SAM0.1$, while SAM helps reduce variance and stabilizes SCRUB unlearning given various pretrained models.

## G.2 TINY-IMAGENET

We summarize detailed unlearning settings on Tiny-ImageNet. For L1-Sparse, we use unlearn lr= 0.002 and $\alpha = 1 \times 10^{-4}$. For SCRUB, we use unlearn lr= 0.002, msteps= 8, kd_T= 3.5, $\beta = 0.01$, and $\gamma = 0.99$. For RL and SalUn, we use unlearn lr= 0.015, and use $30\%$ model parameters for SalUn. For NegGrad and Sharp MinMax, we use unlearn lr= 0.005 and $\alpha = 0.99$, and use $10\%$ model parameters for unlearning on $\mathcal{F}$ and the rest for learning on $\mathcal{R}$.

From the results in Tab. 12, we observe consistent improvement by using SAM except few cases. SCRUB performs more steadily than on CIFAR-10. While RL and SalUn perform well on other datasets, they do not appear to be effective on Tiny-ImageNet.

Table 13: Detailed accuracies of previous methods on Tiny-ImageNet and CIFAR-10.

| **TinyImageNet** | | $\mathcal{A}$ =SGD | | | | $\mathcal{A}$ =ASAM 0.1 | | | | $\mathcal{A}$ =ASAM 1.0 | | | | $\mathcal{A}$ =SAM 0.1 | | |
|---|---|---|---|---|---|---|---|---|---|---|---|---|---|---|---|---|
| **Random $\mathcal{F}_{rand}$** | Retain | Forget | Test | ToW | Retain | Forget | Test | ToW | Retain | Forget | Test | ToW | Retain | Forget | Test | ToW |
| L1-Sparse | 79.247 | 52.233 | 49.61 | 74.669 | 82.722 | 54.217 | 50.81 | 77.545 | 84.63 | 59.583 | 53.01 | 77.143 | 76.005 | 63.017 | 49.56 | 64.372 |
| +ASAM 1.0 | 89.379 | 59.5 | 54.37 | **82.753** | 90.81 | 60.933 | 54.35 | **82.853** | 92.005 | 63.517 | 53.7 | **81.168** | 94.674 | 74.333 | 55.25 | **75.347** |
| SCRUB | 92.112 | 58.117 | 53.65 | 85.793 | 94.315 | 60.75 | 54.58 | 86.425 | 96.268 | 66.5 | 55.01 | 83.457 | 99.801 | 88.233 | 58.99 | 69.101 |
| +ASAM 1.0 | 97.965 | 57.717 | 56.94 | **94.881** | 98.941 | 61.833 | 58.13 | **93.095** | 99.521 | 68.333 | 57.66 | **86.975** | 99.962 | 97.267 | 61.05 | 61.704 |
| RL | 64.504 | 63.233 | 46.59 | 52.668 | 67.506 | 66.433 | 47.49 | 53.849 | 70.309 | 69.883 | 48.16 | 54.424 | 75.016 | 73.5 | 49.21 | 56.397 |
| +ASAM 1.0 | 69.356 | 68.733 | 49.22 | **55.043** | 73.517 | 72.033 | 50.97 | **57.345** | 75.88 | 75.617 | 50.38 | 56.384 | 81.006 | 79.683 | 50.94 | **57.632** |
| SalUn | 69.39 | 68.45 | 50 | 55.735 | 70.087 | 68.767 | 49.54 | 55.806 | 73.207 | 71.783 | 50.12 | 56.721 | 82.877 | 81.467 | 53.36 | **59.206** |
| +ASAM 1.0 | 75.013 | 74.333 | 52.65 | **58.042** | 77.101 | 75.917 | 53.16 | **58.876** | 81.039 | 79.233 | 52.89 | **59.248** | 88.021 | 87.417 | 54.81 | 58.998 |
| NegGrad | 84.286 | 47.867 | 50.51 | 83.499 | 87.031 | 48.467 | 51.45 | 86.662 | 86.575 | 52.2 | 51.28 | 83.148 | 99.979 | 99.167 | 62.51 | 60.706 |
| +ASAM 1.0 | 90.907 | 50.45 | 54.47 | **91.894** | 93.681 | 51.35 | 53.66 | **93.094** | 96.343 | 54.167 | 54.31 | **93.902** | 98.031 | 62.767 | 55.21 | **88.59** |
| MinMax | 81.8 | 52.833 | 51.14 | 77.977 | 82.115 | 54.017 | 50.91 | 77.209 | 81.418 | 55.433 | 50.32 | 75.025 | 68.67 | 54.217 | 46.99 | 61.615 |
| +ASAM 1.0 | 87.654 | 43.183 | 53.4 | **93.426** | 88.273 | 43.083 | 52.86 | **93.613** | 91.947 | 43.6 | 53.37 | **97.617** | 94.517 | 48.5 | 53.72 | **96.466** |
| **CIFAR10** | | $\mathcal{A}$ =SGD | | | | $\mathcal{A}$ =ASAM 0.1 | | | | $\mathcal{A}$ =ASAM 1.0 | | | | $\mathcal{A}$ =SAM 0.1 | | |
| **Random $\mathcal{F}_{rand}$** | Retain | Forget | Test | ToW | Retain | Forget | Test | ToW | Retain | Forget | Test | ToW | Retain | Forget | Test | ToW |
| L1-Sparse | 86.467 | 82.967 | 82.25 | 85.12 | 89.06 | 85.567 | 84.45 | 87.688 | 86.683 | 83.467 | 82.11 | 84.811 | 90.462 | 87.133 | 84.82 | 87.144 |
| +ASAM 1.0 | 91.438 | 88.333 | 87.23 | **90.352** | 91.674 | 87.767 | 87.24 | **91.087** | 90.938 | 88.7 | 86.94 | **89.268** | 90.886 | 88.633 | 86.43 | **88.792** |
| SCRUB | 90.767 | 86.033 | 86.27 | 90.739 | 68.205 | 67.367 | 66.75 | 63.466 | 80.193 | 78.933 | 77.97 | 77.95 | 15.11 | 14.2 | 15 | 6.089 |
| +ASAM 1.0 | 99.6 | 95.167 | 92.65 | **97.2** | 99.621 | 96.5 | 93.15 | **96.39** | 99.807 | 98.2 | 93.38 | **95.078** | 99.631 | 98.467 | 93.16 | **94.435** |
| RL | 92.774 | 86.6 | 87.22 | **93.186** | 90.569 | 84.2 | 85.17 | 91.02 | 91.445 | 84.133 | 85.81 | 92.591 | 88.736 | 82.533 | 84.12 | 89.524 |
| +ASAM 1.0 | 93.295 | 87.733 | 87.66 | 93.138 | 93.262 | 87.233 | 88.31 | **94.187** | 95.098 | 89.033 | 89.44 | **95.512** | 92.588 | 86.567 | 87.4 | **93.206** |
| SalUn | 96.94 | 88.8 | 89.95 | **98.095** | 95.726 | 87.6 | 89.02 | 97.052 | 95.99 | 88.733 | 89.35 | 96.598 | 96.612 | 89.867 | 89.86 | 96.668 |
| +ASAM 1.0 | 97.771 | 91.8 | 90.55 | 96.666 | 98.24 | 91.867 | 91.41 | **97.917** | 98.029 | 91.6 | 91.2 | **97.757** | 98.055 | 92.833 | 91.37 | **96.755** |
| NegGrad | 97.724 | 93.933 | 90.46 | 94.487 | 98.35 | 94.967 | 91.33 | 94.931 | 98.024 | 94.267 | 90.92 | 94.9 | 96.405 | 93.4 | 89.72 | 93.009 |
| +ASAM 1.0 | 99.074 | 95.8 | 92.39 | **95.83** | 99.248 | 96.133 | 92.04 | **95.332** | 99.219 | 96.2 | 92.42 | **95.602** | 98.579 | 94.767 | 91.97 | **95.964** |
| MinMax | 96.85 | 94.133 | 90.29 | 93.291 | 97.652 | 94.933 | 90.6 | 93.594 | 97.881 | 95.1 | 90.5 | 93.558 | 96.498 | 93.533 | 90.22 | 93.451 |
| +ASAM 1.0 | 98.781 | 94.133 | 91.82 | **96.638** | 98.602 | 94 | 91.79 | **96.565** | 98.755 | 94.4 | 91.65 | **96.186** | 97.981 | 93.367 | 91.17 | **95.97** |

## G.3 UNLEARNING WITH STRUCTURED NOISE

We consider a noisy unlearning case where only a corrupted version of $\mathcal{S}$ is available, following corruptions in ImageNet-C (Hendrycks & Dietterich, 2019) to apply glass blur and snow effect to CIFAR-100 with medium severity for additional empirical verification, and report ToWs in Tab. 14: We observe that SAM continues to improve base unlearning methods with even more clear margins.

Table 14: Unlearning with ImageNet-C corruptions on CIFAR-100.

| **Glass Blur** | $\mathcal{A}$=SGD | | | | $\mathcal{A}$=ASAM | | | |
|---|---|---|---|---|---|---|---|---|
| Method | High | Mid | Low | AVG | High | Mid | Low | AVG |
| NG | 67.760 | 78.824 | 75.931 | 74.172 | 76.152 | 85.534 | 82.556 | 81.414 |
| +ASAM | 73.565 | 80.253 | 84.086 | **79.301** | 74.993 | 86.567 | 86.296 | **82.619** |
| SharpMinMax | 66.110 | 76.852 | 73.387 | 72.116 | 66.837 | 79.023 | 78.435 | 74.765 |
| +ASAM | 75.327 | 89.859 | 79.104 | **81.430** | 74.089 | 92.737 | 84.921 | **83.916** |
| **Snow** | $\mathcal{A}$=SGD | | | | $\mathcal{A}$=ASAM | | | |
| Method | High | Mid | Low | AVG | High | Mid | Low | AVG |
| NG | 77.394 | 83.328 | 83.196 | 81.306 | 75.041 | 86.424 | 86.838 | 82.768 |
| +ASAM | 76.759 | 84.168 | 86.053 | **82.327** | 76.520 | 83.774 | 89.343 | **83.212** |
| SharpMinMax | 70.880 | 78.806 | 77.652 | 75.779 | 69.650 | 79.139 | 81.344 | 76.711 |
| +ASAM | 77.188 | 90.997 | 83.933 | **84.039** | 80.533 | 93.383 | 87.779 | **87.232** |

This is because that structured noise applying to the images affects the dataset's signal and noise vectors ($\varphi$ and $\xi_i$), causing a corrupted dataset with worse initial signal-noise ratio, but it does not affect update dynamics and the gained results under our theoretical framework, as corrupted images are still visually recognizable, and SGD still overfits more to the added noise.

## G.4 SAM WITH ADAM AND VIT

We also verify that our observations generalize to different base optimizers and architectures. We experiment CIFAR-100 unlearning using ViT-Small (Dosovitskiy et al., 2020) and AdamW (Loshchilov & Hutter, 2017), and summarize our priliminary results in Tab. 15. For pretraining, we use AdamW with starting lr 0.0001, weight decay 0.05, and set patch size to 4 for ViT-Small on CIFAR-100. Other experiment settings are unchanged. For unlearning, we have unlearn lr 0.0006 for NegGrad and for Sharp MinMax. Adam demands much smaller lr than SGD and

Table 15: Unlearning with ViT-Small and AdamW on CIFAR-100.

| | $\mathcal{A}$=SGD | | | | $\mathcal{A}$=ASAM | | | |
| | High | Mid | Low | AVG | High | Mid | Low | AVG |
|---|---|---|---|---|---|---|---|---|
| **NG** | 80.445 | 82.854 | 84.385 | 82.561 | 78.750 | 82.223 | 86.767 | 82.580 |
| **+ASAM** | 82.880 | 83.084 | 83.402 | **83.122** | 82.839 | 81.354 | 87.507 | **83.900** |
| **SharpMinMax** | 14.794 | 42.055 | 95.222 | 50.690 | 14.279 | 42.017 | 94.833 | 50.376 |
| **+ASAM** | 76.343 | 95.573 | 103.372 | **91.763** | 76.664 | 93.966 | 105.868 | **92.166** |

is more sensitive to unlearn lr tuning. ViTs perform worse than ResNets on smaller datasets (test accuracies of pretrained models are 57%).

## G.5 RELEARNING ATTACKS

We present relearning attack experiments in Tab. 16 to demonstrate SAM's unlearning robustness below. We take the unlearned models to relearn the whole $\mathcal{F}$ for one epoch with a small relearning lr, and measure the increase in forget accuracies. Reported are the averaged increase across $\mathcal{F}_{high}, \mathcal{F}_{mid}, \mathcal{F}_{low}$. We observe that SAM enhanced $\mathcal{U}$ are more resilient to relearning attacks with smaller increases. We note that these experiments highlight the robustness of our approach and hope that this encourages future works for deeper investigation into the role of loss landscape geometry for robust unlearning.

Table 16: Average increase of forget accuracies after relearning 1 epoch on $\mathcal{F}$ on CIFAR-100 and ResNet50. We observe that SAM enhanced unlearning is consistently more resilient to relearning attacks with less increase on forget accuracy.

| | Relearn lr=0.002 | | Relearn lr=0.003 | | Relearn lr=0.004 | |
| | $\mathcal{A}$=SGD | $\mathcal{A}$=ASAM | $\mathcal{A}$=SGD | $\mathcal{A}$=ASAM | $\mathcal{A}$=SGD | $\mathcal{A}$=ASAM |
|---|---|---|---|---|---|---|
| **NG** | 8.644 | 10.333 | 11.167 | 13.256 | 12.789 | 14.7 |
| **+ASAM** | **8.533** | **9.289** | **11.033** | **11.533** | 13.022 | **13.389** |
| **SharpMinMax** | 13.1 | 15.067 | 15.589 | 17.5 | 16.144 | 18.511 |
| **+ASAM** | **8.333** | **8.8** | **10.667** | **11.2** | **12.711** | **12.667** |
| **RL** | 7.122 | 8.556 | 8.5 | 9.589 | 9.622 | 10.989 |
| **+ASAM** | **6.222** | **7.378** | **7.444** | **8.489** | **8.367** | **9.467** |

## G.6 RUNTIME AND EFFICIENT SAMS

We implement momentum SAM (MSAM) for unlearning on CIFAR-100. As shown in Tab. 17, MSAM not only outperforms vanilla SAM by much less computation overhead but can also outperform by average ToWs for some unlearning methods. This is plausible, as recent efficient SAMs reduce computation with more informative perturbation directions than stochastic by momentum buffer, sparsity, prior gradients, sharpness-sensitive data, etc. But there is no clear theoretical justification of MSAM rather than trying to stabilize the noise and reduce overhead of SAM. This warrants a deeper study beyond our scope – our focus is to show superiority of loss landscape based methods for unlearning without worrying about speed (just like the original SAM paper), and we leave deeper theoretical/algorithmic improvements and empirical evaluations for speedups for future work. We notice that while outperforming SGD with much less computation overhead than vanilla SAM, MSAM does not outperform ASAM on SharpMinMax and SalUn. As we also observe that MSAM behaves differently from ASAM on different forget sets, further and deeper investigation is needed to study the interactions between MSAM and weight masking to improve the performance. Our results have effectively demonstrated an example of a faster SAM variant that predictably benefits unlearning with less overhead.

## G.7 KLOM SCORES

We follow (Georgiev et al., 2024) to compute KLoM of NegGrad with SGD and SAM and report the KL measures on $\mathcal{F}, \mathcal{R}, \mathcal{D}_{test}$ across $\mathcal{F}$ of different difficulties, report means and 95%-percentiles in Tab. 18. Given a pretrained model, we observe that SAM in unlearning also helps close the gap

Table 17: ToWs of MSAM across different $\mathcal{F}$ in addition to reported results of baseline and SAM-enhanced unlearning on CIFAR-100. We observe that MSAM not only costs much less computation overhead than SAM but can also outperform by ToW for some settings, since it leverages a smarter perturbation based on momentum buffer.

| | $\mathcal{A}$=SGD | | | | $\mathcal{A}$=ASAM 1.0 | | | | Runtime |
|---|---|---|---|---|---|---|---|---|---|
| | $\mathcal{F}_{\text{high}}$ | $\mathcal{F}_{\text{mid}}$ | $\mathcal{F}_{\text{low}}$ | AVG | $\mathcal{F}_{\text{high}}$ | $\mathcal{F}_{\text{mid}}$ | $\mathcal{F}_{\text{low}}$ | AVG | |
| L1-Sparse | 63.448 | 68.686 | 53.991 | 62.042 | 61.252 | 68.197 | 61.47 | 63.64 | 165.3 |
| +ASAM | 66.903 | 75.554 | 58.967 | 67.141 | 65.117 | 73.754 | 62.517 | 67.129 | 323.6 |
| +MSAM | 68.768 | 76.378 | 64.932 | 70.026 | 70.885 | 76.342 | 65.068 | 70.765 | 201.3 |
| NG | 78.334 | 83.335 | 83.718 | 81.796 | 77.274 | 78.59 | 85.443 | 80.436 | 309.5 |
| +ASAM | 80.806 | 81.465 | 87.052 | 83.108 | 78.731 | 79.264 | 93.249 | 83.748 | 610.3 |
| +MSAM | 81.811 | 85.568 | 91.176 | 86.185 | 73.291 | 77.43 | 91.691 | 80.804 | 352.4 |
| SharpMinMax | 70.767 | 76.692 | 82.853 | 76.771 | 65.925 | 74.526 | 80.127 | 73.526 | 317 |
| +ASAM | 82.27 | 94.913 | 86.504 | 87.896 | 84.521 | 87.761 | 84.381 | 85.554 | 631 |
| +MSAM | 79.079 | 73.057 | 88.157 | 80.098 | 77.034 | 72.944 | 93.819 | 81.266 | 398.1 |
| RL | 68.464 | 84.395 | 72.4 | 75.086 | 66.689 | 86.411 | 69.677 | 74.259 | 173.3 |
| +ASAM | 69.952 | 86.779 | 74.409 | 77.047 | 69.73 | 91.124 | 80.321 | 80.392 | 344.1 |
| +MSAM | 73.032 | 87.608 | 76.537 | 79.059 | 72.656 | 90.675 | 81.027 | 81.453 | 216 |
| SalUn | 69.926 | 83.056 | 71.73 | 74.904 | 67.355 | 89.768 | 79.095 | 78.739 | 172.8 |
| +ASAM | 73.268 | 92.225 | 88.175 | 84.556 | 67.715 | 93.401 | 89.289 | 83.468 | 340.9 |
| +MSAM | 70.011 | 89.214 | 81.069 | 80.098 | 68.548 | 92.289 | 82.757 | 81.198 | 213.6 |

between unlearned models and retrained models, even when the standard way to retrain models with SGD does not favor SAM as they adopt different optimization dynamics. We observe similar trends as measuring performance closeness with ToWs: while SAM does not improve KL closeness on $\mathcal{F}$, it reduces the KL divergence on $\mathcal{R}$ and $\mathcal{D}_{\text{test}}$ and often halves KL on $\mathcal{R}$. Adding SAM (vs. SGD) reduces the distance to the retrained reference across memorization levels; e.g., with SGD-retrained reference the average distance drops from 0.0973 w/ SGD to 0.0827 w/ SAM. The reported KL at 95%-percentiles in the second table also show that SAM reduces KL even at tails (smaller variances). We use $N = 10$ with 10 bins for our KLoM measurements (pretraining 10 models and retraining 30 models for each $\mathcal{F}$, and unlearning 120 models for all settings).

Table 18: Mean and 95%-percentile KLoM on CIFAR-100 and ResNet50 after NegGrad unlearning. [SGD, SAM] denotes SGD-pretrained and SAM-unlearned. We observe that SAM enhanced unlearning consistently improves KLoM across different $\mathcal{F}$. Similar to what we observe with ToWs, SAM performs better on retain and testset. Based on 95%-percentile KLoM scores, we observe that SAM enhanced unlearning consistently improves KLoM on tails too, which also indicates the better stability of SAM unlearning.

| **KLoM Mean** | $\mathcal{A}$=SGD | | | $\mathcal{A}$=SGD | | | $\mathcal{A}$=SGD | | | AVG |
|---|---|---|---|---|---|---|---|---|---|---|
| $\mathcal{A}, \mathcal{U}$ | Forget | Retain | Test | Forget | Retain | Test | Forget | Retain | Test | |
| SGD, SGD | 0.1294 | 0.0721 | 0.1293 | 0.1221 | 0.0669 | 0.1259 | 0.0284 | 0.0747 | 0.1271 | 0.0973 |
| SGD, SAM | 0.1384 | **0.0411** | **0.1076** | 0.1589 | **0.0331** | **0.0983** | **0.0163** | **0.0411** | **0.1093** | **0.0827** |
| SAM, SGD | 0.1549 | 0.0676 | 0.126 | 0.1513 | 0.066 | 0.1246 | 0.025 | 0.0658 | 0.1248 | 0.1007 |
| SAM, SAM | 0.1714 | **0.0264** | **0.095** | 0.2457 | **0.0253** | **0.0913** | **0.015** | **0.0311** | **0.1008** | **0.0891** |
| **KLoM 95%** | $\mathcal{A}$=SGD | | | $\mathcal{A}$=SGD | | | $\mathcal{A}$=SGD | | | AVG |
| $\mathcal{A}, \mathcal{U}$ | Forget | Retain | Test | Forget | Retain | Test | Forget | Retain | Test | |
| SGD, SGD | 0.3397 | 0.2097 | 0.5212 | 0.4959 | 0.2097 | 0.4901 | 0.0956 | 0.2097 | 0.4959 | 0.3408 |
| SGD, SAM | **0.3397** | **0.2097** | **0.4901** | 0.5681 | **0.0956** | **0.3723** | **0.0956** | **0.2097** | **0.4901** | **0.319** |
| SAM, SGD | 0.4901 | 0.2097 | 0.5212 | 0.5371 | 0.2097 | 0.4959 | 0.0956 | 0.2097 | 0.5212 | 0.3656 |
| SAM, SAM | **0.4901** | **0.0956** | **0.3723** | 0.8111 | **0.0956** | **0.3397** | **0.0956** | **0.0956** | **0.4901** | **0.3206** |

# H COMPLETE VISUALIZATIONS

In this section, we provide complete visualizations of feature space and loss landscapes of pretrained models, NegGrad unlearned models, and Sharp MinMax unlearned models, comparing SGD with SAM across all memorization levels. The observations are generally consistent across memorization levels, with $\mathcal{F}_{\text{high}}$ being more noticeable.

## H.1 LOSS LANDSCAPE

Inspired by Wu et al. (2017), we quantify the flatness by basin ratio, which is the percentage of perturbed losses whose deviation from original loss $\leq 0.5 \cdot$ stddev. Fig. 6 shows loss landscapes of SAM and SGD before and after unlearning on $\mathcal{D}_{\text{test}}$ and $\mathcal{F}_{\text{high}}$. We observe SAM has higher basin ratios (flatter landscape) than SGD for pretrained model and MinMax unlearned model as expected. Surprisingly, SGD can become flatter after unlearning. We conjecture that the gradient ascent might be implicitly regularizing SGD which had more overfitting than SAM during pretraining. We leave the further characterization of loss landscapes to future work.

## H.2 FEATURE VISUALIZATION

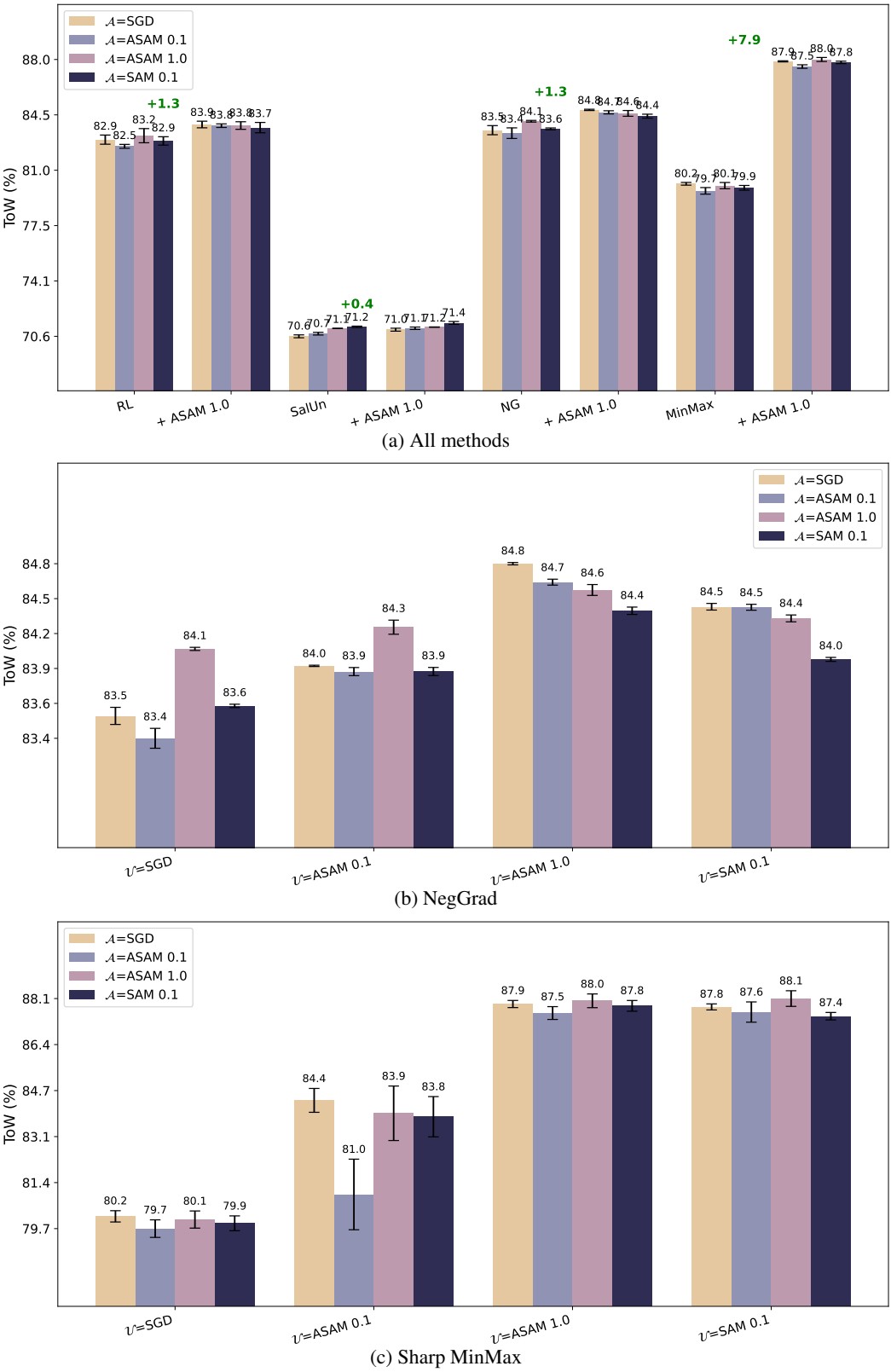

Figure 4: 95% confidence intervals ($\mu \pm 2\sigma$) of unlearning methods on ImageNet, in accordance to Tab. 1 and Tab. 3. We run each setting three times with different seeds and compute the statistical significance. SAM consistently improves base $\mathcal{U}$, and we observe ASAM 1.0 to bring largest improvement steadily.

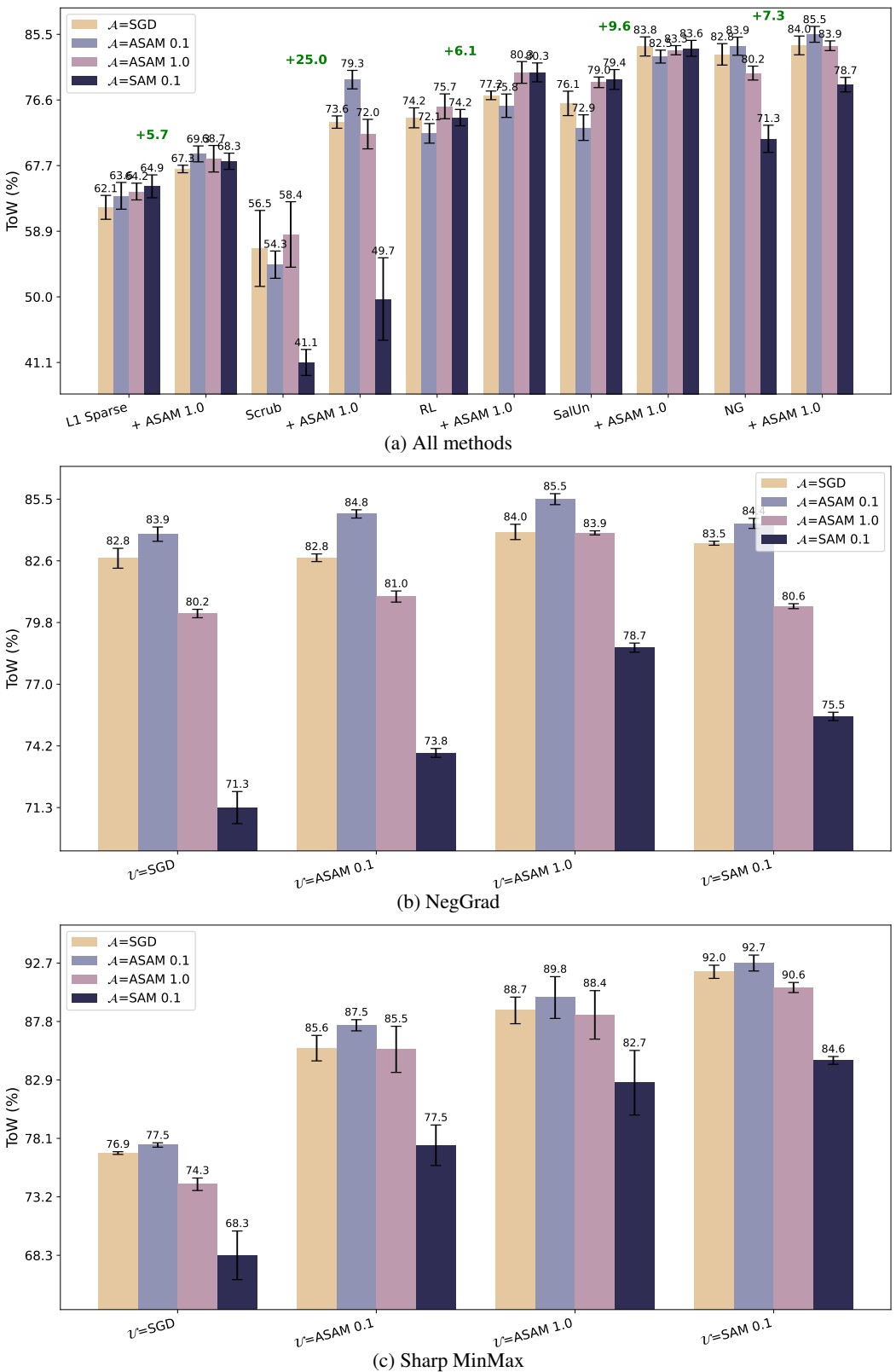

Figure 5: 95% confidence intervals $(\mu \pm 2\sigma)$ of unlearning methods on CIFAR-100, in accordance to Tab. 1 and Tab. 3. We run each setting three times with different seeds and compute the statistical significance. SAM not only improves ToW of the based methods, but also more robust against variance than SGD.

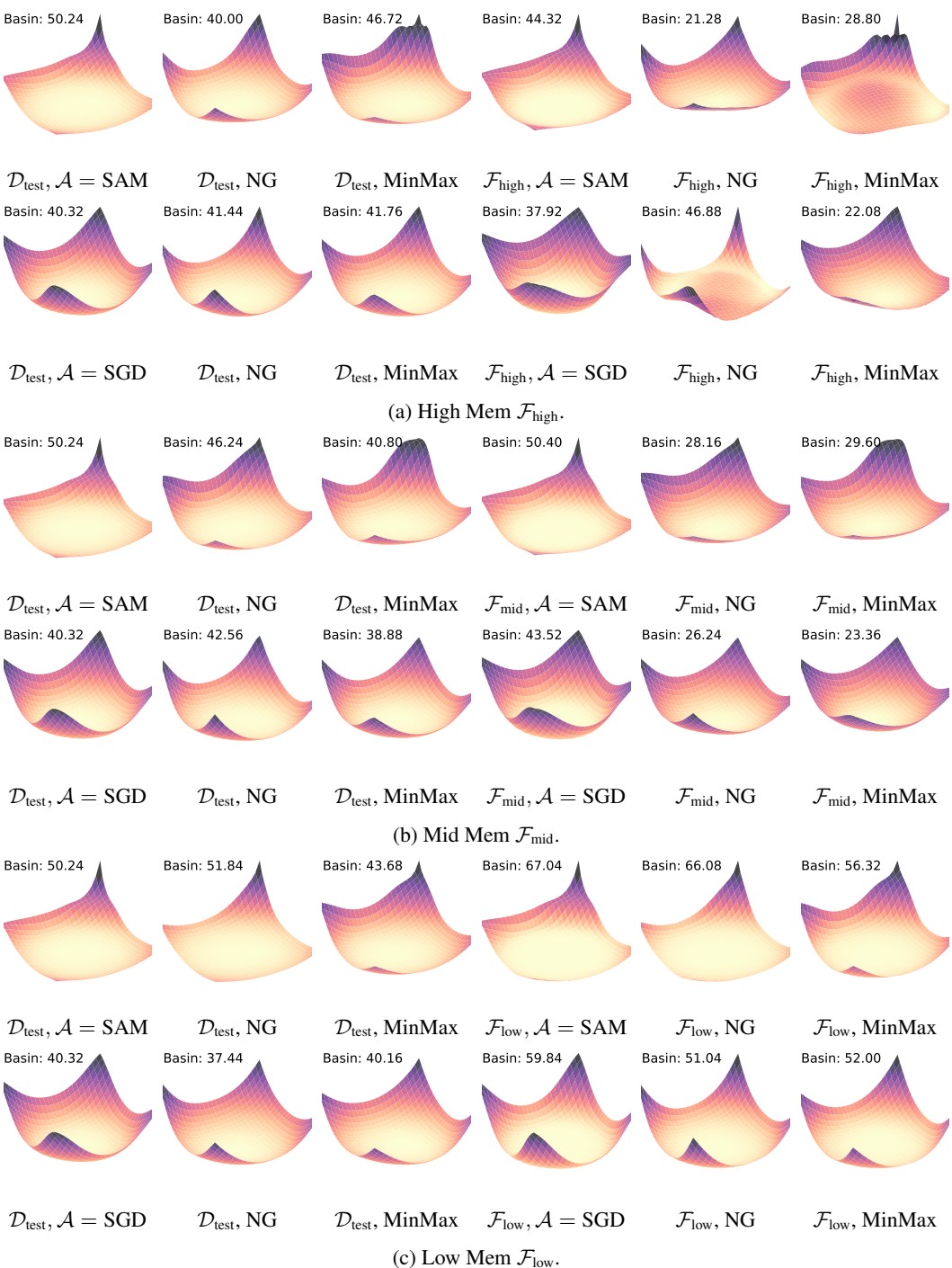

Figure 6: Loss landscapes of SAM and SGD on $\mathcal{D}_{\text{test}}$ and all $\mathcal{F}$. As memorization level goes down, $\mathcal{F}$ becomes easier to unlearn and SGD shows less to no "regularizing" effect as we have discussed on $\mathcal{F}_{\text{high}}$. The general trend preserves with decreasing memorization levels and SAM is generally flatter before and after unlearning.

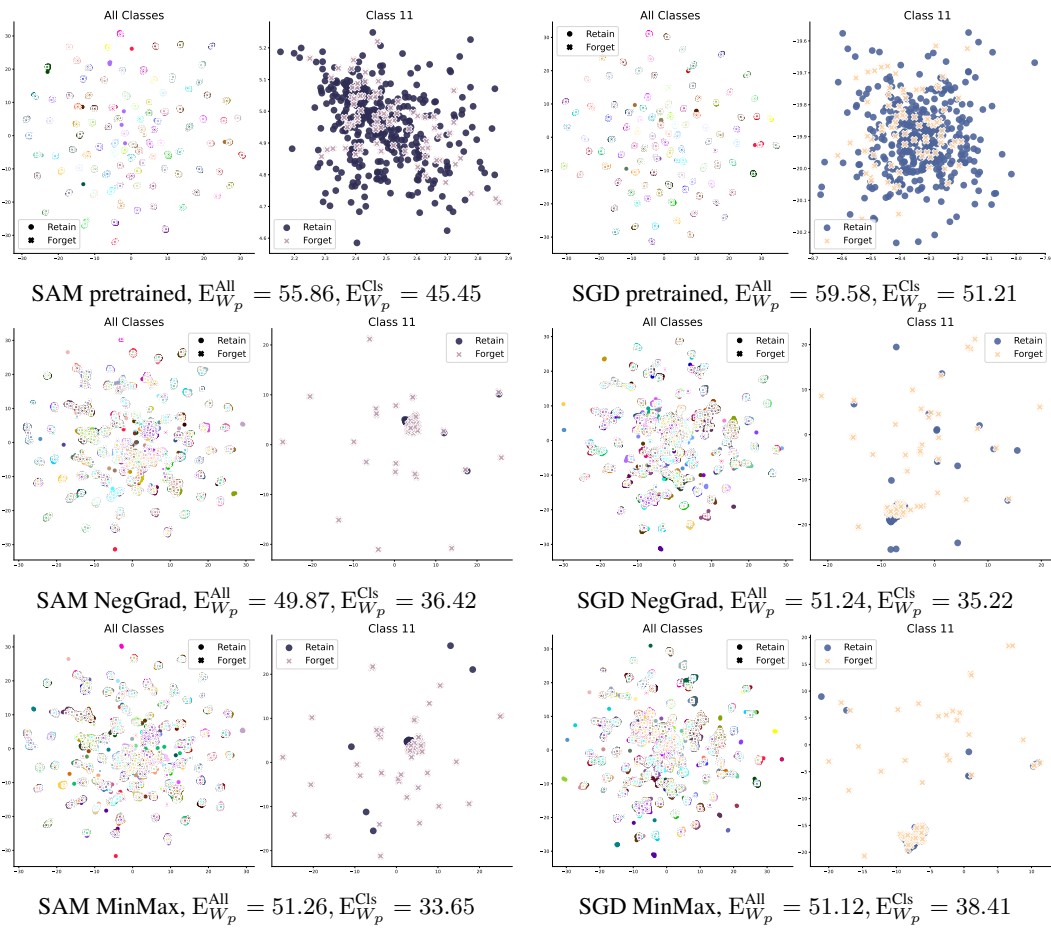

Figure 7: UMAP (McInnes et al., 2018) feature analysis on High Mem $\mathcal{F}_{\text{high}}$. We observe SGD unlearning forms a more obvious clump in all-classes panels while SAM unlearning better maintains class clusters. From classwise panels, we observe that SAM effectively pushes forget samples away while gathering retain samples to a dense cluster, while SGD also scatters retain samples during unlearning, suggesting overfitting.

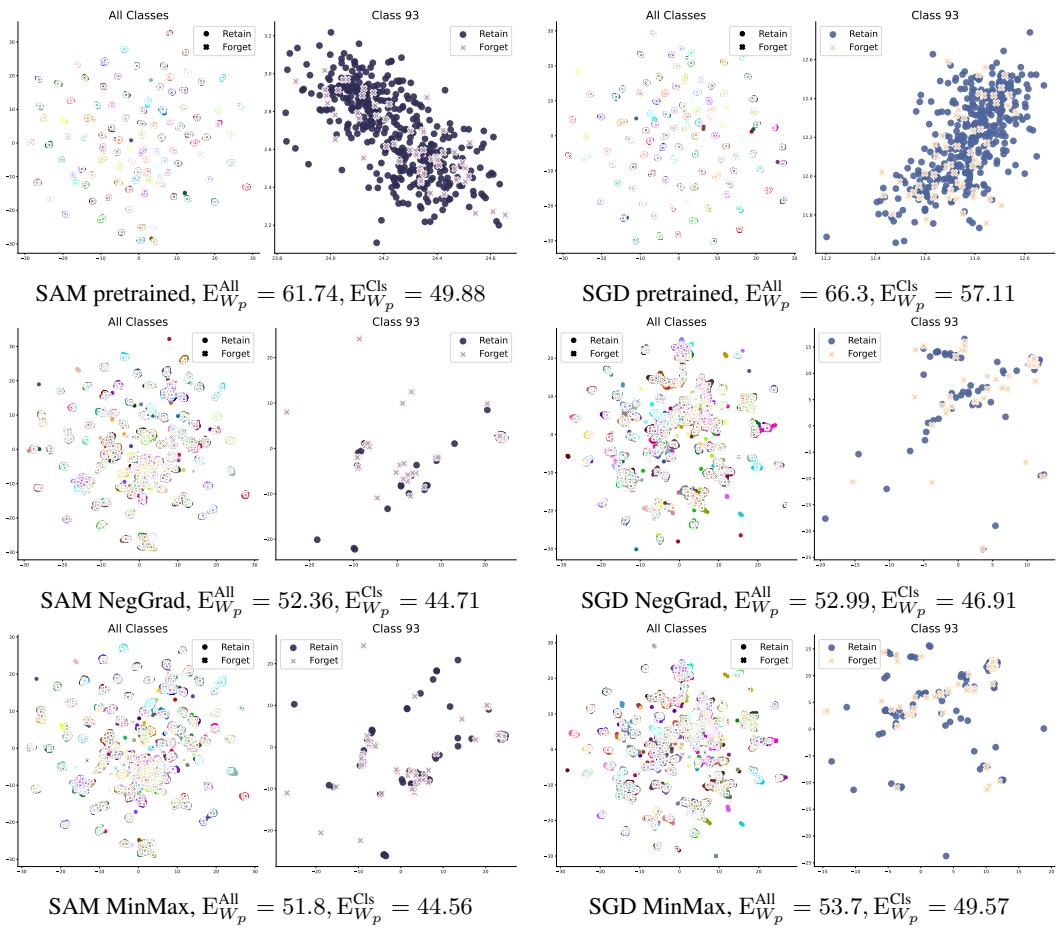

Figure 8: UMAP (McInnes et al., 2018) feature analysis on Mid Mem $\mathcal{F}_{\mathrm{mid}}$. At all-class level, we observe that SAM better maintains class clusters after unlearning while SGD is forming a more evident clump of features; at classwise level, we observe that while both push away forget features, SGD also scatters retain features further, suggesting overfitting. This also explains the larger clump of SGD at all-class level.

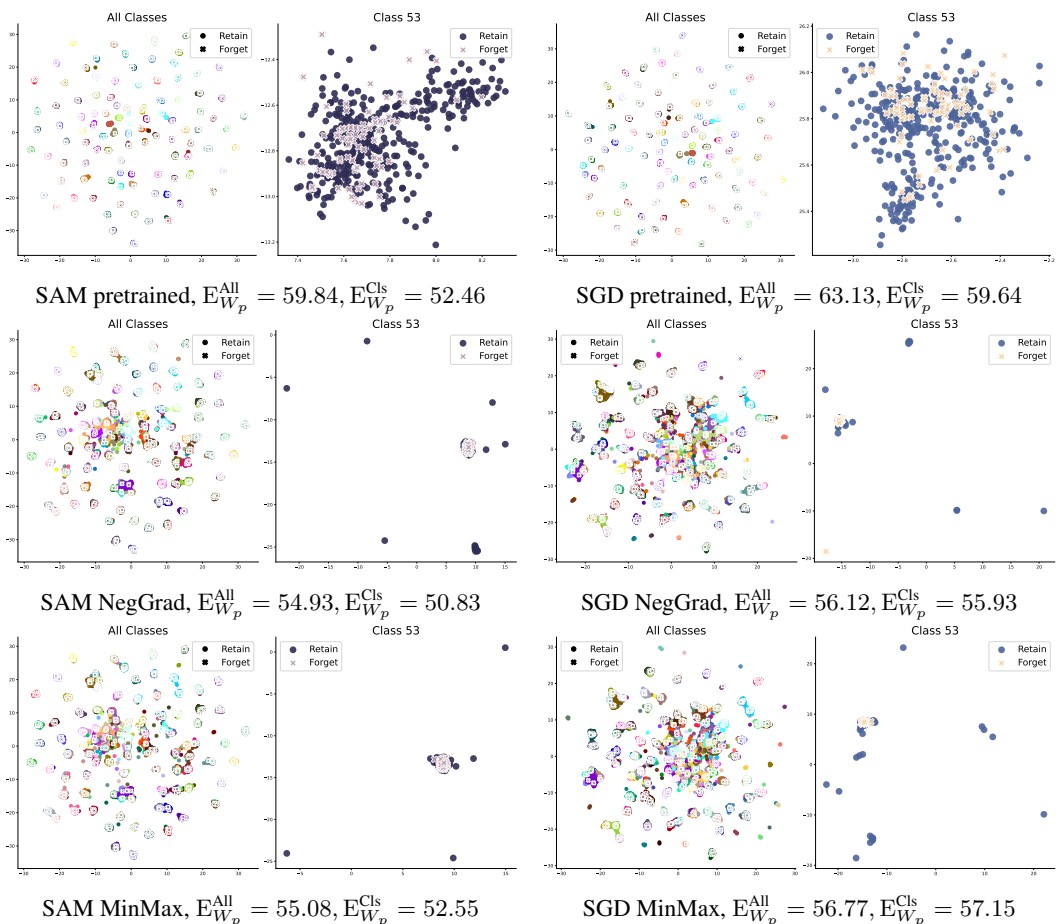

Figure 9: UMAP (McInnes et al., 2018) feature analysis on Low Mem $\mathcal{F}_{\text{low}}$. We observe SGD unlearning forms a more obvious clump in all-classes panels while SAM unlearning better maintains class clusters. From classwise panels, as $\mathcal{F}_{\text{low}}$ requires less unlearning and the model can generalize to $\mathcal{F}_{\text{low}}$, forget samples do not move much as expected. But on SGD MinMax we still observe that SGD scatters more retain samples away.

