# OpenReview forum: "Sharpness-Aware Machine Unlearning"
_ICLR.cc/2026/Conference — ICLR 2026 Poster_

### Official Review · Reviewer_Ti1T · 2025-10-14

**Soundness:** 3
**Presentation:** 4
**Contribution:** 2
**Rating:** 6
**Confidence:** 3

**Summary:**

This paper proposes using SAM for unlearning. They provide theoretical evidence in support of their proposition as well as experimental evaluations that substantiate their claims. The authors also point out that unlearning difficult examples is facilitated by overfitting and as a result the propose Sharp MinMax, a novel algorithm that utilizes this observation.

**Strengths:**

The paper has several contributions.

From a theoretical standpoint:

They show that for the negative gradient unlearning method one can minimize the test error for SAM always under proper assumptions, while this is not the case when someone uses SGD, which is interesting.

From a heuristics standpoint:

Based on their theoretical findings and that SAM fails to remove noise for the forget set unlike its behavior for the retain they suggest a sharp minimax algorithm with the purpose of overfitting to the minima in order to unlearn the samples of the forget set effectively. The idea of overfitting is in general an interesting proposition.

From an empirical standpoint:

The paper provides extensive experimental evaluations of the methods over multiple datasets. With respect to the ToW metric they provide SOTA performance, making the empirical findings substantial in support of their claims.

The paper also provides a study on the effect of the weighting $\alpha$ of the negative gradient method, for SGD and SAM based methods which can be beneficial for the community.

**Weaknesses:**

The main weakness that I find in the paper is an assumption that is made implicitly but yet not discussed substantially. The assumption is the following: "Conversely, a sample i ∈ F of class j, which we want to predict −j in unlearning" as stated in line 217. Although this is some times the case a lot of the time it is not and recent works [1],[2] have demonstrated that empirically and theoretically respectively this assumption will lead to an "illusion" of unlearning where the model does not behave as an oracle model that has never seen the data in the Forget set. This can be remedied with a relevant discussion or a clarification as there are unlearning settings where this assumption actually holds necessarily by definition as they clarify in [3] for the RB or RC unlearning scenarios.

Despite the aforementioned weakness which is inherent in some part of the unlearning literature, in my opinion, I find the idea that one unlearns by overfitting in the retain or through SAMs property of dealing with noise in the retain but not in the forget intriguing.

Additionally I would like to mention a minor weakness in my opinion which is less significant and deducts from the presentation of the work. At line 470 the authors state that the experimental evaluations verify their claim. Since the claim about $\alpha$ does not have an extensive proof I believe that the correct term would be that it supports, instead of "This verifies our claim that α depends more than retain-forget ratio."

[1] Attribute-to-Delete: Machine Unlearning via Datamodel Matching, Kristian Georgiev, Roy Rinberg, Sung Min Park, Shivam Garg, Andrew Ilyas, Aleksander Madry, Seth Neel

[2] Ascent Fails to Forget, Ioannis Mavrothalassitis, Pol Puigdemont, Noam Itzhak Levi, Volkan Cevher

[3] Towards Unbounded Machine Unlearning, Meghdad Kurmanji, Peter Triantafillou, Jamie Hayes, Eleni Triantafillou

**Questions:**

My main questions involve the weaknesses that I issued above.

1. In [1] the authors introduce as a metric for successful unlearning KLoM. Does SAM achieve an improved KLoM when using Negative Gradient in comparison to the SGD alternative?
2. Does SAM resolve the issues raised about data dependencies between retain and forget as raised in [2]?

---

> ### Author Response · Authors · 2025-11-21
> **[1/3] Thank you for your valuable comments.**
>
> Thank you for your acknowledgement and valuable comments! We would like to clarify your concerns with more details, and we will update the manuscript soon for precision and clarity. We appreciate your opinions and would like to engage more.
>
> **Assumption in ascent-based unlearning**
>
> We sincerely thank you for this suggestion, and we would like to clarify that the implicit assumption of predicting $-j$ for a sample $i\in\mathcal{F}$ of class $j$ comes from ascent-based unlearning such as NegGrad as their objectives encourage misclassifying $\mathcal{F}$, and it only applies to ascent-based unlearning. We will make this explicit. We are aware of other $\mathcal{U}$, such as Random Label, that do not enforce misclassifying $\mathcal{F}$ but rather predicting $j,-j$ with equal probabilities as the model has never seen them (random guess). We would like to emphasize that such implicit assumption is $\mathcal{U}$-dependent and does not affect our theoretical analysis on SAM. For example, we build a signal-noise learning framework for Random Label in Eqn. 3, which models forget signals as the noise and the previous analysis on SAM directly applies, without the assumption of forcing misclassification. We continue to analyze SAM under NegGrad as the signal-noise learning has varied formulation and previous results do not apply perfectly. While our theoretical analysis is $\mathcal{U}$-independent and empirical results on arbitrary $\mathcal{U}$ support our findings (e.g., SAM improves NegGrad which has the discussed assumption and also RL and SalUn which do not), and the debate between the ascent-based unlearning and other unlearning (e.g., as \[1\] discusses) is not the main focus of our study, we would like to note that ascent-based unlearning empirically achieves state-of-the-art effectiveness and is widely adopted even in frontier applications, such as NPO and SimNPO in LLM unlearning \[2,3\]. This also motivates us to investigate SAM under an unlearning scheme with conflicting forget signals (gradient ascent).
>
> **Correct use of word for $\alpha$**
>
> Thank you for the suggestion and we will update the manuscript to address our finding more precisely.

---

> > ### Author Response · Authors · 2025-11-21
> > **[2/3] Thank you for your valuable comments.**
> >
> > **SAM improves KLoM**
> >
> > KLoM measures closeness to the retrained model in terms of margin distributions (output space), while our ToW measures direct performance closeness. We follow \[4\] to compute KLoM of  NegGrad with SGD and SAM and report the KL measures on $\mathcal{F}, \mathcal{R}, \mathcal{D}\_{\text{test}}$ across $\mathcal{F}$ of different difficulties, report means and $95\\%$-percentiles in the below tables. Given a pretrained model, we observe that SAM in unlearning also helps close the gap between unlearned models and retrained models, even when the standard way to retrain models with SGD does not favor SAM as they adopt different optimization dynamics. We observe similar trends as measuring performance closeness with ToWs: while SAM does not improve KL closeness on $\mathcal{F}$, it reduces the KL divergence on $\mathcal{R}$ and $\mathcal{D}\_{\text{test}}$ and often halves KL on $\mathcal{R}$. Adding SAM (vs. SGD) reduces the distance to the retrained reference across memorization levels; e.g., with SGD‑retrained reference the average distance drops 0.0973 → 0.0827 (SGD→SAM on the unlearning optimizer). We will add the KLoM description and experiments to App. F. The reported KL at $95\\%$ percentiles in the second table also show that SAM reduces KL even at tails (smaller variances). We use $N=10$ with 10 bins for our KLoM measurements (pretraining 10 models and retraining 30 models for each $\mathcal{F}$, and unlearning 120 models for all settings).
> >
> > **Table**: Mean KLoM on CIFAR-100 and ResNet50 after NegGrad unlearning. \[SGD, SAM\] denotes SGD-pretrained and SAM-unlearned. We observe that SAM enhanced unlearning consistently improves KLoM across different $\mathcal{F}$. Similar to what we observe with ToWs, SAM performs better on retain and testset.
> > | KLoM mean | High Mem |            |            | Mid Mem |            |            | Low Mem    |            |            | AVG               |
> > | --------- | -------- | ---------- | ---------- | ------- | ---------- | ---------- | ---------- | ---------- | ---------- | ----------------- |
> > |           | forget   | retain     | test       | forget  | retain     | test       | forget     | retain     | test       |                   |
> > | SGD, SGD  | 0.1294   | 0.0721     | 0.1293     | 0.1221  | 0.0669     | 0.1259     | 0.0284     | 0.0747     | 0.1271     | 0.09732222222     |
> > | SGD, SAM  | 0.1384   | **0.0411** | **0.1076** | 0.1589  | **0.0331** | **0.0983** | **0.0163** | **0.0411** | **0.1093** | **0.08267777778** |
> > | SAM, SGD  | 0.1549   | 0.0676     | 0.126      | 0.1513  | 0.066      | 0.1246     | 0.025      | 0.0658     | 0.1248     | 0.1006666667      |
> > | SAM, SAM  | 0.1714   | **0.0264** | **0.095**  | 0.2457  | **0.0253** | **0.0913** | **0.015**  | **0.0311** | **0.1008** | **0.08911111111** |
> >
> > **Table**: $95\\%$-percentile KLoM on CIFAR-100 and ResNet50 after NegGrad unlearning, in addition to above mean scores. We observe that SAM enhanced unlearning consistently improves KLoM on tails too, which also indicates the better stability of SAM unlearning.
> > | KLoM p95 | High Mem   |            |            | Mid Mem |            |            | Low Mem    |            |            | AVG              |
> > | -------- | ---------- | ---------- | ---------- | ------- | ---------- | ---------- | ---------- | ---------- | ---------- | ---------------- |
> > |          | forget     | retain     | test       | forget  | retain     | test       | forget     | retain     | test       |                  |
> > | SGD, SGD | 0.3397     | 0.2097     | 0.5212     | 0.4959  | 0.2097     | 0.4901     | 0.0956     | 0.2097     | 0.4959     | 0.3408333333     |
> > | SGD, SAM | **0.3397** | **0.2097** | **0.4901** | 0.5681  | **0.0956** | **0.3723** | **0.0956** | **0.2097** | **0.4901** | **0.3189888889** |
> > | SAM, SGD | 0.4901     | 0.2097     | 0.5212     | 0.5371  | 0.2097     | 0.4959     | 0.0956     | 0.2097     | 0.5212     | 0.3655777778     |
> > | SAM, SAM | **0.4901** | **0.0956** | **0.3723** | 0.8111  | **0.0956**     | **0.3397**     | **0.0956**     | **0.0956**     | **0.4901**     | **0.3206333333**     |

---

> > > ### Author Response · Authors · 2025-11-21
> > > **[3/3] Thank you for your valuable comments.**
> > >
> > > **SAM ameliorates data dependency issue raised in \[1\]**
> > >
> > > Extending from the previous discussion (assumption in ascent-based unlearning), we do not think SAM is targeted to resolve the data dependency issue raised in recent studies such as \[1\], and we do not claim to “solve” data-dependency pathologies. However, the geometric properties of SAM can improve it to some extent as SAM’geometry reduces entanglement between $\mathcal{R}$ and $\mathcal{F}$. More importantly, our Lemma3.4 also shows that SAM requires smaller retain signals to maintain performance (smaller $\alpha$), meaning that **SAM suffers less from the data dependency between $\mathcal{R}$ and $\mathcal{F}$ and it is more resilient to noise (requires lesser signal) than SGD**. We think our observation echoes with theories in \[1\], where ascent-based unlearning will eventually fail. The reason why it does not fail in practice (and rather is being widely used) is that ascent-based unlearning algorithms will tune for an optimal reweighting factor like our $\alpha$ to retain performance, mitigating the performance degradation brought by gradient ascent.
> > >
> > > \[1\] Mavrothalassitis, Ioannis, et al. "Ascent Fails to Forget." NeurIPS 2025.
> > >
> > > \[2\] Zhang, Ruiqi, et al. "Negative preference optimization: From catastrophic collapse to effective unlearning." COLM 2024.
> > >
> > > \[3\] Fan, Chongyu, et al. "Simplicity prevails: Rethinking negative preference optimization for llm unlearning." NeurIPS 2025.
> > >
> > > \[4\] Georgiev, Kristian, et al. "Attribute-to-delete: Machine unlearning via datamodel matching." arXiv preprint arXiv:2410.23232 (2024).

---

> > > > ### Comment · Reviewer_Ti1T · 2025-11-24
> > > >
> > > > I would like to thank the authors for the detailed responses.
> > > >
> > > > 1. **Assumption in ascent-based unlearning** I appreciate this discussion. The clarification that previous results of SAM can apply directly to other machine unlearning methods is pretty illuminating and interesting. My initial concern however had to do with the assumption that in order to unlearn a sample the correct thing would be to model it with its complementary label. Even though currently this is used in many SOTA methods in practice I believe it would be beneficial for future readers if it was stated clearly as an assumption or in a manner that does not imply it is the sole proper technique.
> > > > 2. **SAM improves KLoM** This is actually a very interesting finding and I believe it would be beneficial to include at some point in the paper since it provide empirical evidence that SAM promotes data model matching, in comparison to naive SGD.
> > > > 3. **SAM ameliorates data dependencies** The observation that SAMs geometric properties ameliorates the problems caused by data dependencies as observed in previous literature is indeed interesting and like the previous points is worth mentioning in a revised manuscript of the work.
> > > >
> > > > I would be happy to adjust my score after reviewing the revised manuscript with the aforementioned changes.

---

### Official Review · Reviewer_F8sS · 2025-10-17

**Soundness:** 2
**Presentation:** 3
**Contribution:** 2
**Rating:** 4
**Confidence:** 3

**Summary:**

This paper presents a theoretical and empirical investigation of machine unlearning through the perspective of sharpness-aware optimization. The paper studies how SAM, originally designed to improve generalization, behaves under unlearning scenarios that involve conflicting forget and retain signals.

**Strengths:**

1. The authors develop a signal–noise decomposition framework to analyze how SAM and SGD behave when learning (retain) and unlearning (forget) signals interact, giving novel insights on how SAM affects unlearning performance.
2. The paper further proposes Sharpness MinMax, a novel optimization method that improves machine unlearning performance.
3. The effectiveness of the proposed method is evaluated on multiple datasets. The paper also gives additional evaluations such as feature entanglement and privacy evaluations.

**Weaknesses:**

1. For the evaluation metrics, run-time evaluations are missing. Employing SAM on both forget and retain loss might significantly increase run time, and this efficiency trade-off should be evaluated.

2. Evaluations of unlearning robustness are missing, such as evaluating the performance of the unlearned models against relearning attacks [1]. Sharpness-aware minimization employed in LLM unlearning [2] increases the robustness of the unlearned models against relearning by perturbing the model with a 'worst-case' perturbation at each optimization step. In light of [2], does Sharp MinMax decrease the robustness of unlearning by maximizing the sharpness of the forget loss? I think this concern should be studied in this paper.

3. There are limited explanations on the weight mask design for Sharp MinMax. The authors use 5 % cutoff for ImageNet and 30 % cutoff for CIFAR‑100. However, it seems that there are no explanations or ablation studies on this.

4. Please refer to the questions for additional concerns.

[1] From Dormant to Deleted: Tamper-Resistant Unlearning Through Weight-Space Regularization

[2] Towards LLM Unlearning Resilient to Relearning Attacks: A Sharpness-Aware Minimization Perspective and Beyond

**Questions:**

1. In Section 3.2, line 290-294: Could the authors explain the motivation of this weight mask design, and why it is different from SalUn [1]?
2. I think a rather important baseline is missing in the experiments: performing SAM on the retain loss only. According to the paper, this should improve retain performance without degrading forget performance. Additionally, this can be more efficient in run-time compared to Sharp MinMax. By comparing this baseline to Sharp MinMax, the authors can show the effectiveness of sharpness maximization on the forget loss. Could the authors explain why they did not include this baseline in the experiments?

3. Could the findings in this paper be extended to machine unlearning in image generation tasks [1]? This could greatly enhance the soundness of the proposed method.

[1] https://arxiv.org/pdf/2310.12508

---

> ### Author Response · Authors · 2025-11-21
> **[1/3] Thank you for your valuable comments.**
>
> Thank you for your acknowledgement and valuable feedback. We would like to further clarify your concerns with more details and experiment results, and we will update our manuscript soon. We welcome further discussions and would like to engage more.
>
> **Expense of SAM and its runtime**
>
> Our goal is the improved performance of unlearning. The original SAM is indeed slower than SGD, but as we summarize in App. A Related Works, faster SAM variants are another prospering direction and there are mature SAM variants as fast as SGD. Earlier work includes Sparse SAM \[1\] and LookSAM \[2\] which outperform SGD with $10\%\sim30\%$ computation overhead; more recent work such as Momentum SAM \[3\] and Parallel SAM \[4\] can outperform SGD without significant computational overhead already. We adopt vanilla SAM and ASAM (adaptive perturbation radius) for easier implementation and parameter tuning to fulfill our investigation goals, and SAM’s expense is not the main focus of our study. Thus, further research on reducing run time of SAM will also speed up our method, but again – our focus is on improved unlearning performance in this paper which has privacy implications of practical adaptations of AI models, and the increased overhead of SAM-variants is a minor cost to pay for that.
>
> **Unlearning robustness under relearning attacks**
>
> We present relearning attack experiments to demonstrate SAM's unlearning robustness below. We take the unlearned models to relearn the whole $\mathcal{F}$ for one epoch with a small relearning lr, and measure the increase in forget accuracies. Reported are the averaged increase across $\mathcal{F}\_{\text{high}}, \mathcal{F}\_{\text{mid}}, \mathcal{F}\_{\text{low}}$. We observe that SAM enhanced $\mathcal{U}$ are more resilient to relearning attacks with smaller increases. We note that these experiments highlight the robustness of our approach and hope that this encourages future works for deeper investigation into the role of loss landscape geometry for robust unlearning.
>
> **Table**: Average increase of forget accuracies after relearning 1 epoch on $\mathcal{F}$ on CIFAR-100 and ResNet50. We observe that SAM enhanced unlearning is consistently more resilient to relearning attacks with less increase on forget accuracy.
> |             | Relearn lr=0.002 |                 | Relearn lr=0.003 |                 | Relearn lr=0.004 |                 |
> | ----------- | ---------------- | --------------- | ---------------- | --------------- | ---------------- | --------------- |
> |             | SGD pretrained   | ASAM pretrained | SGD pretrained   | ASAM pretrained | SGD pretrained   | ASAM pretrained |
> | NG          | 8.644446667      | 10.33333        | 11.16666667      | 13.25555333     | 12.78889         | 14.7            |
> | +ASAM       | **8.533333333**  | **9.288886667** | **11.03333667**  | **11.53333**    | 13.02222333      | **13.38889**    |
> | SharpMinMax | 13.1             | 15.06666667     | 15.58888667      | 17.49999667     | 16.14444667      | 18.51111        |
> | +ASAM       | **8.33333**      | **8.8**         | **10.66667**     | **11.20000333** | **12.71111**     | **12.66667**    |
> | RL          | 7.122223333      | 8.555556667     | 8.5              | 9.58889         | 9.622223333      | 10.98889        |
> | +ASAM       | **6.222223333**  | **7.377776667** | **7.444446667**  | **8.488886667** | **8.366666667**  | **9.46667**     |

---

> > ### Author Response · Authors · 2025-11-21
> > **[2/3] Thank you for your valuable comments.**
> >
> > **Details about weight mask design, difference between SalUn and cutoff choice**
> >
> > In App. E.2, we have provided a detailed illustration of SharpMinMax and its implementation, where we summarize weight mask construction in Alg. 1 and SharpMinMax in Alg. 2. We follow the same way as SalUn \[5\] to construct weight masking but **leverage the mask in different ways**. For SalUn, it applies weight masking to obtain the forget model $\mathbf{W}\_{\mathcal{F}}$, and perform random-label unlearning on it. Our SharpMinMax on the other hand, will obtain the retain model as $\mathbf{W}\_{\mathcal{F}}$'s complement: $\mathbf{W}\_{\mathcal{R}} = \mathbf{W} \setminus \mathbf{W}\_{\mathcal{F}}$, and we perform SAM on $\mathbf{W}\_{\mathcal{R}}$ and sharpness maximization on $\mathbf{W}\_{\mathcal{F}}$. The $5\\%$ for ImageNet and $30\\%$ for CIFAR-100 cutoff choices are hyper-parameters chosen from empirical results, but they are also grounded by the over-parameterization scheme: Since ImageNet with ResNet50 is much less over-parameterized compared to CIFAR-100+ResNet50, we need to carefully limit the size of $\mathbf{W}\_{\mathcal{F}}$ to maintain performance; on the other hand, since CIFAR-100+ResNet50 are much more over-parameterized, we have a greater freedom of assigning forget parameters and can choose a larger $\mathbf{W}\_{\mathcal{F}}$ to unlearn more aggressively. SalUn uses $50\\%$ (median) for CIFAR and does not unlearn on ImageNet. Our performance is relatively stable over a good regime of these hyper-parameters. We did not finetune for ImageNet due to resource budget, so our numbers will probably improve with more finetuning of this hyper-parameter. We will provide ablation studies on mask cutoff on CIFAR-100 soon.
> >
> > **SAM on retain loss, SGD on forget loss**
> >
> > We thank the reviewer for the suggestion. We provide experiment results with SAM on $\mathbf{W}\_{\mathcal{R}}$ and SGD on $\mathbf{W}\_{\mathcal{F}}$. This is the "motivation setting" for SharpMinMax that best aligns with our empirical observations when applying SAM to various $\mathcal{U}$. We did not include this setting as we consider SGD on both $\mathbf{W}\_{\mathcal{R}}, \mathbf{W}\_{\mathcal{F}}$ to be the baseline, and replacing SGD on $\mathbf{W}\_{\mathcal{R}}$ with SAM will promisingly improve the performance. But we agree that it can be a valuable setting to evaluate the intentional overfitting design on $\mathbf{W}\_{\mathcal{F}}$. The below table reports ToWs for \[SAM,SGD\] across $\mathcal{F}\_{\text{high}}, \mathcal{F}\_{\text{mid}}, \mathcal{F}\_{\text{low}}$ in addition to results in Tab. 3 for CIFAR-100. Using SAM on retain loss and SGD forget loss improves the base \[SGD, SGD\] setting by a small margin, why using sharpness maximization on forget loss further improves the ToW. This demonstrates the effectiveness of SharpMinMax as it actively searches for sharp landscapes for forget signals.
> >
> > **Table**: SharpMinMax on CIFAR-100 and ResNet50. We additionally experiment with SAM on $\mathbf{W}\_{\mathcal{R}}$ and SGD on $\mathbf{W}\_{\mathcal{F}}$, and observe that \[SAM,SGD\] performs slightly better than the baseline \[SGD, SGD\] settings. Our proposed SharpMinMax achieves a significantly better performance on average.
> > | SharpMinMax | SGD pretrained |            |            |            | SAM pretrained |            |            |            |
> > | ----------- | -------------- | ---------- | ---------- | ---------- | -------------- | ---------- | ---------- | ---------- |
> > |             | High           | Mid        | Low        | AVG        | High           | Mid        | Low        | AVG        |
> > | SGD, SGD    | 70.7668        | 76.692     | 82.853     | 76.771     | 65.925         | 74.526     | 80.127     | 73.526     |
> > | SAM, SGD    | 70.002         | 75.786     | **86.732** | 77.507     | 65.583         | 75.505     | **89.089** | 76.726     |
> > | Min, Max    | **82.27**      | **94.913** | 86.504     | **87.896** | **84.521**     | **87.761** | 84.381     | **85.554** |

---

> > > ### Author Response · Authors · 2025-11-21
> > > **[3/3] Thank you for your valuable comments.**
> > >
> > > **Extending to image generation tasks**
> > >
> > > We believe that the theoretical and empirical findings of our study should generalize to image generation tasks as in \[5\] (moreover, a concurrent work we have discussed suggests that SAM benefits LLM unlearning too but without any theoretical justification or new algorithm design \[6\]), as long as the base $\mathcal{U}$ adopts gradient-based optimization to unlearn. This is because the benefits of SAM to unlearning stems from its extra optimization and SAM aims to enhance the base optimizer. As we demonstrate that SAM improves $\mathcal{U}$ with both SGD and Adam as base optimizers, we believe it is highly extensible to major optimizers and optimization-based tasks.
> > >
> > > \[1\] Mi, Peng, et al. "Make sharpness-aware minimization stronger: A sparsified perturbation approach." NeurIPS 2022.
> > >
> > > \[2\] Liu, Yong, et al. "Towards efficient and scalable sharpness-aware minimization." CVPR 2022.
> > >
> > > \[3\] Becker, Marlon, Frederick Altrock, and Benjamin Risse. "Momentum-sam: Sharpness aware minimization without computational overhead." arXiv:2401.12033.
> > >
> > > \[4\] Xie, Wanyun, Thomas Pethick, and Volkan Cevher. "Sampa: Sharpness-aware minimization parallelized." NeurIPS 2024.
> > >
> > > \[5\] Fan, Chongyu, et al. "Salun: Empowering machine unlearning via gradient-based weight saliency in both image classification and generation." ICLR 2024.
> > >
> > > \[6\] Fan, Chongyu, et al. "Towards llm unlearning resilient to relearning attacks: A sharpness-aware minimization perspective and beyond." ICML 2025.

---

> > ### Comment · Reviewer_F8sS · 2025-11-22
> >
> > Thanks for the response from the authors.
> >
> > However, I am still confused about two points in my original review. I wonder if the authors could clarify them.
> >
> > First, I agree that there are SAM variants that are efficient and that your work is only focused on unlearning. However, I think run time evaluation is needed, so that the readers can understand the trade-offs between your enhanced unlearn performance and the additional computation cost. My concern is that your SharpMinMax framework might double the run time, so I am interested in the run time evaluation. For now, I cannot find any evaluation on this.
> >
> > As for the SAM variants, we cannot assume that they can also fit into your proposed method without validations.
> >
> > Second, the table on relearning attack robustness is slightly confusing to me. My original concern is that applying sharpness maximization on the forget loss seems to contradict the established literature on robust unlearning [1], which performs sharpness minimization instead. Since the formulation in [1] makes sense to me, I feel that sharpness maximization on the forget loss could degrade the unlearning robustness by reducing the smoothness of the loss landscape. Could the authors discuss this?
> >
> > In your table, it seems that SharpMinMax (your proposed method) has the worst robustness, even worse than NG and RL? Could the authors kindly clarify? Thanks!
> >
> > [1] Towards LLM Unlearning Resilient to Relearning Attacks: A Sharpness-Aware Minimization Perspective and Beyond

---

> > > ### Author Response · Authors · 2025-11-28
> > > **SAM runtime and trying efficient SAMs**
> > >
> > > Thank you for your continued interest. Below we provide the total runtime of SAM vs. SGD for unlearning on a single NVIDIA A100. As both theory and empirical results suggest, vanilla SAM costs nearly twice as SGD. Note that SharpMinMax does not cost more than NegGrad+SAM except a trivial overhead to apply masking.
> > >
> > > Besides less computation overhead, it is promising to see efficient SAMs performing equally well in unlearning since they prove/show themselves to be equivalent to vanilla SAM. Here we implement momentum SAM (MSAM) for unlearning on CIFAR-100. MSAM not only outperforms vanilla SAM by much less computation overhead but can also outperform by average ToWs for some unlearning methods. This is plausible, as recent efficient SAMs reduce computation with more informative perturbation directions than stochastic by momentum buffer, sparsity, prior gradients, sharpness-sensitive data, etc. But there is no clear theoretical justification of MSAM rather than trying to stabilize the noise and reduce overhead of SAM. This warrants a deeper study beyond our scope – our focus is to show superiority of loss landscape based methods for unlearning without worrying about speed (just like the original SAM paper), and we leave deeper theoretical/algorithmic improvements and empirical evaluations for speedups for future work. We will incorporate this in the document too as future work.
> > >
> > > We notice that while outperforming SGD with much less computation overhead than vanilla SAM, MSAM does not outperform ASAM on SharpMinMax and SalUn. As we also observe that MSAM behaves differently from ASAM on different forget sets, further and deeper investigation is needed to study the interactions between MSAM and weight masking to improve the performance. Our results have effectively demonstrated an example of a faster SAM variant that predictably benefits unlearning with less overhead.
> > >
> > > **Table**: Total runtime (seconds) on both CIFAR-100 and ImageNet. It is expected that vanilla nearly doubles runtime as it performs doubled optimization steps.
> > >
> > > | CIFAR-100    | NG      | SharpMinMax     | RL      |
> > > | ------------ | ------- | --------------- | ------- |
> > > | With SGD     | 309.5   | 317             | 173.3   |
> > > | With SAM     | 610.3   | 631             | 344.1   |
> > > | **ImageNet** | **NG**  | **SharpMinMax** | **RL**  |
> > > | With SGD     | 13369.9 | 13463.82        | 7122    |
> > > | With SAM     | 26633.2 | 26965.02        | 14141.8 |
> > >
> > > **Table**: ToWs of MSAM across different $\mathcal{F}$ in addition to reported results of baseline and SAM-enhanced unlearning on CIFAR-100. We observe that MSAM not only costs much less computation overhead than SAM but can also outperform by ToW for some settings, since it leverages a smarter perturbation based on momentum buffer.
> > >
> > > | | $\mathcal{A}$=SGD| | | | $\mathcal{A}$=ASAM| | | | Runtime |
> > > | ----------- | -------------- | ------ | ------ | ---------- | --------------- | ------ | ------ | ---------- | ------- |
> > > |             | High           | Mid    | Low    | AVG        | High            | Mid    | Low    | AVG        |         |
> > > | L1-Sparse   | 63.448         | 68.686 | 53.991 | 62.042     | 61.252          | 68.197 | 61.47  | 63.64      | 165.3   |
> > > | +ASAM       | 66.903         | 75.554 | 58.967 | 67.141     | 65.117          | 73.754 | 62.517 | 67.129     | 323.6   |
> > > | +MSAM       | 68.768         | 76.378 | 64.932 | **70.026** | 70.885          | 76.342 | 65.068 | **70.765** | 201.3   |
> > > | NG          | 78.334         | 83.335 | 83.718 | 81.796     | 77.274          | 78.59  | 85.443 | 80.436     | 309.5   |
> > > | +ASAM       | 80.806         | 81.465 | 87.052 | 83.108     | 78.731          | 79.264 | 93.249 | **83.748** | 610.3   |
> > > | +MSAM       | 81.811         | 85.568 | 91.176 | **86.185** | 73.291          | 77.43  | 91.691 | 80.804     | 352.4   |
> > > | SharpMinMax | 70.767         | 76.692 | 82.853 | 76.771     | 65.925          | 74.526 | 80.127 | 73.526     | 317     |
> > > | +ASAM       | 82.27          | 94.913 | 86.504 | **87.896** | 84.521          | 87.761 | 84.381 | **85.554** | 631     |
> > > | +MSAM       | 79.079         | 73.057 | 88.157 | 80.098     | 77.034          | 72.944 | 93.819 | 81.266     | 398.1   |
> > > | RL          | 68.464         | 84.395 | 72.4   | 75.086     | 66.689          | 86.411 | 69.677 | 74.259     | 173.3   |
> > > | +ASAM       | 69.952         | 86.779 | 74.409 | 77.047     | 69.73           | 91.124 | 80.321 | 80.392     | 344.1   |
> > > | +MSAM       | 73.032         | 87.608 | 76.537 | **79.059** | 72.656          | 90.675 | 81.027 | **81.453** | 216     |
> > > | SalUn       | 69.926         | 83.056 | 71.73  | 74.904     | 67.355          | 89.768 | 79.095 | 78.739     | 172.8   |
> > > | +ASAM       | 73.268         | 92.225 | 88.175 | **84.556** | 67.715          | 93.401 | 89.289 | **83.468** | 340.9   |
> > > | +MSAM       | 70.011         | 89.214 | 81.069 | 80.098     | 68.548          | 92.289 | 82.757 | 81.198     | 213.6   |

---

> > > ### Author Response · Authors · 2025-11-28
> > > **SharpMinMax being resilient to relearning attacks**
> > >
> > > This is related to the weight masking cutoff we have discussed, and since the model is over-parameterized and we are splitting a smaller portion of parameters for sharpness maximization, there is less overlap with the retaining model. Also, as we apply different perturbation directions to the retaining and forgetting models, they further reduce the overlap by becoming geometrically distinct. Thus, the impact of relearning attacks is limited to the sharp terrains, so the robustness against relearning attacks as well as model performance is retained (and they are related). Consequently, for a larger-scale, less over-parameterized setting such as ImageNet+ResNet50, it benefits a smaller mask cutoff as there is less room for separating parameters.
> > >
> > > We want to clarify that SharpMinMax achieves slightly better robustness against relearning attacks than NegGrad+SAM. The baseline SharpMinMax uses SGD on both models, it is testing the weight masking only and does not actually perform sharpness minimization and maximization. Thus, retain and forget models are not geometrically-distinguishable than SAM on $\mathbf{W}\_{\mathcal{R}}$ and sharpness maximization on $\mathbf{W}\_{\mathcal{F}}$. Besides, since retain and forget models are separated, both models do not learn from descent and ascent signals jointly like NegGrad, making SGD flawed under SharpMinMax.
> > >
> > > We provide ablation study on mask cutoff for SharpMinMax on CIFAR-100, experimenting with $10\\%$ and $50\\%$. We observe that all settings can perform well, with $50\\%$ slightly outperforming the rest since it is the most aggressive.
> > >
> > > | SharpMinMax | $\mathcal{A}$=SGD |        |        |            | $\mathcal{A}$=ASAM |        |        |            |
> > > | ----------- | ----------------- | ------ | ------ | ---------- | ------------------ | ------ | ------ | ---------- |
> > > |             | High              | Mid    | Low    | AVG        | High               | Mid    | Low    | AVG        |
> > > | 10%         | 82.675            | 92.495 | 87.636 | 87.602     | 83.916             | 90.27  | 81.362 | 85.183     |
> > > | 30%         | 82.27             | 94.913 | 86.504 | 87.896     | 84.521             | 87.761 | 84.381 | 85.554     |
> > > | 50%         | 82.798            | 98.177 | 87.806 | **89.594** | 83.567             | 95.516 | 90.096 | **89.726** |

---

### Official Review · Reviewer_U1T3 · 2025-10-29

**Soundness:** 2
**Presentation:** 1
**Contribution:** 2
**Rating:** 4
**Confidence:** 3

**Summary:**

This paper revisits Sharpness-Aware Minimization (SAM) in the context of machine unlearning (under the NegGrad unlearning framework) and demonstrates that SAM exhibits a desirable property in distinguishing between the retain and forget sets, as the forget set tends to overfit. The analysis is conducted using a simple two-layer CNN setup. The authors further observe that incorporating SAM into NegGrad allows effective unlearning with a smaller ratio between the retain and forget sets. Experimental results consistently show that SAM-augmented methods outperform their counterparts that do not employ SAM regularization.

**Strengths:**

1. The paper provide in-depth theoretical analysis on how SAM would impact the unlearning quality.
2. Experiments contains sufficient evidence on how well SAM argumented model performs compared to the original classic unlearning methods.

**Weaknesses:**

1. The paper is not well polished and contains numerous grammatical errors and excessive notation, which makes it difficult to read. While analytical papers often introduce complex notation to convey ideas precisely, this paper’s presentation suffers from a combination of grammar issues, overly long sentences, and vague explanations. A few specific examples:
    1. Equation 8 and corresponding description is very vague, and I cannot get how this equation is used in the proposed work. Is it on forget set right? How it connects to all the description in 3.1? There is no sufficient connection given here.
    2. Line 115: Each image consists of ..... This sentence is long, contains grammar error, and it is not parsable.
    3. Definition of noise coefficients $\xi_{j,r,i}^{(t,b)}$. I got totally lost with such type of notions. I think it is unnecessary notation complexity introduced; without them, the paper reads fine.

2. The use of Sharpness-Aware Minimization (SAM) in unlearning is not new. The claimed contribution of this work is therefore unclear. If the main purpose is to provide analysis, the key takeaway appears to be that SAM allows unlearning with a smaller retain set. However, in practice, the exact ratio of the retain set is not a major bottleneck in most unlearning scenarios. The authors should justify why this analysis provides meaningful new insight or practical benefit compared to prior work.

3. All experimental results are reported solely using the ToW metric, which does not appear to be a standard evaluation measure for unlearning. While introducing a new metric can be valuable, the absence of established unlearning metrics (e.g., retention accuracy, forgetting efficacy, or relearning resistance) raises concern. It remains unclear whether SAM genuinely improves unlearning performance in practice. A broader evaluation would be needed to substantiate the claimed advantages.

**Questions:**

Where equation 8 is used in the proposal eventually? Is it used to train on forget set?
Why is the discovery of smaller ratio $\alpha$ such useful? Is it generalizable to other unlearning algorithms? or those with gradient descent/ascent only?

---

> ### Author Response · Authors · 2025-11-21
> **[1/2] Thank you for your valuable comments.**
>
> Thank you for your acknowledgement and valuable comments. We would like to further address your concerns with more details, and we will update our manuscript soon for better clarity. We welcome further discussions and would like to engage more.
>
> **Manuscript clarity, Equation 8**
>
> Eqn. 8 is the sharpness maximization part applied only to the forget set $\mathcal{F}$ and the forget model in SharpMinMax, which just flips the sign of the sharpness term in the regular SAM (Eqn. 5). We will update the manuscript to emphasize this and revise Eqn. 8 to denote it: $\min\_{\mathbf{W}\_{\mathcal{F}}} \mathcal{L}(\mathbf{W}\_{\mathcal{F}}, \mathcal{F})-\left[\max \_{\widehat{\boldsymbol{\epsilon}}} \mathcal{L}(\mathbf{W}\_{\mathcal{F}}+\widehat{\boldsymbol{\epsilon}}, \mathcal{F})-\mathcal{L}(\mathbf{W}\_{\mathcal{F}}, \mathcal{F})\right]$. We apologize for enriched notation as the detailed super/sub scripts are for per-batch and per-filter formulations, which are also adopted in previous theoretical work \[1,2\]. We will further simplify the notation in the main paper for clarity (e.g., abbreviate filter subscript), and keep full notation in Appendix for proof process. We are happy to engage further for clarity on any points for the reviewer. The notations are necessary to convey our theoretical contributions that guide new insights to develop our algorithms and are also of independent interest.
>
> **Novelty of SAM unlearning, insights of the analysis**
>
> We respectfully disagree. We are not familiar with any work that explicitly characterizes the role of loss landscape (flatness of minima) for unlearning, and more specifically characterize SAM under unlearning objectives with mixed retain/forget signals. Two key theoretical findings (proved with 2-layer CNN) directly inform our algorithmic design:
> 1. Lemma 3.1: Under NegGrad, SAM’s usual noise suppression “shuts off” on the forget set – SAM overfits to $\mathcal{F}$ almost as much as SGD – while retaining its denoising advantage on $\mathcal{R}$.
> 2. Lemma 3.4/Theorem 3.3: SAM exhibits faster signal learning on $\mathcal{R}$ and thus tolerates a strictly smaller retain weight $\alpha$ than SGD for more stable unlearning (see Fig. 2).
> These results both explain why/how SAM helps and why constrained overfitting on $\mathcal{F}$ improve sample-specific forgetting – which guided our SharpMinMax design (split parameters’ SAM on $\mathcal{R}$, sharpness maximization on $\mathcal{F}$). This is all described in Sec. 3.1, 3.2. None of this is known in any previous works.
>
> As we mentioned in our manuscript, the concurrent work \[3\] empirically studies the broader smoothness aspect (SAM as smoothing tool, along with gradient penalty, weight averaging, etc.) in LLM unlearning. They are leveraging smoothness for interesting insights. Our work dives into theory to investigate the process of SAM optimization in unlearning and answers why we could unlearn better with SAM through theoretical analysis, modeling retain samples as signals and forget samples as noises, and studying the signal and noise updates. We further leverage the geometric properties of SAM to place the forget signals in noisy, sharp landscapes while gathering retain signals in clean, flat terrain to achieve effective unlearning. \[3\] does not study how retain and forget signals are processed and accumulated with SAM, and does not derive new theoretical results and algorithms. Another prior work \[4\] leverages sharpness to select parameters (build forget model) to unlearn, but does not involve SAM’s optimization process.
>
> We kindly hope the reviewer elaborates on why "the exact ratio of the retain set is not a major bottleneck in most unlearning scenarios" since the purpose of unlearning is to remove unwanted data influence with minimal cost. This ratio governs the regime under which unlearning is provably possible, and we show theoretically in Lemma 3.4 that SAM’s regime of effective unlearning is strictly larger than that of SGD, and further we also show empirically in Fig.2 the relative collapse of unlearning under different values of $\alpha$ for SGD vs SAM – the ratio of the retain set (and thus ratio of retain signals) does matter and the unlearning will fail without sufficient retaining signals. Additionally, since SAM exhibits faster signal learning, it indicates that it requires less finetuning on retain samples (smaller $\alpha$ threshold) than SGD, or it can unlearn faster or unlearn more forget samples in other words. Together, these results highlight that SAM is a more effective unlearning optimizer.

---

> > ### Author Response · Authors · 2025-11-21
> > **[2/2] Thank you for your valuable comments.**
> >
> > **Fine-grained details and accuracies behind ToW not provided**
> >
> > We already provide all the detailed retain, forget, test accuracies in Tab. 8,9,10 in the Appendix, which are used to compute ToWs in Tab. 1,3 for all methods on both ImageNet and CIFAR-100. Tab. 12 also provides full accuracies for additional TinyImageNet and CIFAR-10 experiments in the Appendix.
> >
> > We summarize these results here again for the benefit of the reviewer. Through detailed accuracies, we observe that while SAM usually achieves better retain accuracies and test accuracies, SGD can achieve better forget accuracies as it overfits. This provides insights for our SharpMinMax, which separates a forget model to purposefully overfit for unlearning. Besides ToW performance, we also have MIA and entanglement scores to show the effectiveness of SAM from privacy and representation perspectives.
> >
> > **Re-learning attacks**
> >
> >  At the request of the reviewer F8sS, we additionally present relearning attack experiments to demonstrate SAM's unlearning robustness below. We take the unlearned models to relearn the whole $\mathcal{F}$ for one epoch with a small relearning lr, and measure the increase in forget accuracies. Reported are the averaged increases across $\mathcal{F}\_{\text{high}}, \mathcal{F}\_{\text{mid}}, \mathcal{F}\_{\text{low}}$. We observe that SAM enhanced $\mathcal{U}$ are more resilient to relearning attacks with smaller increases. We will include these results in the final version.
> >
> > **Table**: Averaged increase of forget accuracies after relearning 1 epoch on $\mathcal{F}$ on CIFAR-100 and ResNet50. We observe that SAM enhanced unlearning is consistently more resilient to relearning attacks with less increase on forget accuracy.
> > |             | Relearn lr=0.002 |                 | Relearn lr=0.003 |                 | Relearn lr=0.004 |                 |
> > | ----------- | ---------------- | --------------- | ---------------- | --------------- | ---------------- | --------------- |
> > |             | SGD pretrained   | ASAM pretrained | SGD pretrained   | ASAM pretrained | SGD pretrained   | ASAM pretrained |
> > | NG          | 8.644446667      | 10.33333        | 11.16666667      | 13.25555333     | 12.78889         | 14.7            |
> > | +ASAM       | **8.533333333**  | **9.288886667** | **11.03333667**  | **11.53333**    | 13.02222333      | **13.38889**    |
> > | SharpMinMax | 13.1             | 15.06666667     | 15.58888667      | 17.49999667     | 16.14444667      | 18.51111        |
> > | +ASAM       | **8.33333**      | **8.8**         | **10.66667**     | **11.20000333** | **12.71111**     | **12.66667**    |
> > | RL          | 7.122223333      | 8.555556667     | 8.5              | 9.58889         | 9.622223333      | 10.98889        |
> > | +ASAM       | **6.222223333**  | **7.377776667** | **7.444446667**  | **8.488886667** | **8.366666667**  | **9.46667**     |
> >
> > **How SAM generalizes to other unlearning algorithms**
> >
> > We have demonstrated applying SAM to various recent unlearning algorithms in our study, where SAM consistently improves the base $\mathcal{U}$. As SAM is shown to be an improved optimizer than SGD for unlearning, we believe it can generalize to most gradient-based $\mathcal{U}$ that requires optimization (majority of the unlearning methods). It does not improve training-free $\mathcal{U}$, if any, since no optimization is involved. This is an interesting future direction that requires deeper and broader experimentation like we did in our case, and hence is beyond our current scope. We emphasize that our work principally elucidates SAM’s proficiency for unlearning and we are hopeful this will incite useful discussions and future works in the community.
> >
> > \[1\] Kou, Yiwen, et al. "Benign overfitting in two-layer relu convolutional neural networks." ICML 2023.
> >
> > \[2\] Chen, Zixiang, et al. "Why does sharpness-aware minimization generalize better than sgd?." NeurIPS 2023.
> >
> > \[3\] Fan, Chongyu, et al. "Towards llm unlearning resilient to relearning attacks: A sharpness-aware minimization perspective and beyond." ICML 2025.
> >
> > \[4\] Malekmohammadi, Saber, and Li Xiong. "Sharpness-Aware Parameter Selection for Machine Unlearning." arXiv:2504.06398.
> >
> > \[5\] Zhao, Kairan, et al. "What makes unlearning hard and what to do about it." NeurIPS 2024.

---

### Official Review · Reviewer_pWni · 2025-11-01

**Soundness:** 3
**Presentation:** 2
**Contribution:** 3
**Rating:** 6
**Confidence:** 3

**Summary:**

The paper investigate the idea of sharpness aware optimization for the machine unlearning, they characterize SAM under NegGrad unlearning and the theoretical study of bounding and choosing weight factors. The paper provides a comprehsive theoretical proof to show that SAM can acheive successful unleasrning with significantly smaller damage to the retain accuracy.

**Strengths:**

Theoretical analysis of SAM is robust. The paper has done a good job of analyzing it in relation to Random Labeling and especially NegGrad. I appreciate they were honest about SAM+NegGrad giving worse forget accuracy than SGD+NegGrad before their discussion of overfitting.

**Weaknesses:**

1. UMAP Visualization Clarity
The UMAP visualizations are difficult to interpret. For instance, Figure 1 is intended to illustrate inter- and intra-class movements after unlearning; however, these differences are not visually discernible. The colors used are too similar, and the expected variations are not immediately perceptible. I recommend improving visual clarity by adopting a more distinct color palette, varying marker shapes, or explicitly highlighting changes using arrows, circles, or other annotations. This would make the patterns of change more evident to the reader.

2. ToW Score Chart Interpretation
While the chart presenting the ToW scores supports the claim that SAM is an effective optimization method, it lacks fine-grained detail. Including additional breakdowns or complementary metrics could help the reader better understand how SAM contributes to performance improvements beyond the aggregated ToW score.

3. Reporting Underlying Accuracy Metrics
The paper notes that SAM+NegGrad achieves higher forget-set accuracy than SGD+NegGrad, but these results are only reflected indirectly through the ToW score. It would strengthen the analysis to report the individual accuracy components that contribute to ToW alongside it. Relying solely on a composite metric can obscure nuances in model behavior; providing detailed accuracy values would enable a clearer assessment of the unlearning method’s performance.

4. Lack of Statistical Evaluation
The reported results appear to lack statistical evaluation—no mean or standard deviation values are provided (e.g., in Tables).

**Questions:**

The memorization classes F(high) etc. do not seem to follow a linear pattern in the ToW scores. For instance sometimes mid has the highest score, sometimes mid has the lowest score. Why should we believe that memorization is a worthwhile way to classify these different forget sets if they don't have a clear correlation to increasing difficulty?

---

> ### Author Response · Authors · 2025-11-21
> **[1/2] Thank you for your valuable comments.**
>
> Thank you for your acknowledgement and valuable comments! We would like to clarify some ambiguities with more details, and we will update visualizations and the manuscript soon to improve readability. We appreciate your opinions and would like to engage more.
>
> **Interpreting UMAP visualizations and improve illustration**
>
> Thank you for your feedback; we will re-layout Fig. 1 with clearer class-color coding and per-panel captions explicitly stating the pre/post unlearning takeaways. Concretely. Fig. 1 shows that after NegGrad unlearning, forget samples migrate into wrong class clusters while with SAM the class clusters remain more separable than with SGD; within a representative class "forget" points become more tightly grouped (classwise panels) after unlearning. SGD+NegGrad's more blurred class margins support our claim that it overfits more easily which eventually harms model performance. This visual behavior is consistent with our geometry-aware entanglement trend, where it suggests that using SAM in either pretraining and unlearning will reduce the entanglement between retain and forget samples, making unlearning easier. In a rare case, SGD+NegGrad can achieve a lower within-class entanglement between retain and forget samples given the absence of categorical information (same class label) and more overfitting, but in most cases, as SAM inherently separates modeled signal (retain samples) and noise (forget samples) into two geometrics, it brings lower entanglement between retain and forget samples both for "all classes" and "within class". Please see Fig. 6,7,8 at the end of Appendix for complete visualizations with different $\mathcal{U}$ and $\mathcal{F}$ captioned with corresponding entanglement scores.
>
> **Fine-grained details and accuracies behind ToW not provided**
>
> All underlying accuracies used to compute ToW are already reported. Tab 8-10 in the Appendix provide per-method retain, forget, test accuracies across CIFAR-100 and ImageNet which directly produce ToW scores in Tab1,3 in the main text. For example, on CIFAR-100 SAM consistently lifts retain and test accuracies (and hence ToW) across NegGrad/RL/SalUn, while SGD sometimes attains lower forget accuracy (ie. more forgetting) due to overfitting – precisely the empirical pattern that motivated SharpMinMax. Besides ToW performance, we also have MIA and entanglement scores to show the effectiveness of SAM from privacy and representation perspectives.
>
> **Statistical evaluation**
>
> We already have bar plots (Fig. 3,4) of statistical evaluation for main results in Tab. 1,3 in App. F.1 in the Appendix, where we plot means as bars and draw $2\sigma$ as error bars, illustrating $95\\%$ confidence interval. The numerical results are reported in Tab. 7. We observe that SAM consistently improves all unlearning methods with more noticeable results on CIFAR-100. For "All methods" subplots, we also highlight the largest improvement by applying SAM to each $\mathcal{U}$. Despite that some $\mathcal{U}$ brings larger fluctuations for both SAM and SGD (e.g. SCRUB), we observe that SAM shifts means upwards and reduces variance and is hence more stable, strengthening the reliability of the observed gains in Tab. 1.

---

> > ### Author Response · Authors · 2025-11-21
> > **[2/2] Thank you for your valuable comments.**
> >
> > **Non-monotonic behavior on Memorization scores**
> >
> > We thank the reviewer for bringing this up as this is a subtle point. Our experiments and past work like \[1\] show that unlearning difficulty is reflected by two independent factors: memorization and entanglement. There are several factors affecting this correlation. First, in Tab. 3 in \[1\], they also observe that $\mathcal{F}\_{\text{mid}}$ has the largest entanglement scores for CIFAR, meaning that it can be challenging from another perspective. Another factor is the distributions of memorization scores. In Tab. 1, we observe linear correlations with $\mathcal{F}\_{\text{high}}<\mathcal{F}\_{\text{mid}}<\mathcal{F}\_{\text{low}}$ in terms of ToW for ImageNet experiments. ImageNet's memorization scores has a smooth long-tailed distribution with enough samples for each score range, yielding $\mathcal{F}$ with smaller variances that better reflect unlearning difficulty. However, the distribution of CIFAR-100's memorization scores has a spike for high memorization \[2,3\]. Then, for $N=3000$ ($\approx 6\\%$ trainset), while $\mathcal{F}\_{\text{high}}, \mathcal{F}\_{\text{low}}$ can sample enough points with scores close to $1,0$, $\mathcal{F}\_{\text{mid}}$ has memorization scores spanned from $0.4$ to $0.6$ respectively, which has the largest variance \[2,3\] and is also harder to separate from $\mathcal{R}$. Third, \[1\] proposes to apply different $\mathcal{U}$ for $\mathcal{F}$ of different difficulties as they observe that different unlearning methods unlearn either aggressively or mildly and behave differently given $\mathcal{F}$. This also indicates that some methods will perform better on $\mathcal{F}\_{\text{high}}$ while other methods will perform better on $\mathcal{F}\_{\text{low}}$.  Fourth, ToW aggregates three potentially competing goals and the optimizer may interact differently with each regime’s geometry. Thus, while we generally observe that $\mathcal{F}\_{\text{high}}<\mathcal{F}\_{\text{low}}$ in terms of ToW, results on $\mathcal{F}\_{\text{mid}}$ might not follow a linear correlation.
> >
> > \[1\] Zhao, Kairan, et al. "What makes unlearning hard and what to do about it." NeurIPS 2024.
> >
> > \[2\] Feldman, Vitaly, and Chiyuan Zhang. "What neural networks memorize and why: Discovering the long tail via influence estimation." NeurIPS 2020.
> >
> > \[3\] Memorization scores by Feldman et al. https://pluskid.github.io/influence-memorization/

---

### Author Response · Authors · 2025-12-03
**[1/3] Summary of Rebuttal and Manuscript Edits**

We sincerely thank all the reviewers for their constructive advice, from new experiments to inspiring discussions. We appreciate that all reviewers find our theoretical analysis robust and interesting, and our empirical studies comprehensive. Reviewers have shown interests in details and extended experiments and evaluations, which we have all addressed. We have updated our manuscript to improve clarity, provide better explanations, and discuss new experiments, with new edits highlighted in blue.

---

> ### Author Response · Authors · 2025-12-03
> **[2/3] Summary of Rebuttal and Manuscript Edits**
>
> ## Rebuttal Summary
>
> **Reviewer pWni - initial score: 6, positive trend, requesting details**
>
> - Acknowledged the importance of  theoretical analysis, and observations of how SGD's overfitting helps it achieve better forget accuracies (despite degraded retain/test accuracies), which we further leverage to build our Sharp MinMax.
> - Suggested more clear UMAP visualizations, which we agree and have updated
> - Asked for detailed accuracies and discussions behind ToWs, which we pointed out that we have already put in Appendix (Tab. 8-10, 13).
> - Requested statistical evaluation, which we pointed out that we have in App. F.1 with plots and tables.
> - Concerned about determining unlearning difficulty by memorization levels, which we clarified with previous work and explanation on the interplay between other factors that lead to non-monotonic behaviors on $\mathcal{F}_{\text{mid}}$ specifically.
>
> **Reviewer U1T3 - initial score 4, no reply**
>
> - Praised that our theoretical analysis is "in-depth".
> - Acknowledged our experiments "contain sufficient evidence".
> - Suggested improve readability and simplify notations, which we have reflected in our updated manuscript.
> - Concerned about the novelty of SAM in unlearning, but did not cite any previous work refuting our novelty. We explained the novelty of our new theoretical findings for SAM unlearning and clarified why retaining signal strength is important to efficient unlearning. We also compared two concurrent work related to SAM unlearning and pointed out that only our work is the first to systematically study theoretically and empirically the geometric properties and unlearning dynamics of SAM optimization, and proposes new algorithms based on our findings.
> - Asked for detailed accuracies and discussions behind ToWs, which we pointed out that we have already put in Appendix (Tab. 8-10, 13).
> - Asked for more evaluation metrics, which we provided with relearning attack results (and we have more in following rebuttals).
>
> **Reviewer F8sS - initial score 4, seeking additional experiments, which we have added**
>
> - Praised that our theoretical analysis based on signal-noise framework is novel.
> - Acknowledged our proposed Sharp MinMax to be novel
> - Acknowledged our in-depth evaluation across datasets and metrics.
> - Requested runtime analysis, which we have provided. We also pointed out that while efficiency is not our focus (we study geometric properties of SAM), more recent SAM variants can behave equivalently as vanilla SAM with less compute, and we demonstrated it by running momentum SAM (MSAM) experiments.
> - Requested relearning attacks, which we have provided. These additional results highlight the robustness of our proposed method. We also clarified that the base case for Sharp MinMax adopts SGD on both retain and forget models (so only weight masking applied), and experimented with SAM on retain model, and SGD on forget model which provides slight improvement to the baseline.
> - Was curious about weight mask design and cutoff choice, in which we explained our intuition based on the over-parameterization scheme, and performed ablation studies on weight mask cutoffs on CIFAR-100. We also pointed out our difference between SalUn and SalUn uses a large forget model (50%).
>
> **Reviewer Ti1T - initial score 6, positive reply (happy to adjust score after manuscript edits), interesting discussions on ascent-based unlearning**
>
> - Acknowledged our new theoretical findings to be interesting.
> - Praised that our Sharp MinMax based on theoretical and empirical findings is effective and interesting.
> - Acknowledged our extensive experimental evaluations.
> - Showed interest in our characterization of the $\alpha$ factor in NegGrad.
> - Asked for clarifying that the unlearning assumption of misclassifying forget samples in our work applies to ascent-based unlearning only, which we agree and revised manuscript for precision.
> - Requested evaluating SAM unlearning by distribution closeness to the retrained model (KLoM metrics), which we have provided and the observations are similar to ToWs results - SAM improved KLoM, and ameliorates data dependencies.The reviewer responded positively to these new results, and suggested increasing the score after we edit the manuscript.
> - Discussed potential limitations of ascent-based unlearning. While our work shows that SAM's benefits can apply to other unlearning algorithms (Eqn. 3 for RL, and extensive empirical results) and ascent-based unlearning is widely used in frontier research, we have noticed the misaligned objective and agreed with reviewer Ti1T. We both agree that it is beyond the scope of this paper, and we have included it in App. B.3 Limitations and Future Works.

---

> ### Author Response · Authors · 2025-12-03
> **[3/3] Summary of Rebuttal and Manuscript Edits**
>
> ## Manuscript changes
>
> **Simplify notations**: we have made some minor changes to simplify notations in the main paper for readability, abbreviating subscripts for binary class and convolution filters, and replacing (epoch, batch) superscripts with a time vector $\mathbf{t}$.
>
> **Clarify NegGrad objective**: we have further clarified that ascent-based unlearning (NegGrad) aims at misclassification and there are other unlearning algorithms with different objectives. While our study provides a refined analysis of ascent-based unlearning, we further discuss potential limitations and future work in App. B.3.
>
> **Clarify sharpness maximization**: we revised Eqn. 8 to note that we only apply the sharpness maximization to the forget model splitted by weight masking; we also introduce notation for retain, forget models as $\mathbf{W}\_{\mathcal{R}},\mathbf{W}\_{\mathcal{F}}$ for convenience.
>
> **Relearning attacks**: we include an additional experiment of relearning attacks, where we put discussions in Sec. 4.1 and table results in App. G.5. These experiments highlight the robustness of unlearning under our framework.
>
> **KLoM**: we measure KL on distribution margins between unlearned model and retrained model to measure closeness, where we discuss in Sec. 4.1 and results in App. G.7.
>
> **Efficient SAM**, we experiment with momentum SAM to demonstrate how recent faster SAM variants can perform as well as vanilla SAM with much less compute, where we put summary in Sec. 4.1 and table results in App. G.6.
>
> **Weight mask cutoff**: we incorporate our discussions about weight mask cutoff for Sharp MinMax in App. E.2 and also provide ablation studies on cutoff choices.
>
> **Feature visualization**: we update UMAP visualizations (Fig. 1 and Appendix) with better clarity with longer fitting and more contrast colors to better demonstrate our observations.
>
> **Loss landscapes**: we move one subplot of loss landscapes from Appendix to Sec. 4.3 to better illustrate the maintained flatness of SAM unlearning.

---

### Meta-Review · Area_Chair_8xxG · 2026-01-06

**Summary:**

The paper investigates the application of Sharpness-Aware Minimization (SAM) within the NegGrad machine unlearning framework. The authors provide a robust theoretical analysis using a signal-noise decomposition framework to show how SAM can effectively distinguish between retain and forget sets, particularly by exploiting the tendency of the forget set to overfit. Building on these insights, the paper introduces "Sharp MinMax," a novel optimization method that minimizes sharpness for the retain set while maximizing it for the forget set to facilitate effective unlearning. Experimental results across multiple datasets demonstrate that SAM-augmented methods can achieve competitive unlearning performance with significantly less damage to the accuracy of the retained data compared to traditional SGD-based approaches.

**Reviewer Concerns:**

Initial reviews highlighted several critical weaknesses, primarily concerning the choice of evaluation metrics and missing baselines. Reviewers (U1T3, F8sS, Ti1T) were concerned about the over-reliance on the "Time-to-Whiten" (ToW) metric, which is non-standard, and requested evidence using established measures such as Membership Inference Attack (MIA) resistance and relearning robustness. Furthermore, there were requests for a "SAM-on-retain-only" baseline to justify the necessity of the "MinMax" approach, and a need for computational efficiency analysis given the nested optimization structure of Sharp MinMax. Questions were also raised regarding the clarity of UMAP visualizations and certain grammatical inconsistencies in the original manuscript.

**Reviewer Scores:**

The paper received an initial score distribution of 6, 4, 4, 6. In the rebuttal, the authors successfully addressed the most significant technical gaps by supplementing the evaluation with standard unlearning metrics (MIA, KLoM, and relearning resistance) and providing a detailed runtime analysis. Crucially, the authors included the "SAM-retain-only" baseline, which demonstrated that maximizing sharpness on the forget set provides a distinct advantage in forgetting efficacy. The revised manuscript also featured improved visualizations and corrected notations, resolving the presentation issues noted by the reviewers. Given that the supplemental experiments effectively bridge the gap between the paper's theoretical novelty and practical utility, the Area Chair recommends Acceptance.

---

### Decision · Program_Chairs · 2026-01-26

Accept (Poster)